# Peptidoglycan-dependent NF-κB activation in a small subset of brain octopaminergic neurons controls female oviposition

Ambra Masuzzo[1], Gérard Manière[2], Annelise Viallat-Lieutaud[1], Émilie Avazeri[1], Olivier Zugasti[1], Yaël Grosjean[2], C Léopold Kurz[1]*, Julien Royet[1]*

[1]Aix-Marseille Université, CNRS, IBDM, Marseille, France; [2]Centre des Sciences du Goût et de l'Alimentation, AgroSup Dijon, CNRS, INRA, Université Bourgogne Franche-Comté, Dijon, France

**Abstract** When facing microbes, animals engage in behaviors that lower the impact of the infection. We previously demonstrated that internal sensing of bacterial peptidoglycan reduces *Drosophila* female oviposition via NF-κB pathway activation in some neurons (Kurz et al., 2017). Although we showed that the neuromodulator octopamine is implicated, the identity of the involved neurons, as well as the physiological mechanism blocking egg-laying, remained unknown. In this study, we identified few ventral nerve cord and brain octopaminergic neurons expressing an NF-κB pathway component. We functionally demonstrated that NF-κB pathway activation in the brain, but not in the ventral nerve cord octopaminergic neurons, triggers an egg-laying drop in response to infection. Furthermore, we demonstrated via calcium imaging that the activity of these neurons can be directly modulated by peptidoglycan and that these cells do not control other octopamine-dependent behaviors such as female receptivity. This study shows that by sensing peptidoglycan and hence activating NF-κB cascade, a couple of brain neurons modulate a specific octopamine-dependent behavior to adapt female physiology status to their infectious state.
DOI: https://doi.org/10.7554/eLife.50559.001

**\*For correspondence:**
leopold.kurz@univ-amu.fr (CLK);
julien.royet@univ-amu.fr (JR)

**Competing interests:** The authors declare that no competing interests exist.

## Introduction

Since eukaryotes live in an environment heavily contaminated by microorganisms, they have forged, over time, extremely complex relationships. Some of these interactions are affecting tissues and organs other than those whose function is to directly eliminate invading microorganisms. Along these lines, growing evidence indicates that bidirectional communication between the gut microbiota and the Central Nervous System (CNS) impacts host behaviors including anxiety, cognition, nociception and social interactions (*Cryan and O'Mahony, 2011*; *Sharon et al., 2016*). Moreover, by modifying its behavior, an infected host can lower some of its physiological activities to concentrate its energy on pathogen elimination (*Adamo, 1999*; *Adamo, 2014*). On the other hand, manipulating host behavior is a way for microbes to reduce the defenses of their hosts (*Elya et al., 2018*; *Keesey et al., 2017*). The notion that hosts can react to microbes by changing their behaviors is called behavioral immunity and refers to a suite of mechanisms that allows organisms to detect the potential presence of disease-causing invaders and to engage in comportments that reduce the consequences of the infection at the level of the organism, the group and/or its offspring (*de Roode and Lefèvre, 2012*; *Müller and Pawelec, 2014*). Although reported for a long time in invertebrates and mammals, only recent studies, mainly in *D. melanogaster* and *C. elegans*, have started to unravel the molecular aspects of these peculiar host-microbe interactions and especially to directly link

behavioral changes and immune activation (*Cao et al., 2017*; *Kacsoh et al., 2015a*; *Kacsoh et al., 2015b*; *Kacsoh et al., 2013*; *Lee and Mylonakis, 2017*; *Toda et al., 2019*; *Yanagawa et al., 2014*; *Zhai et al., 2018*). In a previous work, we have shown that some components of the NF-κB signaling cascade, which represents a major immune module in both invertebrates and vertebrates, are expressed outside classical immune organs and more precisely in some cells of the brain and the Ventral Nerve Cord (VNC) (*Kurz et al., 2017*). By performing functional studies, we demonstrated that the detection of a universal bacterial metabolite, called peptidoglycan (PGN), by neurons reduces female oviposition. Moreover, we demonstrated that octopamine (OA)-producing neurons which regulate many behaviors in flies, including oviposition, are also implicated in tuning egg-laying rate in response to bacteria (*Kurz et al., 2017*).

In the present study, we used genetic intersectional strategy to precisely map the neurons that, upon peptidoglycan sensing, trigger a reduction of female oviposition. Our results demonstrated that, out of around 20 neurons distributed in the brain and the VNC and expressing the immune modulator PGRP-LB (PeptidoGlycan-Recognition-Protein-LB), only a few are octopaminergic. We further demonstrated that peptidoglycan sensing and NF-κB activation in the VM III octopaminergic neuronal sub-cluster present in the brain is sufficient to modulate egg-laying, in infected females. Using calcium imaging, we showed that stimulation of brains by purified peptidoglycan blocks VM III octopaminergic neurons activity. Finally, our data are consistent with a model in which this peptidoglycan-dependent inhibition of brain neuronal activity impairs a specific ovulation event, called follicle trimming, hence functionally linking peptidoglycan detection to the reduction of egg production in infected females.

## Results

### The neuronal subpopulation of pLB1+ cells regulates egg-laying

pLB1-Gal4 is a reporter fly line that potentially recapitulates the *in vivo* expression pattern of one isoform of the immune regulator PGRP-LB (*Kurz et al., 2017*). By digesting bacteria-derived peptidoglycan inside the cells, this enzyme reduces the impact of peptidoglycan-dependent NF-κB signaling in cells that express it, thus acting as a negative regulator of the signaling cascade (*Charroux et al., 2018*). We have previously shown that cells expressing Gal4 in the pLB1 pattern (called pLB1+ cells) regulate egg-laying behavior in response to bacterial infection. The fact that the pLB1 expression pattern in the adult CNS delineates a network (*Figure 1A–B*) and that ectopic expression of proteins able to modify neuronal activity (such as Tetanus Toxin (TTx), Kir2.1 or Transient Receptor Potential cation channel, subfamily A, member 1 (TRPA1)) in these cells was sufficient to impact female egg-laying, suggested that at least some of the pLB1+ cells are neurons able to modulate oviposition (*Kurz et al., 2017*). However, since the pLB1-Gal4 line is also expressed in non-neuronal cells such as enterocytes or pericardiac cells (*Charroux et al., 2018*), we decided to confirm the neuronal identity of CNS-resident pLB1+ cells using imaging and functional assays. For that purpose, we used the flip-out method that allowed us to observe cells simultaneously positive for pLB1 and the pan-neuronal marker synaptobrevin (nSyb; nSyb>FLP/pLB1>stop>mGFP)(*del Valle Rodríguez et al., 2011*). This strategy confirmed the presence of a pLB1+ neuronal circuit in the brain and the VNC (*Figure 1C–D*) and outlined the position of the cell bodies. Considering data from the pLB1-Gal4 expression pattern as well as the intersectional strategy from multiple animals, we generated a map (*Figure 1E*) and a table (*Table 1*) with neuronal fibers and cell bodies of pLB1+ neurons. We detected pLB1+ neuronal projections in the SEZ of the brain (*Figure 1A*). In addition, the intersectional strategy using nSyb-LexA revealed, in the majority of the brains (12/20), a single pLB1+ neuron in the posterior part of the SEZ (*Figure 1C*) and few pLB1+ neurons in the same brain area in a minority of samples (5/20) (*Table 1*). In the VNC, the expression pattern was highly stereotyped with neuronal fibers present in all the segments, from the anterior thoracic segment (T1) to the Abdominal Ganglia (AbdG) (*Figure 1B and E*). From the analyses of all the samples (13/13), a network composed of 12 neurons and two isolated cell bodies localized in the posterior thoracic segment (T3) and the AbdG could be defined (*Table 1*, *Figure 1D–E*).

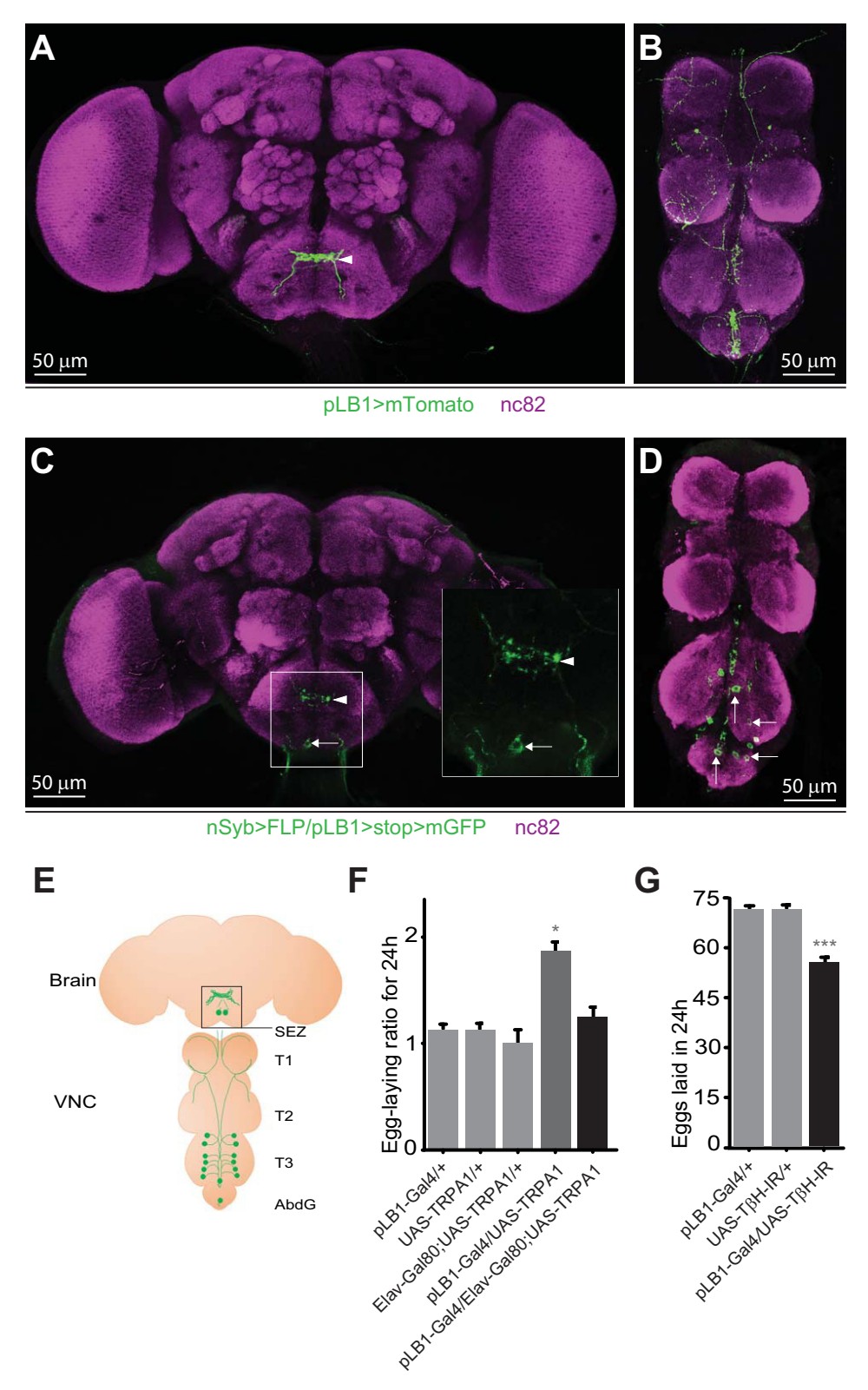

**Figure 1.** pLB1 is expressed in neurons modulating egg-laying via octopamine. (**A, B**); Immunodetection of cells expressing pLB1-Gal4/UAS-Tomato-mCD8 (pLB1>mTomato) in females. For the homogeneity of the different images, the red signal corresponding to Tomato-mCD8 was converted in green. In the brain (**A**), pLB1 is expressed in the Sub Esophageal Zone (SEZ) (arrowhead). In the ventral nerve cord (VNC) (**B**), the network links the brain to T1, T2, T3 and the Abdominal Ganglia (AbdG).(**C, D**); Pattern of cells co-expressing the neuronal markers nSyb and the pLB1 driver (nSyb>FLP/

*Figure 1 continued on next page*

*Figure 1 continued*

pLB1>stop>mGFP). The GFP can only be expressed if the stop sequence inserted upstream of the *gfp* gene is flipped-out in pLB1+/nSyb+ cells. In the brain (**C**), neuronal projections are found in the SEZ (arrowhead) as well as a cell body (arrow). The inserted box is a magnification of the SEZ. In the VNC (**D**), 8 to 14 cell bodies are revealed in T3 and AbdG.(**E**); Map representing projections and cell bodies of neurons expressing pLB1 in brain and VNC. (**F**); Preventing the expression of the transient receptor potential cation channel, subfamily A, member 1 (TRPA1) in pLB1+ neurons impairs the egg-laying increase in non-infected mated females. The egg-laying ratio for 24 hours (24 h) corresponds to the number of eggs laid by a female at the restrictive temperature (29°C) over the average number of eggs laid by females of the same genotype at the permissive temperature (23°C). (**G**); RNAi-mediated inactivation of the octopamine-producing enzyme TβH in pLB1+ cells reduces egg-laying. For (**A–D**), immuno-staining against the neuropil marker nc82 was used to stain the organ. For (**F**), shown is the average egg-laying ratio per 24 hours ± SEM (29°C/23°C) from at least two independent trials with at least 16 females per genotype and condition used. An egg-laying ratio of 1 indicates an absence of difference between the test and the control. For (**G**), shown is the average number of eggs laid per fly per 24 hours ± SEM from at least three independent trials with at least 58 females per genotype and condition used. * indicates p<0.05, *** indicates p<0.0001; non-parametric ANOVA, Dunn's multiple comparison test. Details including n values and genotypes can be found in the detailed lines, conditions and, statistics for the figure section.

DOI: https://doi.org/10.7554/eLife.50559.002

The following source data is available for figure 1:

**Source data 1.** Egg laying raw data for *Figure 1F*.
DOI: https://doi.org/10.7554/eLife.50559.003
**Source data 2.** Egg laying raw data for *Figure 1G*.
DOI: https://doi.org/10.7554/eLife.50559.004

## Some pLB1+/octopaminergic neurons control egg-laying in response to infection

We have previously shown that the over-expression of TRPA1 (an ion channel increasing the excitability of neurons) in pLB1+ cells, enhances oviposition rate (*Kurz et al., 2017*). In order to demonstrate that the identified pLB1+ neurons functionally regulate egg-laying, we tested whether TRPA1 over-expression effect on egg-laying could be suppressed by the expression of the pan-neuronal Gal4 inhibitor, Elav-Gal80 (*Figure 1F*). Whereas TRPA1 expression in pLB1+ cells doubled the number of eggs produced by non-infected females compared to controls when shifted from 23°C to 29°C (from 40 eggs per day to 80 on average, 29°C/23°C ratio of 1.84), this effect was completely suppressed by co-expressing the Elav-Gal80 transgene. These data demonstrated that pLB1+ cells activation modulates egg-laying and that among the pLB1+ cells, a neuronal subgroup is responsible for this control. Our published results demonstrated that peptidoglycan-dependent NF-κB activation in octopaminergic neurons inhibits female egg-laying upon bacterial infection and that over-expressing the OA-producing enzyme Tyramine-β-hydroxylase (TβH) in pLB1+ cells counteracts this phenotype (*Kurz et al., 2017*). However, whether the peptidoglycan effect on egg-laying was mediated cell-autonomously via the modulation of OA signaling in pLB1+ cells remained an open question. The fact that some pLB1+ cells were neurons, led us to hypothesize that these neurons could be octopaminergic and thus able to modulate egg-laying via this neurotransmitter. To test this hypothesis, we analyzed the effects of reducing OA production in pLB1+ cells on egg-laying (*Figure 1G*). OA is synthesized from tyrosine by the sequential actions of Tyrosine decarboxylase (Tdc2) and TβH. Flies in which TβH RNAi was overexpressed in pLB1+ cells (UAS-TβH-IR), and hence OA production reduced, laid significantly fewer eggs than control flies. This result demonstrated that some pLB1+

**Table 1.** Number and position of GFP-positive cells in the CNS of nSyb>FLP/pLB1>stop>mGFP flies.
The cells positive for GFP are counted.

| Organ | pLB1+ neurons | N events/total flies |
|---|---|---|
| Brain | 0 | 3/20 |
| | 1 in the SEZ | 12/20 |
| | 2-5 in the SEZ | 3/20 |
| | 6-10 in the SEZ | 2/20 |
| VNC | 8-14 in T3 and AbdG | 13/13 |

DOI: https://doi.org/10.7554/eLife.50559.071

neurons are indeed octopaminergic and that OA itself is implicated in the ability of these neurons to control egg-laying. Consistently, data from brain single-cell transcriptomics showed that some, but not all, brain Tdc2+ neurons also express mRNAs coding for NF-κB pathway components (*Davie et al., 2018*).

## pLB1+/Tdc2+ neurons are present in the brain VM III sub-cluster and the VNC

To precisely map pLB1+ octopaminergic neurons, we stained brains and VNCs of nSyb>FLP/pLB1>stop>mGFP adult flies with an antibody against Tdc2 (*Figure 2A–B'*). This strategy identified very few pLB1+/Tdc2+ neurons in the brain SEZ (*Figure 2A–A"* and *Table 2A*). Complementary intersectional experiments between pLB1-Gal4 and Tdc2-LexA drivers, together with immunostaining against Tdc2, confirmed the existence in the majority of the stained brains (15/19) of either one or two pLB1+/Tdc2+ neurons in the posterior part of the SEZ (*Figure 2* and *Figure 2—figure supplement 1* and *Table 2B*). Using the same strategy as for the brain, we identified in 12 out of 32 flies tested, between one to three pLB1+/Tdc2+ neurons in the posterior region of the VNCs (*Figure 2B–B'* and *Table 2A* and *Figure 2—figure supplement 1*). In the absence of identified markers for the different octopaminergic clusters, it was difficult to unambiguously map these neurons. However, the position of the pLB1+ neurons along the dorso-ventral axis relative to Tdc2+ cells indicated that pLB1+/Tdc2+ neurons were part of the ventral midline (VM) cluster (*Figure 2—figure supplement 2*) (*Busch et al., 2009*). The latter can be further subdivided into three sub-clusters, based on their position along the antero-posterior axis (VM I, VM II, VM III respectively) (*Busch et al., 2009*; *Schneider et al., 2012*). The single pLB1+/Tdc2+ neuron observed in most of the cases always belonged to the VM III sub-cluster (*Figure 2* and *Figure 2—figure supplement 3* and *Table 2A, B*). These data demonstrated that few brain octopaminergic neurons are pLB1+ and that this sub-population specifically belongs to the VM cluster with an emphasis for a single neuron in the VM III sub-cluster. Since Tdc2 is a marker for both tyraminergic and octopaminergic neurons, it is important to highlight that all Tdc2+ VM neurons have been shown to produce octopamine (*Schneider et al., 2012*). Although 8–14 pLB1+ neurons are reproducibly detected in the VNC, very few are octopaminergic (*Table 2A*). Interestingly, the pLB1+/Tdc2- cells of the T3 segment of the VNC seemed to be 12 interconnected neurons. In an attempt to characterize these non-octopaminergic pLB1+ neurons, we used specific Abs for neuropeptides expressed in a similar location in the VNC. These experiments demonstrated that pLB1+ VNC cells are neither Allatostatin A (*Chen et al., 2016*), Bursicon (*Peabody et al., 2008*), CCAP (Crustacean cardioactive peptide) (*Luan et al., 2006*), nor Leucokinin (*de Haro et al., 2010*) producing neurons (*Figure 2—figure supplements 4–7*).

## pLB1+ neurons selectively control octopaminergic-dependent behaviors

It has been shown that a small subset of Tdc2 and Doublesex (Dsx) positive neurons (Tdc2+/Dsx+) present in the AbdG modulate OA-dependent behaviors, such as female receptivity, male rejection and egg deposition (*Rezával et al., 2014*). Since pLB1+ neurons also regulated egg deposition in mated females via OA (*Figure 1G*), as well as in virgins (*Kurz et al., 2017*), we tested whether these cells were involved in controlling other OA-dependent behaviors. As previously reported, we confirmed that virgin females in which Tdc2 neurons are inactivated (via the overexpression of Kir2.1, a potassium channel that hyperpolarizes neurons), presented an increased receptivity. Indeed, Tdc2-Gal4/UAS-Kir2.1 virgin presented a higher percentage of copulation and a lower latency than controls (*Rezával et al., 2014*) (*Figure 3A–B*). Furthermore, although not statistically significant, we observed a trend which suggested that the remating index of Tdc2-Gal4/UAS-Kir2.1 mated females was higher than in control flies (*Figure 3—figure supplement 1*), confirming previously published data (*Rezával et al., 2014*). Conversely, when we analyzed pLB1-Gal4/UAS-Kir.2.1 females, none of these OA-dependent behaviors were affected (*Figure 3A–B*, *Figure 3—figure supplement 1*). These results suggested that pLB1+/Tdc2+ neurons are different from the ones regulating receptivity and post-mating behaviors in physiological conditions. Consistently, pLB1+/Tdc2+ neurons were also detected in males, whereas Tdc2+/Dsx+ ones are sexually dimorphic and absent in adult males (*Figure 3C–D'* and *Figure 3—figure supplements 2* and *3*) (*Rezával et al., 2014*). Besides intersectional strategy experiments using Dsx-FLP and Dsx-LexA drivers demonstrated that pLB1+ neurons

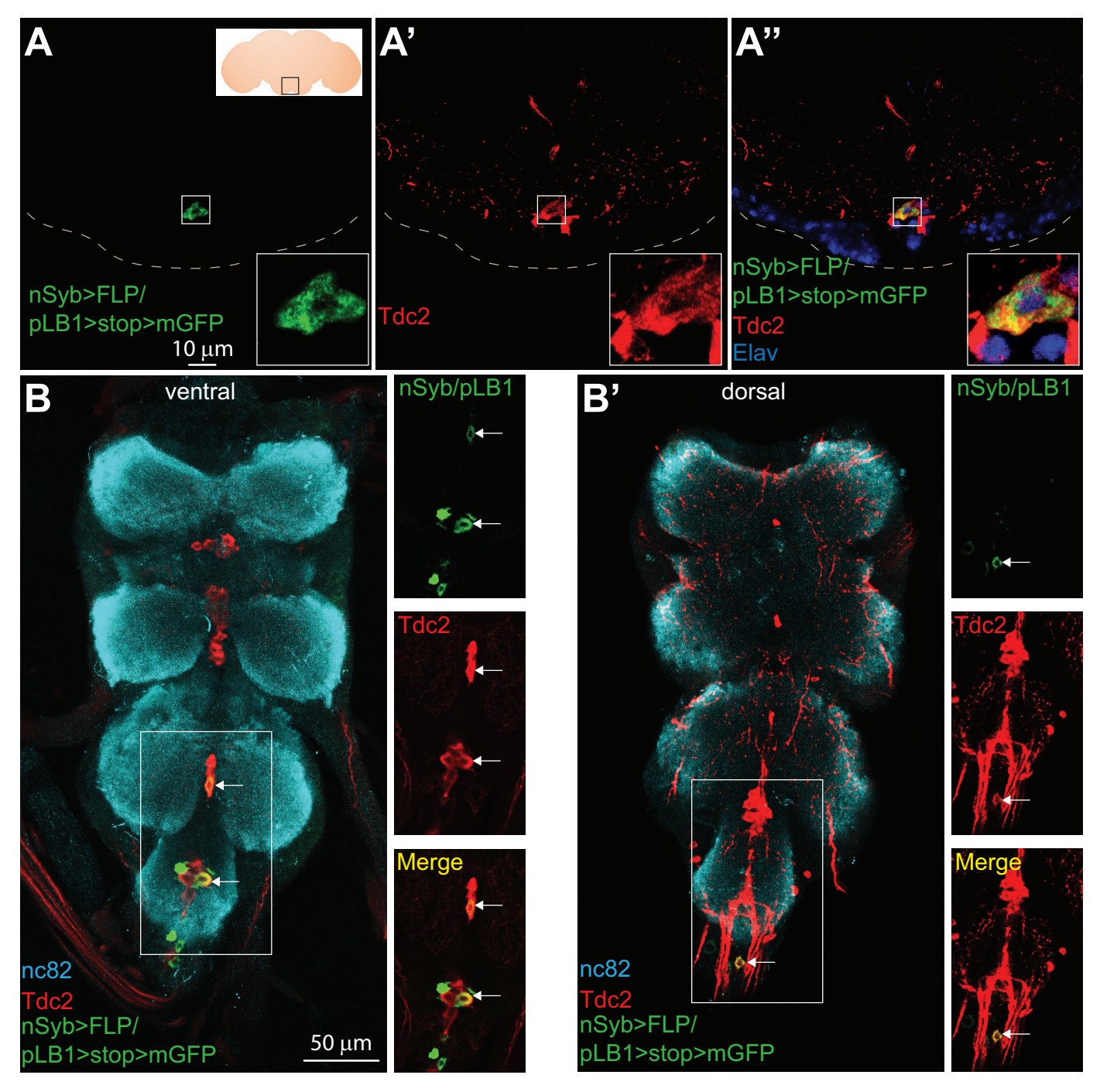

**Figure 2.** Some of the pLB1+ neurons are octopaminergic. (**A–B'**); Immuno-detection in the brain Sub Esophageal Zone (SEZ; **A–A''**) and ventral nerve cord (VNC; **B–B'**) of neurons expressing pLB1 (nSyb>FLP/pLB1>stop>mGFP) (**A,B and B'**) and producing the Tdc2 enzyme (**A', B and B'**). In (**A''**), the nuclear neuronal marker Elav was also immuno-detected. For (**A**), the inserted scheme represents the brain and the empty black square delineates the observed area. For (**A-A''**), the inserted box is a magnification of the outlined box and the dashed line represents the ventral limit of the brain. For (**B-B'**), staining against nc82 was used to delineate the shape of the VNC. For (**B–B'**), the merged channels of the outlined box are separated on the images on the right. Arrows point to pLB1+/Tdc2+ cells in the VNC. Details including genotypes can be found in the detailed lines, conditions and, statistics for the figure section.

DOI: https://doi.org/10.7554/eLife.50559.005

The following figure supplements are available for figure 2:

**Figure supplement 1.** Some octopaminergic neurons are pLB1+.

*Figure 2 continued on next page*

*Figure 2 continued*

DOI: https://doi.org/10.7554/eLife.50559.006

**Figure supplement 2.** Map of Tdc2 expressing neurons in the brain and VNC.

DOI: https://doi.org/10.7554/eLife.50559.007

**Figure supplement 3.** The pLB1+ octopaminergic neurons in the brain belong to the VM III sub-cluster.

DOI: https://doi.org/10.7554/eLife.50559.008

**Figure supplement 4.** pLB1-expressing neurons in the VNC do not produce Allatostatin A.

DOI: https://doi.org/10.7554/eLife.50559.009

**Figure supplement 5.** pLB1+ neurons in the VNC do not produce Bursicon.

DOI: https://doi.org/10.7554/eLife.50559.010

**Figure supplement 6.** pLB1+ neurons in the VNC do not produce CCAP.

DOI: https://doi.org/10.7554/eLife.50559.011

**Figure supplement 7.** pLB1+ neurons in the VNC do not produce Leucokinin.

DOI: https://doi.org/10.7554/eLife.50559.012

in the brain and the VNC were Dsx - (*Figure 3E–F'* and *Figure 3—figure supplement 4*). Altogether, these results demonstrated that the pLB1+/Tdc2+ neurons are different from the Tdc2+/Dsx + neurons that control female receptivity and post-mating behavior, including egg deposition, in uninfected flies.

## NF-κB activation in pLB1+/Tdc2+ neurons reduces egg-laying rate of infected females

The above results led us to functionally test whether pLB1+/Tdc2+ neurons were i) able to control oviposition rate and ii) the ones that modulated egg-laying rate upon peptidoglycan sensing via the NF-κB pathway. For that purpose, we tested the consequences of functionally inactivating pLB1+/ Tdc2+ neurons using an intersectional strategy that combined Gal4/UAS, LexA/Lex-Aop and a flip-able Gal80 inhibitor (*del Valle Rodríguez et al., 2011*). As previously shown, the ectopic expression of the neuronal inhibitor TTx in pLB1+ cells strongly reduced female egg-laying (*Figure 4A*) (*Kurz et al., 2017*). While this effect was suppressed by the ubiquitous expression of the Gal80

**Table 2.** Number and position of pLB1+/Tdc2+ neurons.

(A); pLB1-Gal4, UAS>stop>GFPmCD8; LexAop-FLP/nSyb-LexA brains and ventral nerve cords (VNCs) stained with an anti-Tdc2 Ab. The cells positive for GFP and stained with the anti-Tdc2 Ab (pLB1+/ Tdc2+) are counted (left). The cells positive for GFP and negative for the Tdc2 Ab (pLB1+/Tdc2-) are counted. (B); pLB1-Gal4, UAS>stop>GFPmCD8; LexAop-FLP/Tdc2-LexA brains and VNCs stained with an anti-Tdc2 Ab. The GFP+ cells (pLB1+/Tdc2+) also positive for Tdc2 Ab are counted (left). NR = non relevant. This intersectional strategy only reveals pLB1+/Tdc2+ cells.

| (A) Organ | pLB1+/Tdc2+ neurons | pLB1+/Tdc2- neurons | N events/total flies |
|---|---|---|---|
| | (Strategy 1, see legend for details) | | |
| Brain | 0 | 0 | 1/18 |
| | 1 in the VMV III cluster | 0 | 6/18 |
| | 1 in the VMV III cluster | 2-4 in the SEZ | 3/18 |
| | 2-5 in the VMV III cluster | 0 | 8/18 |
| VNC | 0 | 8-14 in T3 and AbdG | 11/14 |
| | 1 in the AbdG and 2 in T3 | 8-14 in T3 and AbdG | 3/14 |
| **(B) Organ** | **pLB1+/Tdc2+ neurons** | **pLB1+/Tdc2- neurons** | **N events/total flies** |
| | (Strategy 2, see legend for details) | | |
| Brain | 0 | NR | 4/19 |
| | 1 in the VMV III cluster | NR | 13/19 |
| | 2 in the VMV III cluster | NR | 2/19 |
| VNC | 0 | NR | 9/18 |
| | 1 in the AbdG | NR | 7/18 |
| | 2 in the AbdG | NR | 2/18 |

DOI: https://doi.org/10.7554/eLife.50559.072

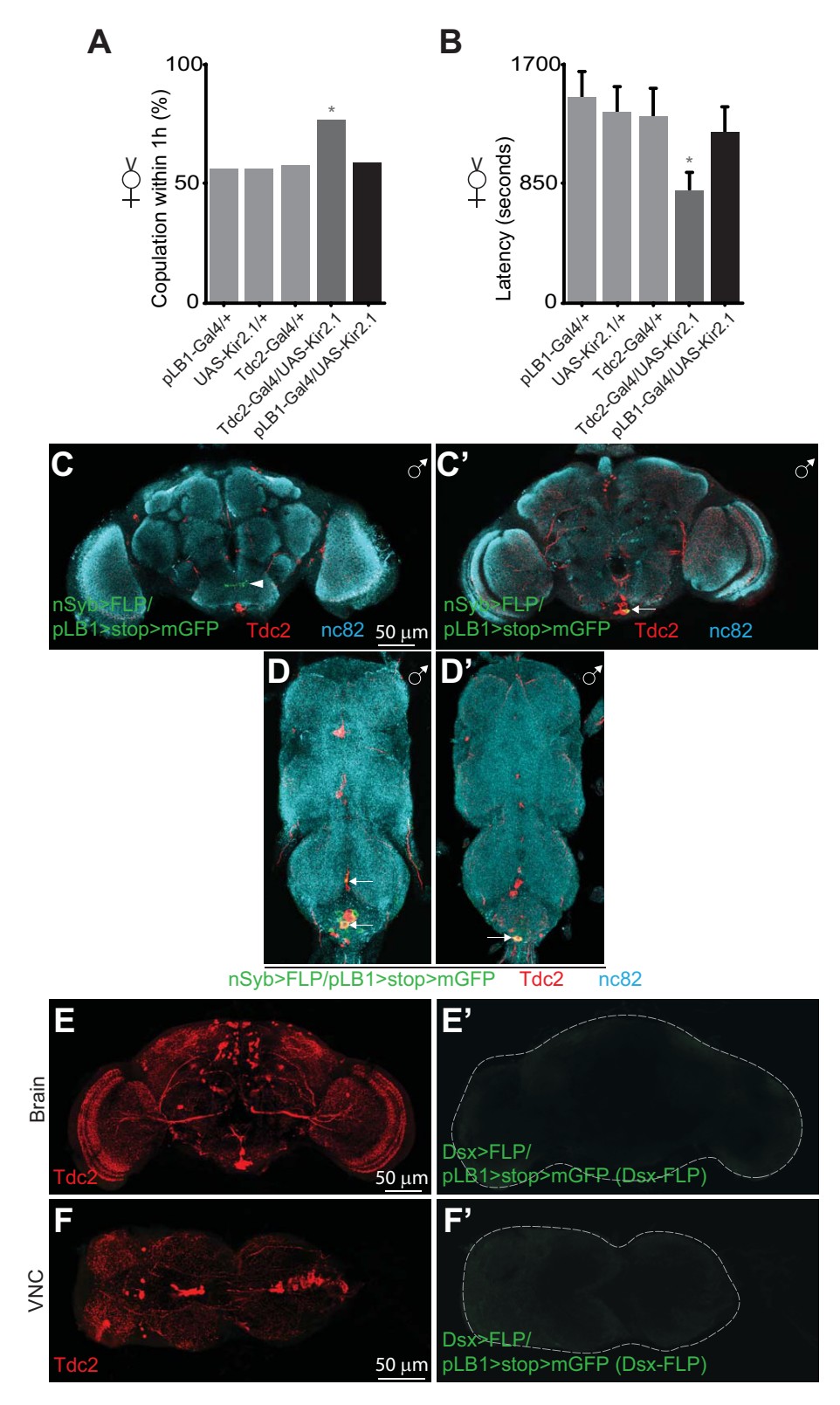

**Figure 3.** pLB1+ neurons are different from Tdc2+/Dsx+ neurons controlling receptivity. Impairing the activity of pLB1+ neurons via UAS-Kir2.1 neither reduces virgin copulation percentage (**A**) nor mating latency (**B**). (**C-D'**); In adult males, immuno-detection in the CNS of neurons expressing pLB1 and producing the Tdc2 enzyme. (**C** and **C'**) are the brain anterior and posterior views, respectively. (**D** and **D'**) are the ventral nerve cord (VNC) ventral and dorsal views, respectively. In adult females, immuno-detection in the brain (**E-E'**) and VNC (**F-F'**) of Dsx+/pLB1+ cells and producing the Tdc2 enzyme;
*Figure 3 continued on next page*

*Figure 3 continued*

no signal for Dsx+/pLB1+ cells is detectable. For (**A**), shown is the copulation percentage for virgins within 1 hour (1h) from six independent trials with a total of 70–80 females per genotype and condition used. All the tested flies were pooled for the calculation and error bars are not appropriate for this kind of representation. For (**B**), shown is the average latency time before mating for virgins from four independent trials with a total of 24 to 40 females per genotype and condition used. For (**C-D'**), staining against nc82 was used to delineate the shape of the brain and VNC. Arrows indicate the position of pLB1+ cell bodies. Arrowheads indicate projections. * indicates p<0.05; Fisher exact t-test (**A**) and non-parametric t-test, Mann-Whitney test (**B**). Details including n values and genotypes can be found in the detailed lines, conditions and, statistics for the figure section.

DOI: https://doi.org/10.7554/eLife.50559.013

The following figure supplements are available for figure 3:

**Figure supplement 1.** pLB1+ neurons may not control remating behavior.
DOI: https://doi.org/10.7554/eLife.50559.014
**Figure supplement 2.** pLB1+/Tdc2+ neurons are present in male brains.
DOI: https://doi.org/10.7554/eLife.50559.015
**Figure supplement 3.** pLB1+/Tdc2+ neurons are present in male VNC.
DOI: https://doi.org/10.7554/eLife.50559.016
**Figure supplement 4.** pLB1+ neurons are not Dsx+.
DOI: https://doi.org/10.7554/eLife.50559.017

inhibitor (Tub-Gal80), it was reestablished when Gal80 expression was specifically suppressed in Tdc2+ cells (*Figure 4A*). This result showed that TTx expression in pLB1+/Tdc2+ cells is sufficient to decrease egg-laying. Conversely, ectopic activation of the thermosensitive TRPA1 protein in pLB1+/Tdc2+ neurons was sufficient to trigger an increase of egg-laying compared to controls (*Figure 4B*). Next, using an RNAi transgene (UAS-Fadd-IR) targeting a cytosolic component of the NF-κB called Fadd (Fas-associated death domain protein), we assayed whether downregulation of NF-κB signaling specifically in pLB1+/Tdc2+ neurons was sufficient to abolish the egg-laying drop observed in females injected with peptidoglycan (*Figure 4C*) (*Leulier et al., 2002*; *Naitza et al., 2002*). While control lines presented an egg-laying drop post peptidoglycan injection, this was no longer the case for females expressing the Fadd-IR transgene in pLB1+/Tdc2+ cells. This result showed that peptidoglycan-mediated activation of the NF-κB pathway in pLB1+/Tdc2+ neurons is the triggering event that reduces egg-laying upon bacterial peptidoglycan exposure.

## Neurons that adapt female egg-laying behavior to infectious status are located in the brain

Next, to delineate which of the pLB1+ neurons located in the brain or the VNC were responsible for the egg-laying modulation in response to bacterial peptidoglycan, we used different strategies. First, we took advantage of the OTD-FLP transgene shown to be expressed in brain but not in VNC neurons (*Asahina et al., 2014*). We first tested whether this transgene was indeed expressed in brain pLB1+ cells. The presence of GFP+/Tdc2+ cells in brains of pLB1-Gal4/UAS>stop>GFP, OTD-FLP adult flies (OTD>FLP/pLB1>stop>mGFP), demonstrated that OTD is expressed in pLB1+/Tdc2+ neurons (*Figure 5—figure supplement 1*). Importantly, animals did not show any staining in the VNC, confirming the specificity of the OTD-FLP driver for the brain neurons (*Figure 5—figure supplement 1*). Once validated, this tool was used to test the effect of inactivating only brain pLB1+ neurons via the potassium channel Kir2.1. While control females laid an average of 60 eggs per day, ubiquitous inactivation of pLB1+ neurons via HS-FLP or targeted inactivation of brain pLB1+ neurons via the OTD-FLP, both resulted in a strong decrease of egg-laying rate with an average of 42 eggs per day (*Figure 5A*). These result was confirmed via a complementary approach using the Tsh-Gal80 abdominal driver inhibitor that blocks Gal4 activation in the thorax, including the VNC while sparing the brain Gal4 expressing cells. The efficiency of the Tsh-Gal80 transgene expression over pLB1-Gal4 was controlled and confirmed via microscopy (*Figure 5—figure supplement 1*). The egg-laying reduction seen upon Gal4-mediated Kir2.1 expression in pLB1+ cells was unaffected by Tsh-mediated expression of Gal80 (*Figure 5B*). Similar conclusions were drawn when UAS-Fadd-IR was used to modulate NF-κB pathway activity *in vivo*. Indeed, the suppression of egg-laying drop 6 hours post peptidoglycan injection mediated by the knockdown of the NF-κB cascade in Tdc2+ and pLB1+ cells was not observed when only thoracic neurons were targeted (*Figure 5C*) and persisted when Tsh-Gal80 was concomitantly expressed (*Figure 5D*). Taken together, these data supported a model

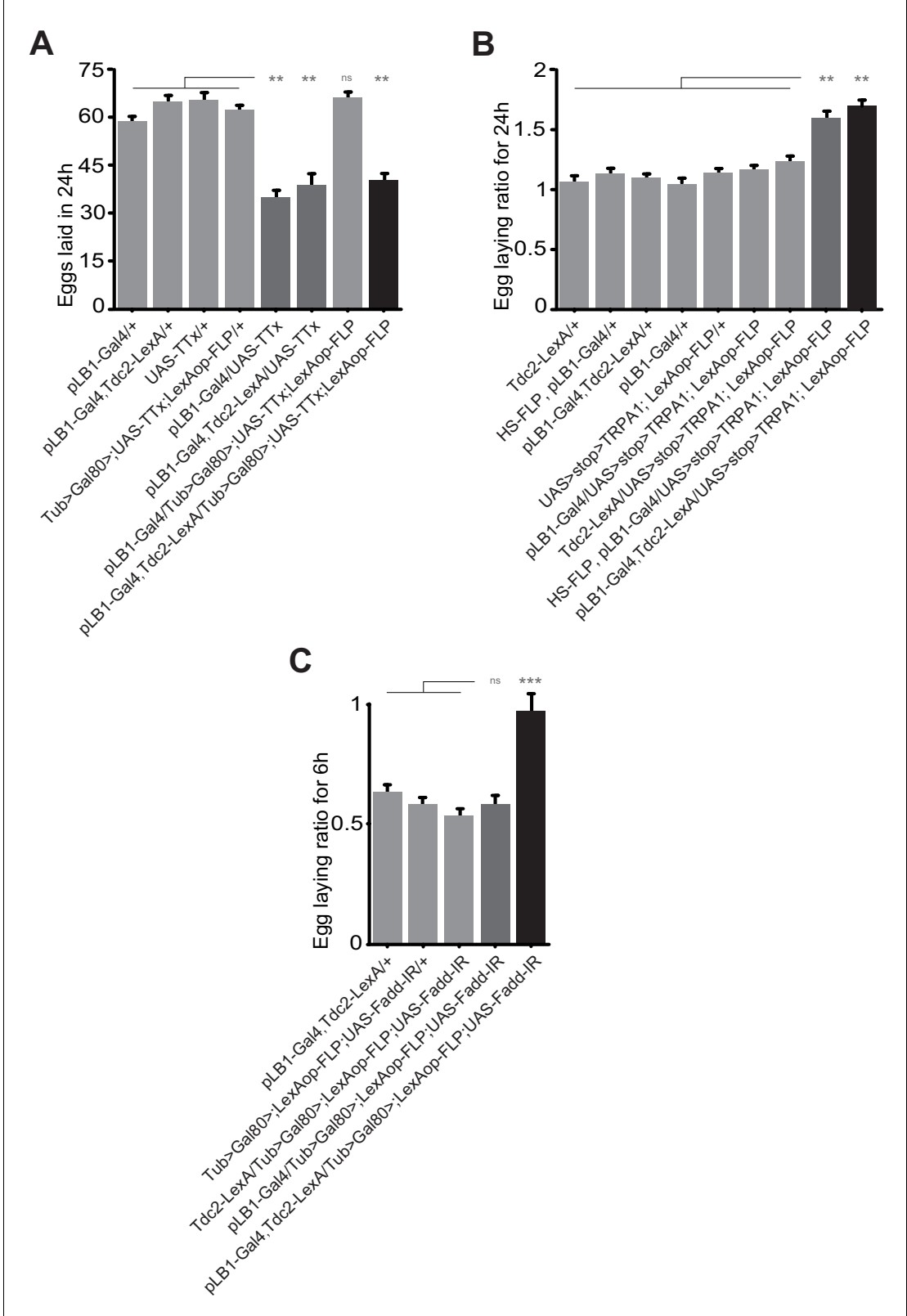

**Figure 4.** Egg-laying drop post peptidoglycan exposure is mediated by pLB1+/Tdc2+ neurons via the NF-κB pathway. (A); Impairing the activity of octopaminergic pLB1+ neurons reduces egg-laying. The ubiquitously expressed Tub-Gal80 that inhibits the activity of Gal4 can be flipped-out in cells expressing the LexA. Thus, only in cells co-expressing the Gal4 and the LexA the UAS-TTx will be expressed. (B); Increasing the activity of pLB1+ neurons augments egg-laying. The ubiquitously expressed Tub-Gal80 can be flipped-out only in cells expressing the heat shock (HS) flippase or the

*Figure 4 continued on next page*

*Figure 4 continued*

LexA. Thus, only in cells co-expressing the Gal4 and the LexA the UAS-TRPA1 will be expressed. (C); Octopaminergic pLB1+ neurons control the egg-laying drop post-peptidoglycan injection via Fadd. Only cells co-expressing the Gal4 and LexA express the Fadd RNAi (UAS-Fadd-IR) transgene. For (A), shown are the average numbers of eggs laid per fly per 24 hours ± SEM from at least two independent trials with at least 20 females per genotype and condition used. For (B), shown are the average egg-laying ratios per fly per 24 hours ± SEM from three independent trials with at least 35 females per genotype and condition used. For (C), shown are the average egg-laying ratios per fly per 6 hours ± SEM from at least two independent trials with at least 20 females per genotype and condition used. * indicates p<0.05; ** indicates p<0.001; *** indicates p<0.0001; n.s. indicates p>0.05, non-parametric ANOVA, Dunn's multiple comparison test. Details including n values and genotypes can be found in the detailed lines, conditions and, statistics for the figure section.
DOI: https://doi.org/10.7554/eLife.50559.018

The following source data is available for figure 4:

**Source data 1.** Egg laying raw data for *Figure 4A*.
DOI: https://doi.org/10.7554/eLife.50559.019
**Source data 2.** Egg laying raw data for *Figure 4B*.
DOI: https://doi.org/10.7554/eLife.50559.020
**Source data 3.** Egg laying raw data for *Figure 4C*.
DOI: https://doi.org/10.7554/eLife.50559.021

in which the brain, and not the VNC pLB1+/Tdc2+ neurons, modulate egg-laying upon peptidoglycan-dependent NF-κB pathway triggering.

## Some neurons that adapt female egg-laying behavior to infectious status are connected to VNC and express endogenous PGRP-LB protein

One important issue relates to the neuronal projections of the brain pLB1+/Tdc2+ cells. Unfortunately, the weak expression of the pLB1-Gal4 driver, prevented us to identify them. To overcome this problem, we generated a pLB1-LexA driver using the same DNA fragment as for the pLB1-Gal4 line. This tool allowed us to perform intersectional strategies to unlock, via the flippase/FRT system, the strong Tdc2-Gal4 driver in pLB1+ cells. In brains of pLB1-LexA/Tdc2-Gal4, UAS>stop>mCD8-GFP; LexAop-FLP (pLB1>FLP/Tdc2>stop>mGFP) flies, we detected five neuronal cell bodies (*Figure 6A*) and their neuronal projections (*Figure 6B–D*). In addition to the neurons already detected with the pLB1-Gal4 driver in the VM cluster, two AL2 octopaminergic neurons were detected (*Figure 6A*, left panel). The position of the identified pLB1+/Tdc2+ neuron within the VM I cluster, its symmetrical (unpaired) nature as well as the shape of its ascending projections make it likely to belong to the OA-VUMa4 class of neurons (*Busch et al., 2009*) (*Figure 6B*). The projection pattern of the two AL2 asymmetrical neurons in the ocular lobes identified them as OA-AL2i1 (*Figure 6C*) (*Busch et al., 2009*). The highly intricate and overlapping pattern of the pLB1+/Tdc2+ VM II neuron projections impaired its precise identification (data not shown). However, this neuron of the VM cluster has unpaired ascending projections, and thus belongs to the VUM class. With regard to pLB1+/Tdc2+ VM III neuron, it sends symmetrical projections descending towards the VNC which look very much like the ones of the described VUMd2 class of neurons, a class exclusively located in the VM III sub-cluster (*Figure 6D*) (*Busch et al., 2009*). Combined with our previous results demonstrating that the brain pLB1+/Tdc2+ neurons can modulate egg-lay following peptidoglycan exposure, we propose that peptidoglycan may interfere with the activity of the VUMd2 neuron that sends projections in the VNC (*Busch et al., 2009*).

Since all functional data relied on the pLB1-Gal4 construct and although previous rescue experiments suggested that this driver at least partially mimics PGRP-LB endogenous pattern (*Kurz et al., 2017*), we generated a PGRP-LB::GFP line in which all endogenous PGRP-LB isoforms were tagged with GFP at the endogenous locus. Brains of PGRP-LB::GFP flies showed intracellular localization of PGRP-LB protein (probably the cytosolic isoforms) in neurons of the octopaminergic VM and AL2 clusters (*Figure 7*). Of interest, few PGRP-LB::GFP+, but Tdc2- cells, were also detected (*Figure 7*, arrow). This result unambiguously demonstrated that the endogenous immune regulator PGRP-LB is produced by a subclass of octopaminergic neurons of the AL2, VM II, and VM III sub-clusters, among which are the pLB1 neurons.

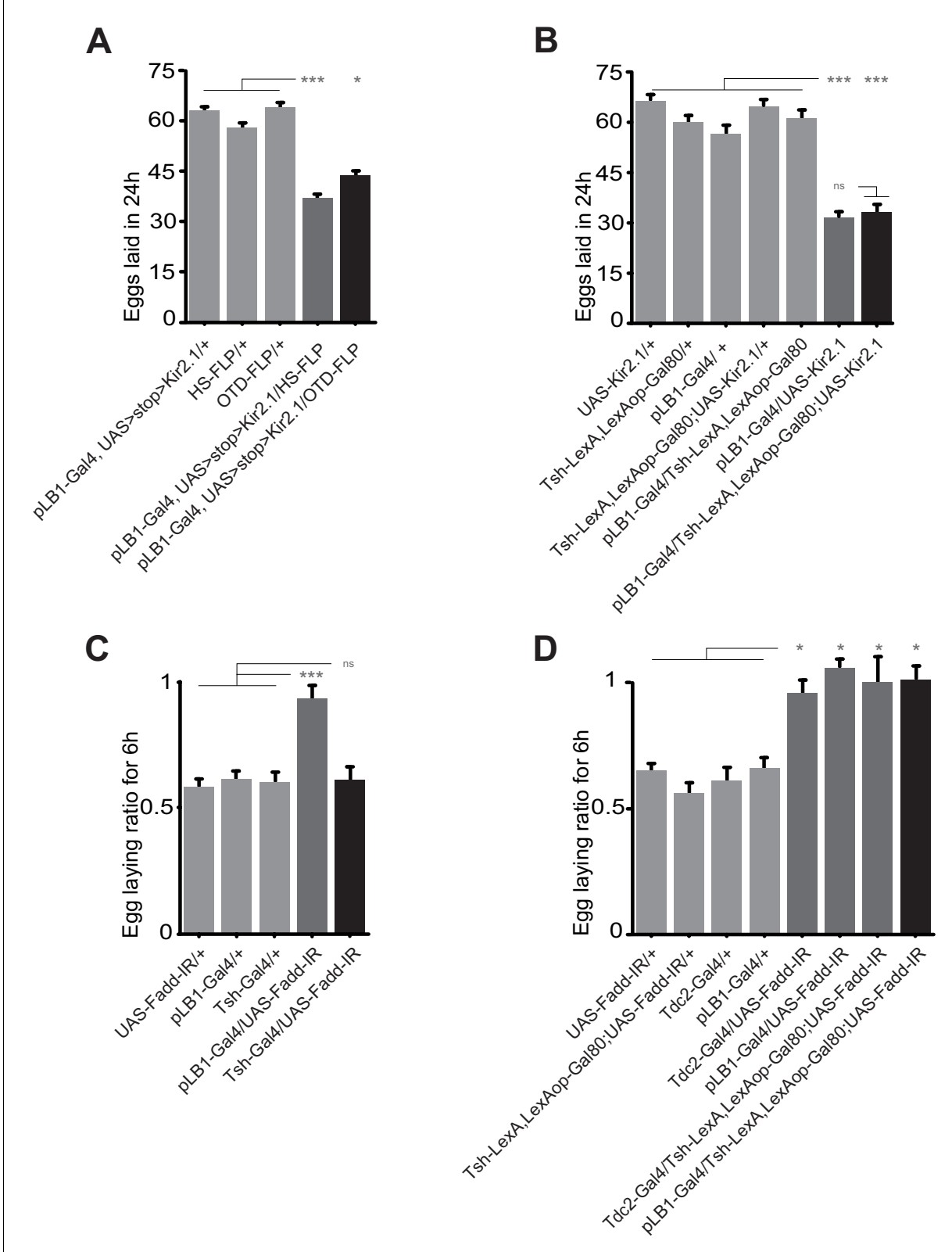

**Figure 5.** Egg-laying drop post peptidoglycan exposure is mediated by the brain, but not the VNC pLB1+ neurons. (**A**); Impairing the activity of pLB1+ cells of the brain reduces egg-laying. Only in cells co-expressing the FLP and the Gal4 will the UAS>stop>Kir2.1 be effective. Heat shock (HS) is ubiquitous while OTD is brain-restricted. (**B**); pLB1+ cells of the VNC are dispensable for the modulation of the egg-lay. The expression of LexAop-Gal80 antagonizes the activity of Gal4, thus preventing the effects of UAS-Kir2.1. Tsh-LexA drives the expression of Gal80 in the fly thorax, including the

*Figure 5 continued on next page*

*Figure 5 continued*

VNC. (**C**); RNAi-mediated Fadd (Fadd-IR) inactivation in the Tsh-Gal4+ cells does not prevent egg-lay drop post peptidoglycan injection. (**D**); RNAi-mediated Fadd inactivation in pLB1+ cells of the brain, but not of the VNC prevents egg-lay drop post peptidoglycan injection. The expression of LexAop-Gal80 antagonizes the activity of Gal4, thus preventing the effects of UAS-Fadd-IR, only in cells co-expressing the Gal4 and the LexA. For (**A and B**), shown are the average numbers of eggs laid per fly per 24 hours ± SEM from at least two independent trials with at least 20 females per genotype and condition used. For (**C and D**), shown are the average egg-laying ratios per fly per 6 hours ± SEM from at least two independent trials with at least 20 females per genotype and condition used. * indicates p<0.05; ** indicates p<0.001; *** indicates p<0.0001; n.s. indicates p>0.05, non-parametric ANOVA, Dunn's multiple comparison test. Details including n values and genotypes can be found in the detailed lines, conditions and, statistics for the figure section.

DOI: https://doi.org/10.7554/eLife.50559.022

The following source data and figure supplement are available for figure 5:

**Source data 1.** Egg laying raw data for *Figure 5A*.
DOI: https://doi.org/10.7554/eLife.50559.024
**Source data 2.** Egg laying raw data for *Figure 5B*.
DOI: https://doi.org/10.7554/eLife.50559.025
**Source data 3.** Egg laying raw data for *Figure 5C*.
DOI: https://doi.org/10.7554/eLife.50559.026
**Source data 4.** Egg laying raw data for *Figure 5D*.
DOI: https://doi.org/10.7554/eLife.50559.027
**Figure supplement 1.** OTD-FLP is expressed in pLB1+ neurons of the head, but not of the thorax whileTsh-LexA, LexAop-Gal80 efficiently silences pLB1-Gal4 in the thorax.
DOI: https://doi.org/10.7554/eLife.50559.023

## Neurons that adapt female egg-laying behavior to infectious status are inhibited by peptidoglycan

To further test the effects of peptidoglycan on brain neurons, we performed calcium imaging using both *in vivo* and *ex vivo* methods. For that purpose, peptidoglycan solution was applied directly onto the brains of alive Tdc2-Gal4/UAS-GCaMP6s flies and GFP fluorescence intensity was monitored over time. We focused on octopaminergic neurons of the VM II/VM III sub-clusters (*Figure 8—figure supplement 1*) which, as shown above, contain the cells that express the pLB1 driver the most consistently and directly contact the VNC. In contrast to the control solution (Ringer's solution) (*Video 1*), brain stimulation by bacterial peptidoglycan induced a rapid and transient decrease of the GFP signal in VM II/III octopaminergic sub-clusters (*Figure 8A–C* and *Video 2*). To more precisely assay the response of pLB1+/Tdc2+ neurons upon peptidoglycan exposure, we performed *ex vivo* calcium imaging on dissected brains in which GCaMP6s was expressed in pLB1+/Tdc2+ cells only (pLB1>FLP/Tub>Gal80>,Tdc2>GCaMP6s, *Figure 8D–F*). For the *ex vivo* experiments with dissected brains, we focused on the posterior part of the brain in which only the pLB1+/Tdc2+ neuron of the VM III cluster is detectable. Fluorescence intensity quantifications showed that direct stimulation of dissected brains by peptidoglycan triggered a reduction of calcium levels in this VM III pLB1+/Tdc2+ neuron (*Video 4*) compared to control (*Video 3*). These results which indicated that peptidoglycan exposure inhibits pLB1+/Tdc2+ neuronal activity are coherent with functional data showing that blockage of pLB1+/Tdc2+ neurons by TTx or Kir2.1 overexpression reduces female egg-laying, a genetically triggered egg-lay drop that mimics the physiological response following peptidoglycan exposure. The differences observed between GCaMP kinetics in *ex vivo* (persistent drop post peptidoglycan stimulation) and *in vivo* (transient) experiments could reflect the fact that, for *in vivo* experiments, the imaged flies were alive with brains still connected to the rest of the nervous system including the VNC. In contrast, dissected brains disconnected from the periphery and the VNC were used for *ex vivo* experiments. Besides, whereas peptidoglycan was added to exposed brains still bathing in the surrounding hemolymph for the *in vivo* settings, it was added to brains bathing in Ringer's solution for *ex vivo* experiments.

## pLB1+ neurons inhibition and peptidoglycan exposure temporary block mature oocytes release

We previously noticed that the egg-laying drop induced by bacterial or peptidoglycan injection was not associated with premature death of early-stage oocytes as described for wasp-exposed flies or

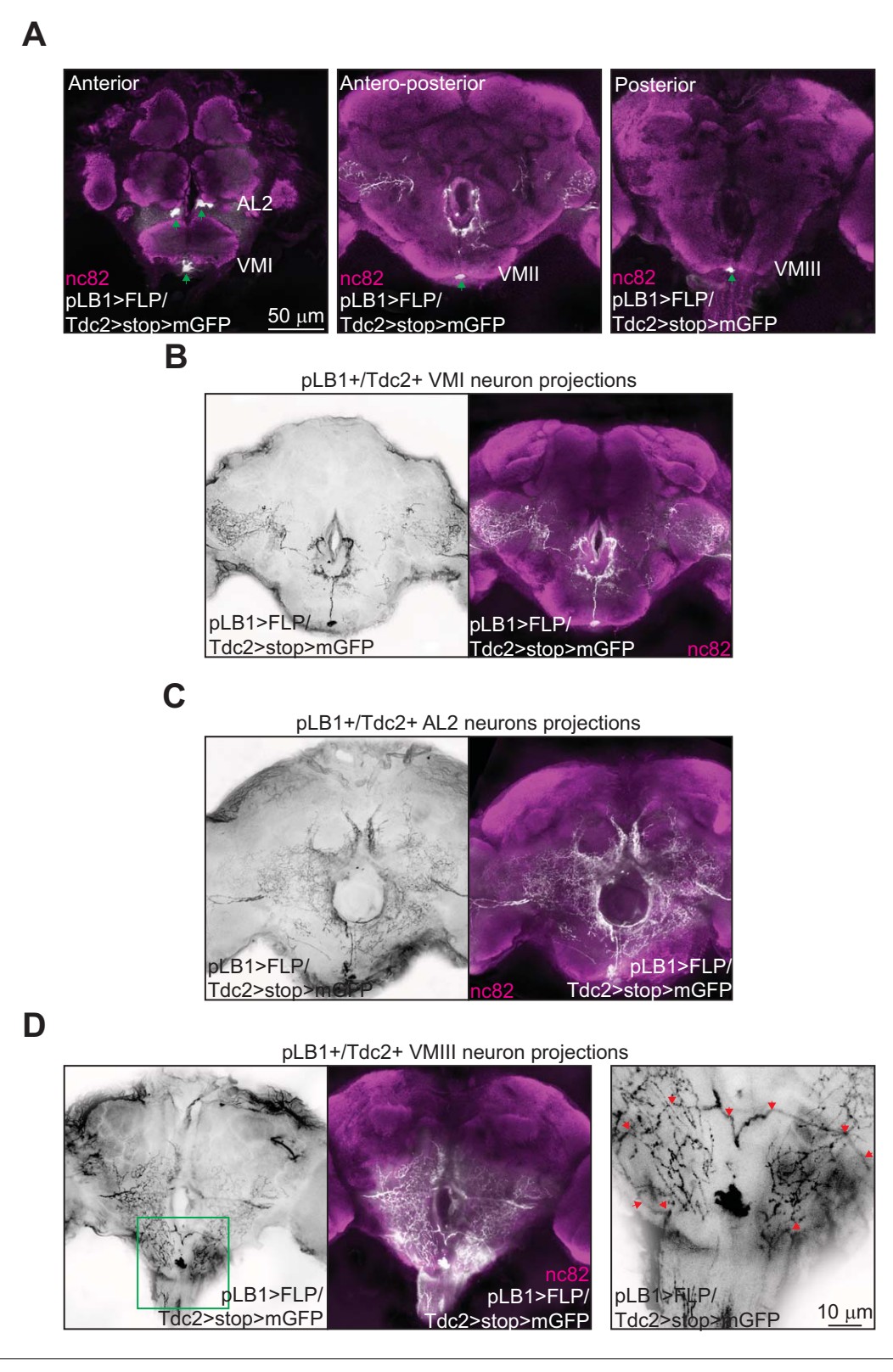

**Figure 6.** Brain pLB1+/Tdc2+ neurons project to the esophagus as well as to the VNC. (A–D); Immuno-detection in the brain of cells co-expressing the Tdc2-Gal4 and the pLB1-LexA drivers (pLB1>FLP/Tdc2>stop>mGFP). The GFP can only be expressed under the control of Tdc2 if the stop sequence inserted upstream of the *gfp* gene is flipped-out in pLB1+/Tdc2+ cells. (A); Five cellular bodies (green arrows) are detected in anterior, antero-posterior and posterior parts of the brain. (B–D); Specific stacks of the brain showing the projection patterns (in black) of the neurons present in VM I (B), AL2 (C)

*Figure 6 continued on next page*

*Figure 6 continued*
and VM III (D). In (D), the area delimited by the green box in the left panel is magnified on the right panel to show the descending projections (red arrows). Staining against nc82 was used to delineate the shape of the brain. Details including genotypes can be found in the detailed lines, conditions and, statistics for the figure section.
DOI: https://doi.org/10.7554/eLife.50559.028

insulin-related defects (*Pritchett and McCall, 2012*), but rather with a rapid accumulation of mature oocytes (stage 14) in the ovaries (*Kacsoh et al., 2015b*; *Kurz et al., 2017*). By quantifying the effects of peptidoglycan exposure over time, we showed that this phenotype was reversible (*Figure 9A,B* and *Figure 9—figure supplement 1*). Whereas 6 hours after peptidoglycan injection, the amounts of stage 14 and 11–13 oocytes increased and decreased respectively, the effects were less pronounced after 24 hours and no longer detectable 48 hours post-injection. We then tested whether similar phenotypes could be obtained by a Kir2.1-mediated temporary inactivation of pLB1+ neurons using the Gal4/Gal80$^{ts}$ system (*Figure 9C–E*). As for peptidoglycan injection, pLB1+ neurons transient inactivation led to stage 14 oocyte accumulation which was fully and progressively rescued by restoring normal pLB1+ neuronal function (*Figure 9C–E*, *Figure 9—figure supplement 2*).

## Peptidoglycan exposure modulates egg-laying by inhibiting follicular cell trimming

An important step occurring before oviposition is the transfer of the mature oocyte from the ovary to the lateral oviduct. This step which requires the rupture of the follicular cell layer and consequently allows the release of the oocyte in the oviduct is called follicle trimming. This process is OA-dependent and allows the mature oocyte to reach the lateral oviduct prior to its fertilization (*Lee et al., 2009*). Therefore, we tested whether the egg-laying drop observed upon inactivation of pLB1+ cells and peptidoglycan exposure were associated with a defect in follicle trimming. Ovaries of control lines (pLB1-Gal4/+ and +/UAS-Kir2.1) and animals with inactivated pLB1+ neurons (pLB1-Gal4/UAS-Kir2.1) were dissected when the females were five day-old, while water-injected and peptidoglycan-injected animals were dissected 6 hours after treatment. Following a DAPI staining, stage 14 oocytes were counted and the ratio of stage 14 oocytes covered by follicular cells quantified. We confirmed that compared to control lines, the inactivation of pLB1+ cells via Kir2.1 overexpression as well as the injection of peptidoglycan led to an accumulation of stage 14 oocytes (*Figure 10A–D, E and G*). Also, we observed a decrease of follicle trimming on mature oocytes after inactivation of pLB1+ cells via Kir2.1 overexpression or peptidoglycan injection compared to controls (*Figure 10F and H*). These results suggested that peptidoglycan exposure leads to a decrease of the OA-dependent rupture of follicular cells around mature oocytes and subsequently reduces the number of eggs laid by infected females.

## Discussion

The present study demonstrated that some brain and VNC neurons are expressing endogenous peptidoglycan degrading enzyme PGRP-LB. Functional genetic intersection and calcium imaging data suggest a model in which, among them, very few brain pLB1+/Tdc2+ neurons belonging to the octopaminergic sub-cluster VM III sense peptidoglycan. The latter is likely to inhibit these (this) neurons leading to an NF-κB-dependent decrease of egg-laying in bacterially infected females (*Kurz et al., 2017*) (*Figure 11*). These results raise the question of the molecular and cellular mechanisms by which bacteria-derived peptidoglycan present in the hemolymph is able to reach this/these brain neuron/s. One model would be that circulating peptidoglycan can cross the blood-brain-barrier to reach brain neurons. In such a case, the selective expression of peptidoglycan sensor and NF-κB signaling components in some neurons, in this case, the pLB1+ neurons will confer them the ability to respond to peptidoglycan. In line with the idea that peptidoglycan can reach the brain, works in mice have shown that bacterial peptidoglycan derived from the gut microbiota can translocate into the brain where it is sensed by specific pattern-recognition receptors of the innate immune system (*Arentsen et al., 2017*; *Arentsen et al., 2015*). However, precisely mapping the peptidoglycan localization in the brain and identifying the exact path followed by this gut-born bacteria metabolite to reach the brain will require peptidoglycan tracing methods which are not yet available.

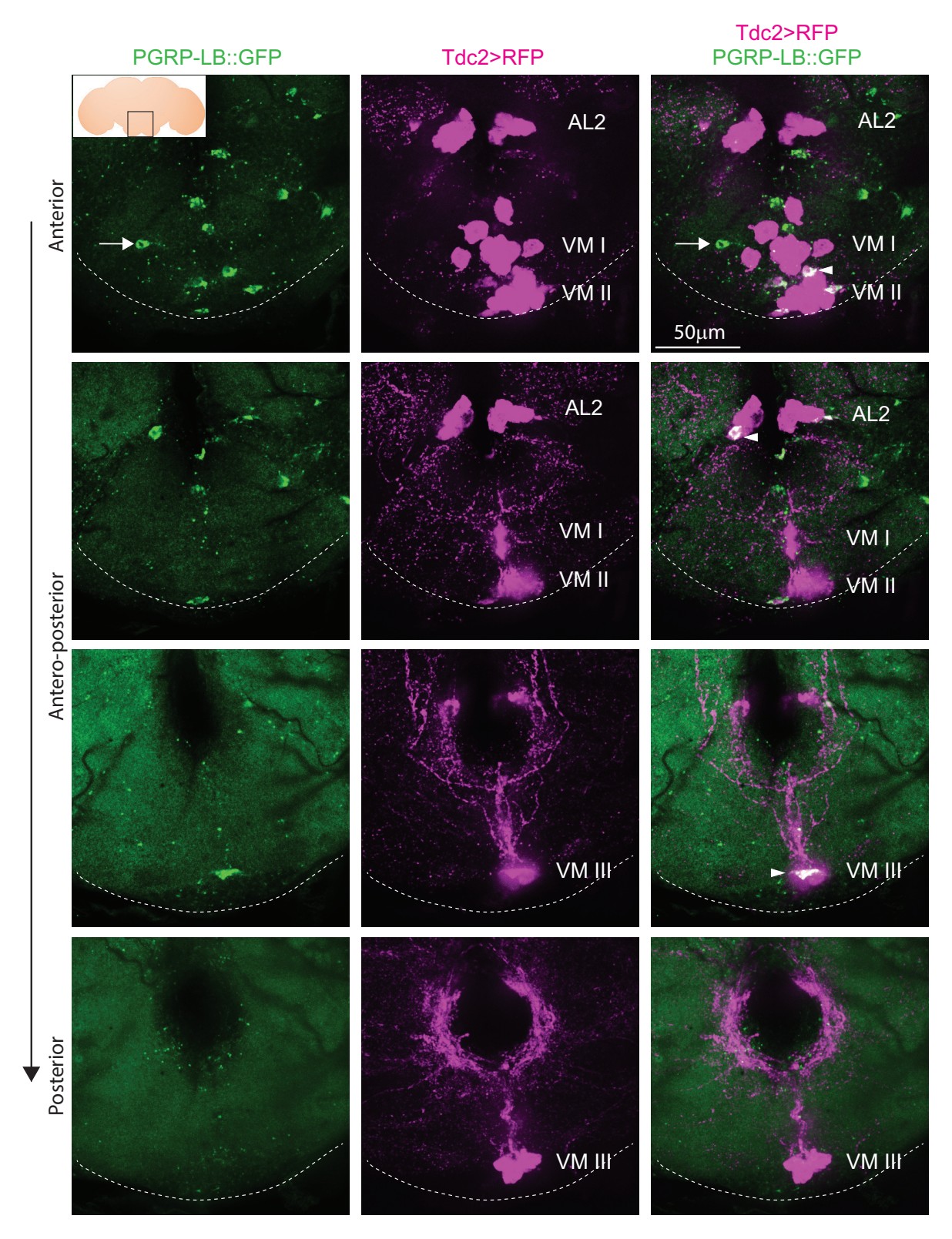

**Figure 7.** Endogenous PGRP-LB::GFP is expressed in Tdc2+ cells of the AL2 and VM clusters. Detection of PGRP-LB::GFP fusion protein as well as Tdc2 cells expressing RFP (Tdc2>RFP) without immunostaining . Only the area of the brain containing the octopaminergic AL2, VM I, VM II, and VM III clusters is shown with stacks corresponding to anterior, antero-medial, postero-medial and posterior views. The inserted scheme represents the brain, the empty black square delineates the area observed and the dashed line represents the ventral limit of the brain. Arrowheads point to Tdc2+/PGRP-

*Figure 7 continued on next page*

*Figure 7 continued*

LB::GFP+ cells and the arrow points to a Tdc2 negative cell containing PGRP-LB::GFP proteins. Details including genotypes can be found in the detailed lines, conditions and, statistics for the figure section.

DOI: https://doi.org/10.7554/eLife.50559.029

Alternatively, peptidoglycan could also be sensed by the pLB1+/Tdc2+ neurons at the level of the peripheric axonal or dendritic projections using a retrograde transport to bring peptidoglycan close to the cell body and hence allowing NF-κB activation. Further works and additional tools will be required to address these important questions.

It would be also important to understand how pLB1+/Tdc2 VM III neurons regulate female egg-laying behavior in response to infection. Our results using the pLB1-LexA driver and a previously published anatomical map of brain octopaminergic neurons (*Busch et al., 2009*) indicate that some pLB1+/VM III neurons are sending descending neurites to the thoracic ganglia via cervical connectives. Analyzing the precise connectivity of these neurons will likely shed some light on these mechanisms.

A recent report showed that gut-resident *Lactobacillus brevis* can modify adult locomotion by acting on octopaminergic neurons via sugar metabolism (*Schretter et al., 2018*). This demonstrates that gut-associated bacteria have multiple ways to interact with the host behaviors. Alternatively, eukaryotes have developed different sensing mechanisms to adapt their behaviors to autochthonous or allochthonous bacteria. However, while the effects mediated by peptidoglycan on host egg-laying behavior are likely to be widespread since peptidoglycan is an universal bacterial cell-wall constituent, only bacteria producing xylose isomerase, such as *Lactobacillus brevis*, are expected to modify the walking activity of the colonized hosts. It should be mentioned that octopamine has also been shown to mediate neural regulation of immunity in *C. elegans* (*Sellegounder et al., 2018*).

Our data indicate that most pLB1+ neurons are not involved in controlling the egg-laying rate in response to infection and that a small fraction of them is octopaminergic. This suggests that most octopaminergic-dependent behaviors are unlikely to be affected by peptidoglycan exposure. Consistently, our data demonstrate that the receptivity of females to males, a behavior that is also under the control of octopamine, is not mediated by pLB1+ neurons. We, therefore, propose that by expressing sensors and effectors of the immune pathway in a small subset of neurons, flies render some of their behaviors controllable by bacteria-derived metabolite while maintaining others bacteria-unsensitive. Interestingly, pLB1+/Tdc2+ neurons that adapt female oviposition rate to their infectious status, are also present in males where they might regulate male-specific OA-dependent behaviors upon infection. One of the reported functions of octopaminergic circuitry is to modulate specific behaviors to environmental conditions (*Crocker and Sehgal, 2008*; *Rezával et al., 2014*; *Youn et al., 2018*). Such a modulatory function seems adapted to integrate immune signals allowing the fly to adapt to an environment enriched in microorganisms. Furthermore, the fact that some pLB1+ neurons are not octopaminergic, which is confirmed by using a PGRP-LB::GFP line in which all endogenous PGRP-LB protein isoforms are tagged, suggested that other neuronally controlled biological processes, yet to be identified, are likely to be influenced by PGN exposure. Identifying the exact nature of the non-octopaminergic pLB1+ neurons will be necessary to reveal the processes that they regulate. In addition, since pLB1+ neurons may represent only a subset of all the neurons that express immune pathway components, one could consider that other behaviors, yet to be identified, are controlled by bacteria.

The data from this study demonstrate that the response of few brain octopaminergic neurons to peptidoglycan is NF-κB pathway dependent, hence probably transcriptional. What are the NF-κB target genes mediating the effects of peptidoglycan in these neurons? Obvious candidates are the enzymes necessary for the production of octopamine itself which are Tdc2 and TβH. Consistently, providing ectopic TβH in pLB1+ neurons was shown to rescue egg-laying drop post infection (*Kurz et al., 2017*). Alternatively, antimicrobial peptides (AMPs) which are the main targets of NF-κB/Relish downstream of the innate immune pathways should also be considered. Although historically identified for their antimicrobial activity (*Ezekowitz and Hoffmann, 1996*), some recent reports indicate that AMPs play some important roles in the fly nervous system. Some AMPs, such as Metchnikowin, Drosocin, and Attacin, are implicated in sleep regulation (*Dissel et al., 2015*). Diptericin, a

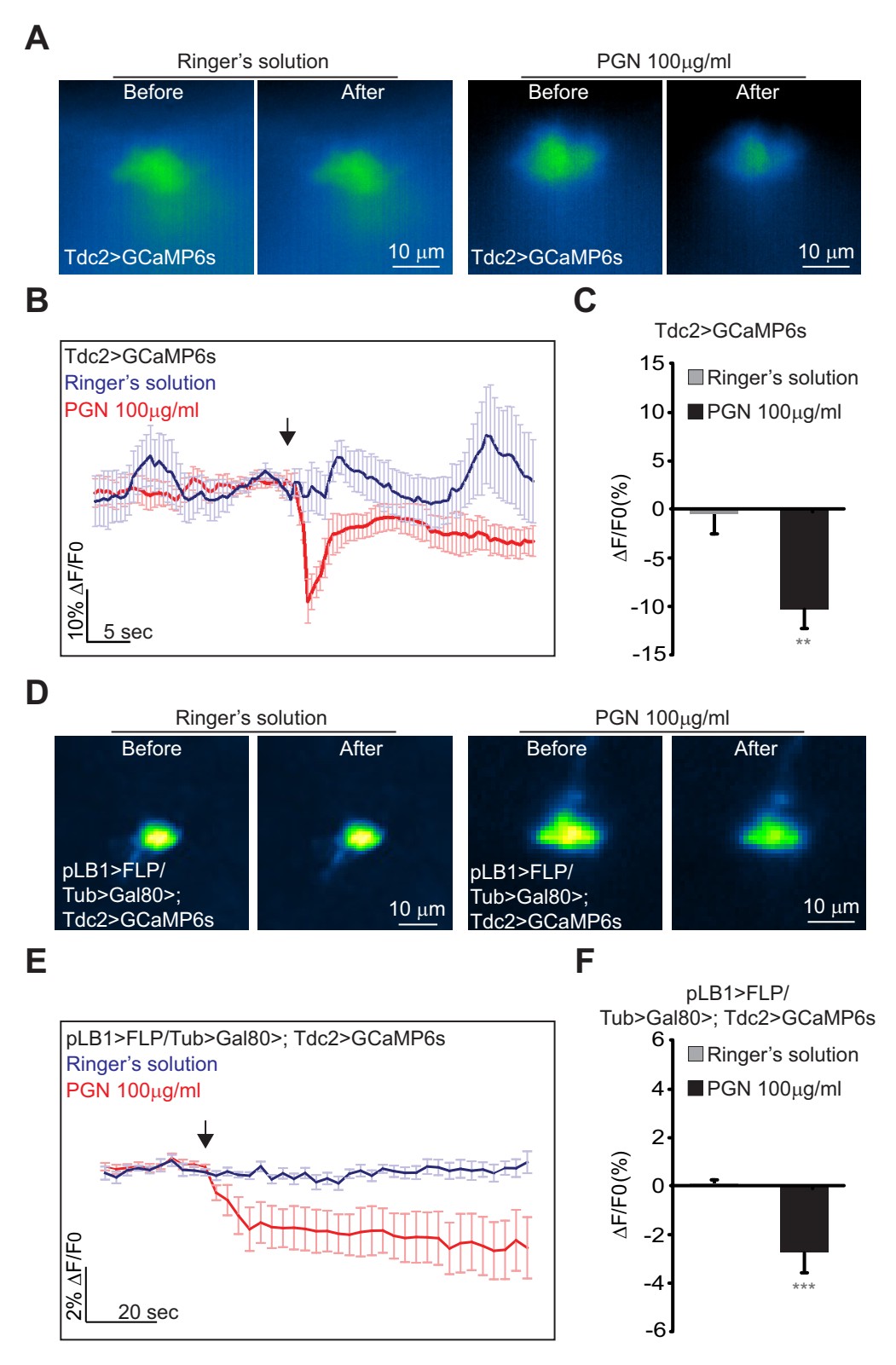

**Figure 8.** Real-time calcium imaging of Tdc2 VM II/III and pLB1+/Tdc2+ VM III neurons exposed to peptidoglycan. (A–F); Real-time calcium imaging using the calcium indicator GCaMP6s to reflect the *in vivo* neuronal activity of Tdc2 neurons (Tdc2>GCaMP6s) in VM II/VM III sub-clusters (A–C) or the *ex vivo* neuronal activity of Tdc2+/pLB1+ VM III neuron pLB1>FLP/Tub>Gal80>,Tdc2>GCaMP6s (D–F). (A); Representative images showing the GCaMP6s intensity before and after addition of either the control Ringer's solution (left panels) or the peptidoglycan (right panels). The images

*Figure 8 continued on next page*

*Figure 8 continued*

represent the average intensity of 4 frames before or after Ringer or peptidoglycan solution. (**B**); Averaged ± SEM time course of the GCaMP6s intensity variations (ΔF/F0 %) for Tdc2+ neurons of the VM II/VM III sub-clusters. The addition of Ringer's solution (n=8 flies) or peptidoglycan (n=13 flies) at a specific time is indicated by the arrow. (**C**); Averaged fluorescence intensity of negative peaks ± SEM for control (n=8) and peptidoglycan-treated flies (n=13). (**D**); Representative images showing the GCaMP6s intensity before and after the addition of either the control Ringer's solution (left panels) or the peptidoglycan (right panels). The images represent the average intensity of 4 frames before or after Ringer or peptidoglycan solution. (**E**); Averaged ± SEM time course of the GCaMP6s intensity variations (ΔF/F0 %) for Tdc2+/pLB1+ neuron of the VM III sub-cluster. The addition of Ringer's solution (n=10 flies) or peptidoglycan (n=12 flies) at a specific time is indicated by the arrow. (**F**); Averaged fluorescence intensity of negative peaks ± SEM for control (n=10) and peptidoglycan-treated flies (n=12) In (**C**), ** indicates p=0.001; in (**F**), *** indicates p=0.0001 non-parametric t-test, Mann-Whitney test. Details including n values and genotypes can be found in the detailed lines, conditions and, statistics for the figure section.

DOI: https://doi.org/10.7554/eLife.50559.030

The following figure supplement is available for figure 8:

**Figure supplement 1.** The *in vivo* and *ex vivo* real-time Calcium imaging approaches focused on neurons present in the VM II/III octopaminergic sub-cluster.

DOI: https://doi.org/10.7554/eLife.50559.031

well-characterized AMP, is important for a kind of non-associative learning, where ethanol preference is modified upon exposure to predatory wasps (*Bozler et al., 2017*). In addition, AMPs expressed in the fly adult head are involved in modulating long-term memory (*Barajas-Azpeleta et al., 2018*). Finally, the innate immune receptor PGRP-LC and downstream signaling are implicated in the regulation of the homeostatic plasticity of neuromuscular junction synapse by NF-κB/Relish-dependent and independent processes (*Harris et al., 2015*; *Harris et al., 2018*). Further experiments will be necessary to test whether any of these cellular mechanisms are also at play in the regulation of neuronal-controlled behaviors by bacteria-derived metabolite in general and by PGN in particular.

Finally, it would be necessary to elucidate how neurons exposed to peptidoglycan modify intracytosolic calcium levels and identify a possible functional link with NF-κB signaling. Our previous and current results demonstrate that the egg-laying drop requires several elements of the IMD pathway, from the PGN-receptor PGRP-LE to the transcription factor NF-κB/Relish. It should be noted that a link between calcium levels and NF-κB activation in neurons has been reported in many mammalian studies (*Lilienbaum and Israël, 2003*; *Lipton, 1997*; *O'Neill and Kaltschmidt, 1997*). In addition, calcineurin, a $Ca^{2+}$-dependent phosphatase was shown to promote NF-κB dependent immune responses in the *Drosophila* larvae (*Dijkers and O'Farrell, 2007*). As for the causality between peptidoglycan stimulation and calcium decrease and despite the kinetic that does not suggest the involvement of a stereotypical signaling cascade, it should first be tested whether this step requires the elements of the IMD (Immune deficiency) pathway. If not, other receptors yet to be identified may mediate this fast response to peptidoglycan. Intriguingly, a recent study using the fly embryo linked a rapid modification of the calcium concentration in fly hemocytes undergoing phagocytosis of apoptotic corpses with the subsequent activity of the JNK pathway, the first event being a pre-requisite for the second (*Razzell et al., 2013*).

# Materials and methods

## Key resources table

| Reagent type (species) or resource | Designation | Source or reference | Identifiers | Additional information |
|---|---|---|---|---|
| Genetic reagent (*D. melanogaster*) | pLB1-Gal4 | (*Kurz et al., 2017*) | | |
| Genetic reagent (*D. melanogaster*) | UAS-Kir2.1 | Bloomington Drosophila Stock Center (*Hardie et al., 2001*) | BDSC Cat# 6595, RRID: BDSC_6595 | |

*Continued on next page*

*Continued*

| Reagent type (species) or resource | Designation | Source or reference | Identifiers | Additional information |
|---|---|---|---|---|
| Genetic reagent (*D. melanogaster*) | UAS-TTx | Bloomington Drosophila Stock Center (*Sweeney et al., 1995*) | BDSC Cat# 28838, RRID: BDSC_28838 | |
| Genetic reagent (*D. melanogaster*) | UAS-TRPA1 | Bloomington Drosophila Stock Center | BDSC Cat# 26264, RRID: BDSC_26264 | |
| Genetic reagent (*D. melanogaster*) | UAS-Fadd-IR | (*Khush et al., 2002*) | | |
| Genetic reagent (*D. melanogaster*) | UAS> stop> GFPmCD8 | Bloomington Drosophila Stock Center | BDSC Cat# 30125, RRID: BDSC_30125 | |
| Genetic reagent (*D. melanogaster*) | nSyb-LexA | Bloomington Drosophila Stock Center | BDSC Cat# 51951, RRID: BDSC_51951 | |
| Genetic reagent (*D. melanogaster*) | Tdc2-LexA | Bloomington Drosophila Stock Center | BDSC Cat# 52242, RRID: BDSC_52242 | |
| Genetic reagent (*D. melanogaster*) | Tub>Gal80> | Bloomington Drosophila Stock Center | BDSC Cat# 38879, RRID: BDSC_38879 | |
| Genetic reagent (*D. melanogaster*) | LexAop-FLP | Bloomington Drosophila Stock Center | BDSC Cat# 55819, RRID: BDSC_55819 | |
| Genetic reagent (*D. melanogaster*) | 8XLexAop2-FLP | Bloomington Drosophila Stock Center | BDSC Cat# 55820, RRID: BDSC_55820 | |
| Genetic reagent (*D. melanogaster*) | UAS>stop > Kir2.1 | | | |
| Genetic reagent (*D. melanogaster*) | UAS>stop> TRPA1 | Bloomington Drosophila Stock Center | BDSC Cat# 66871, RRID: BDSC_66871 | |
| Antibody | Rabbit polyclonal anti-Tdc2 | Abcam | Cat# ab128225, RRID: AB_11142389 | 1:1000 |
| Chemical compound | PGN from *E. coli* | Invivogen | 14C14-MM | |
| Software, algorithm | Fiji | NIH | https://fiji.sc/ | |
| Software, algorithm | GraphPad Prism 6 | GraphPad | RRID: SCR_002798 | |

## *Drosophila melanogaster* strains and maintenance

The following strains were used in this work: pLB1-Gal4 (*Kurz et al., 2017*); UAS-GFPnls (BDSC Cat# 4775, RRID:BDSC_4775); UAS-mCD8-Tomato (kindly provided by F. Schnorrer); UAS-TTx (*Sweeney et al., 1995*), (BDSC Cat# 28838, RRID:BDSC_28838); UAS-TRPA1 (BDSC Cat# 26264, RRID:BDSC_26264); UAS-Kir2.1 (BDSC Cat# 6595, RRID:BDSC_6595) (*Hardie et al., 2001*); UAS-

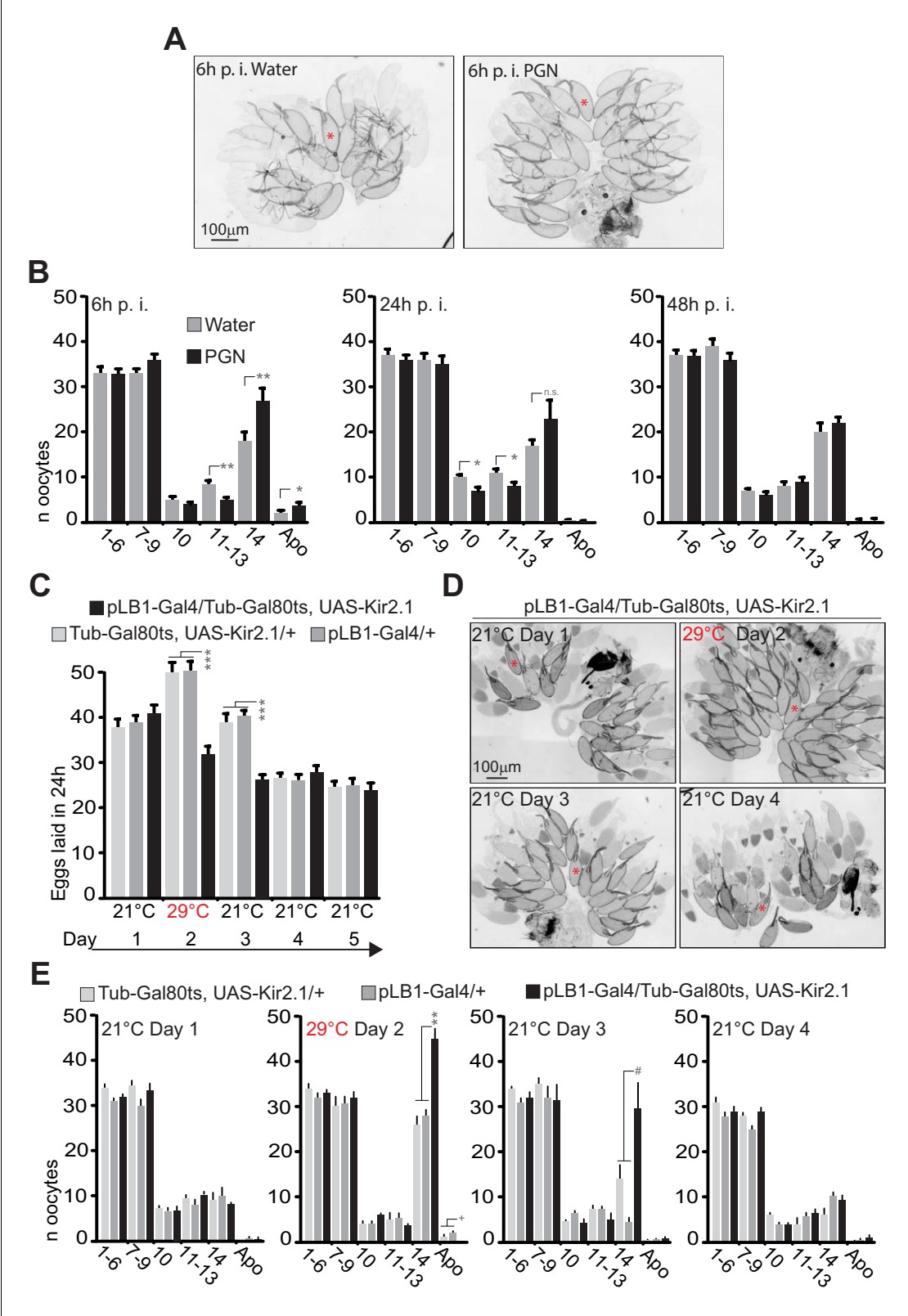

**Figure 9.** Peptidoglycan exposure as well as pLB1+ neurons conditional inactivation lead to a reversible mature oocyte accumulation. (A–B); Injection of peptidoglycan triggers a reversible accumulation of mature oocytes (stage14). (A); 6 hours (6h) post-treatment (p.i.), stage 14 oocytes accumulate in the ovaries of peptidoglycan-injected flies. Transmission light microscopy images of ovaries dissected from control flies (water injected) or peptidoglycan-injected animals 6h post-treatment. (B); peptidoglycan injection modifies the quantity and quality of oocytes by 6 h, leading to an accumulation of

*Figure 9 continued on next page*

Figure 9 continued

mature stage (stage14) eggs. The dynamic over three different time points (6h-24h-48h) post-treatment was assayed. (C–E); pLB1+ neurons reduced activity leads to stage 14 oocyte accumulation. (C); The conditional inactivation of pLB1+ neurons reduces egg-laying. At 21°C, the ubiquitously produced Gal80$^{ts}$ inactivates the Gal4 and thus the Kir 2.1 protein expression. At 29°C, the Gal80$^{ts}$ doesn't inactivate the Gal4, leading to the inhibited activity of pLB1+ neurons via Kir2.1. Switching back the animals to 21°C inhibits the Gal4 activity via Gal80$^{ts}$. (D); Conditional inactivation of the pLB1+ neurons triggers a reversible stage 14 oocytes accumulation. Ovaries images of pLB1-Gal4/Tub-Gal80$^{ts}$, UAS-Kir2.1 flies were acquired with transmission light microscopy for 4 days at two different temperatures. (E); pLB1+ neurons conditional inactivation modifies the quantity and quality of oocytes,leading to an accumulation of stage 14 oocytes. The dynamic over four different time points and two different temperatures is shown. It is important to note that the switch from 21°C to 29°C might be stressful for all the lines since stage 14 oocytes accumulated in all of them. In (A and D), a prototypical stage 14 oocyte is indicated with a red asterisk. In (B), shown are the average numbers over time of different oocyte stages ± SEM from two cumulated independent trials with at least 18 females per genotype and condition used. In (C), shown are the average numbers of eggs laid per fly per 24h ± SEM over 5 days from two cumulated independent trials with at least 59 females per genotype and condition used. In (E), shown are the average numbers of different oocyte stages ± SEM over 4 days for one representative assay out of two independent trials with at least 10 females per genotype and condition used. For (B) and (E), on the x axis, 1–6 corresponds to early stages (from stage 1 to stage 6)oocytes ; 7–9 corresponds to the sum of stages 7, 8 and 9; 10 is for stage 10; 11–13 is for the sum of stages 11, 12 and 13; 14 is for stage 14 and Apo is for apoptotic events, all identified via DAPI staining. * indicates p<0.05; ** indicates p<0.01; *** indicates p<0.0001; [+] and [#] indicate statistically significant differences between the test and the controls, but not all of them (see detailed statistics for *Figure 9E*). In (B), non-parametric t-test, Mann-Whitney test; in (C and E), non-parametric ANOVA, Dunn's multiple comparison test between the genotypes or treatments. Details including n values and genotypes can be found in the detailed lines, conditions and, statistics for the figure section.

DOI: https://doi.org/10.7554/eLife.50559.036

The following source data and figure supplements are available for figure 9:

**Source data 1.** Egg laying raw date for *Figure 9C*.
DOI: https://doi.org/10.7554/eLife.50559.039

**Figure supplement 1.** Peptidoglycan exposure leads to a reversible accumulation of stage 14 oocytes.
DOI: https://doi.org/10.7554/eLife.50559.037

**Figure supplement 2.** Conditional inactivation of pLB1+ neurons leads to a reversible accumulation of stage 14 oocytes.
DOI: https://doi.org/10.7554/eLife.50559.038

Fadd-IR (*Khush et al., 2002*) (Kindly provided by P. Meier); Tdc2-Gal4 (kindly provided by H. Scholz); nSyb-LexA (BDSC Cat# 51951, RRID:BDSC_51951); Elav-Gal80 (kindly provided by D. Herman); UAS>stop>GFPmCD8 (BDSC Cat# 30125, RRID:BDSC_30125); w- (BDSC Cat# 3605, RRID:BDSC_3605); UAS-TβH-IR (BDSC Cat# 27667, RRID:BDSC_27667); Tdc2-LexA (BDSC Cat# 52242, RRID:BDSC_52242); Tub>Gal80> (BDSC Cat# 38879, RRID:BDSC_38879); LexAop-FLP (BDSC Cat# 55819, RRID:BDSC_55819); 8XLexAop2-FLP (BDSC Cat# 55820, RRID:BDSC_55820); UAS>stop>-Kir2.1 (kindly provided by K Anderson's lab); HS-FLP (kindly provided by F. Schnorrer); OTD-FLP (kindly provided by K Anderson's Lab); Tsh-LexA, LexAop-Gal80 (kindly provided by S Birman's Lab); Tsh-Gal4 (kindly provided by M. Landgraf); 13XLexAop2-GFPmCD8 (BDSC Cat# 32203, RRID:BDSC_32203); LexAop-2xhrGFPnls (BDSC Cat# 29955, RRID:BDSC_29955); Dsx-LexA (BDSC Cat# 54785, RRID:BDSC_54785); Dsx-FLP (kindly provided by S. Goodwin). UAS>stop>TRPA1 (BDSC Cat# 66871, RRID:BDSC_66871); Tub-Gal80ts, UAS-Kir2.1 (kindly provided by B. Prud'homme's lab); UAS-GCaMP6s (BDSC Cat# 42746, RRID:BDSC_42746).

Flies were grown at 25°C on a yeast/cornmeal medium in 12h/12h light/dark cycle-controlled incubators. For 1 L of food, 8.2 g of agar (VWR, cat. #20768.361), 80 g of cornmeal flour (Westhove, Farigel maize H1) and 80 g of yeast extract (VWR, cat. #24979.413) were cooked for 10 min in boiling water. 5.2 g of Methylparaben sodium salt (MERCK, cat. #106756) and 4 mL of 99% propionic acid (CARLOERBA, cat. #409553) were added when the food had cooled down.

Cloning pLB1-LexA pLB1 DNA fragment corresponding to pLB1-Gal4 (*Kurz et al., 2017*) was cloned by Gateway into pBP nlsLexA::p65Uw vector (RRID:Addgene_26230). This vector was injected into y$^1$w- P{nos-phiC31\int.NLS}X; P{CaryP}attP40 embryos (modified from BDSC Cat# 25709, RRID:BDSC_25709) and screened in F1 for white + transformants.

## PGRP-LB::GFP

A PGRP-LB::GFP fusion protein transgenic line was obtained by inserting, via CRISPR mediated recombination, the eGFP cDNA at the C-term end of the PGRP-LB protein. The GFP cDNA was inserted in the 3' most coding exon, resulting in all PGRP-LB isoforms (RA; RC and RD) tagged with GFP. The P donor PGRP-LB-GFP was obtained by cloning the GFP cDNA flanked by 1 kb of PGRP-

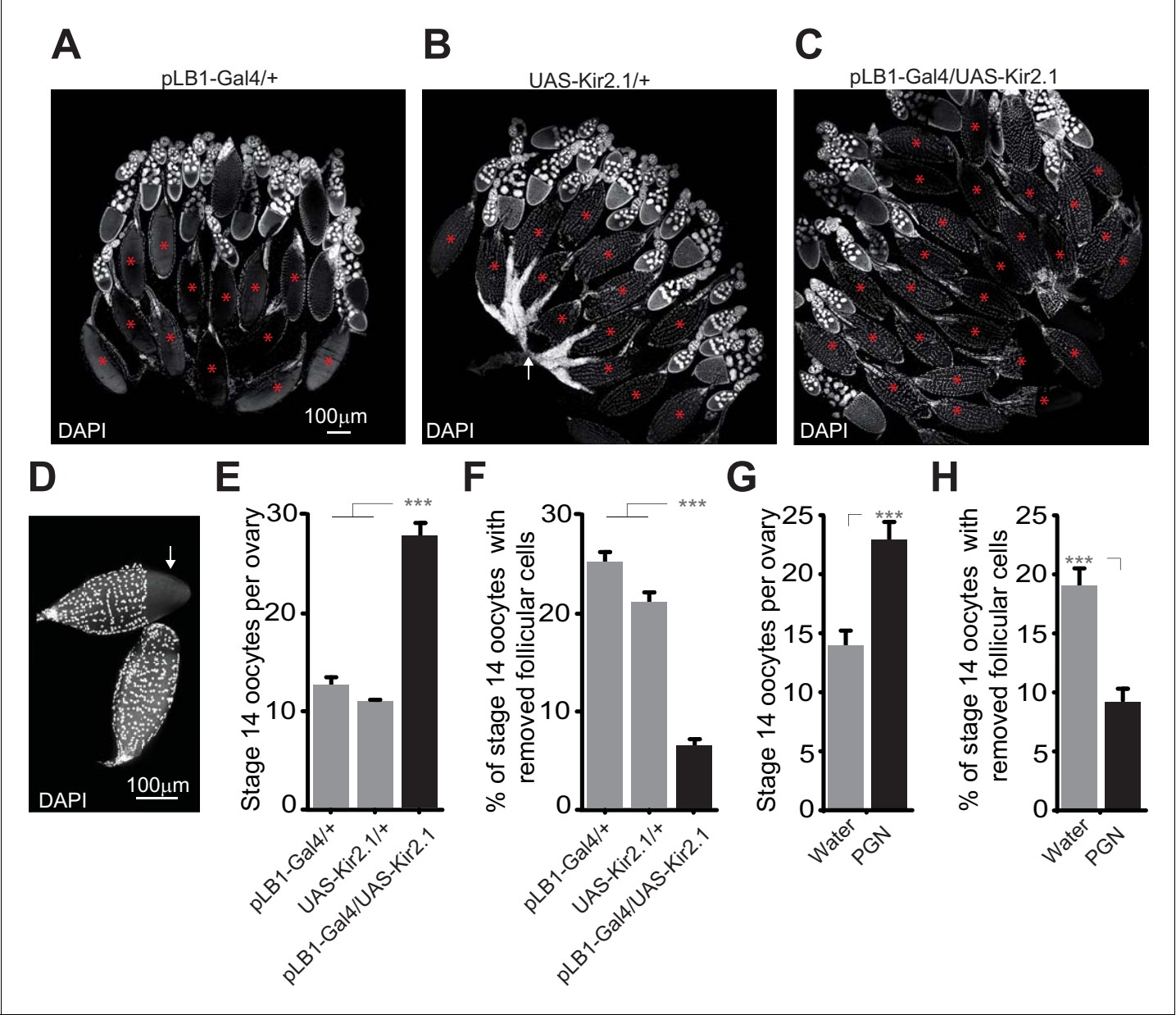

**Figure 10.** Impairing the activity of pLB1+ cells or injecting peptidoglycan leads to an accumulation of mature oocytes and a defect in follicular cells rupture. (A–C); Reducing the activity of pLB1+ cells leads to an accumulation of mature oocytes (stage14). DAPI staining of ovaries from control flies (A and B) or animals with reduced activity of the pLB1+ cells (C). Mature oocytes are indicated with a red asterisk and an oocyte with follicular cells ruptured is indicated with a white arrow. (D); Follicular cells surrounding the mature oocytes are removed before the entry in lateral oviducts. DAPI staining of stage 14 oocytes with follicular cells partly removed (white arrow) or fully covering the oocyte (bottom). (E and G); Stage 14 oocytes accumulate in the ovaries when pLB1+ cells activity is impaired (E) or when peptidoglycan is injected (G). (F and H); The ratio of mature oocytes with removed follicular cells is decreased when pLB1+ cells activity is impaired (F) or when peptidoglycan is injected (H). For (E and G), shown is the average number of stage 14 oocytes per ovary ± SEM in 5 day-old females (E) or 6 hours post-treatment (G) from three independent trials with at least 50 ovaries per genotype and condition used. For (F and H), shown are the ratios of stage 14 oocytes with removed follicular cells ± SEM in 5 day-old females (F) or 6 hours post-treatment (H) from three independent trials with at least 50 ovaries per genotype and condition used. For (E and F), *** indicates p<0.0001; non-parametric ANOVA, Dunn's multiple comparison test. For (G and H), *** indicates p<0.0001; non-parametric t-test, Mann-Whitney test. Details including n values and genotypes can be found in the detailed lines, conditions and, statistics for the figure section.
DOI: https://doi.org/10.7554/eLife.50559.040

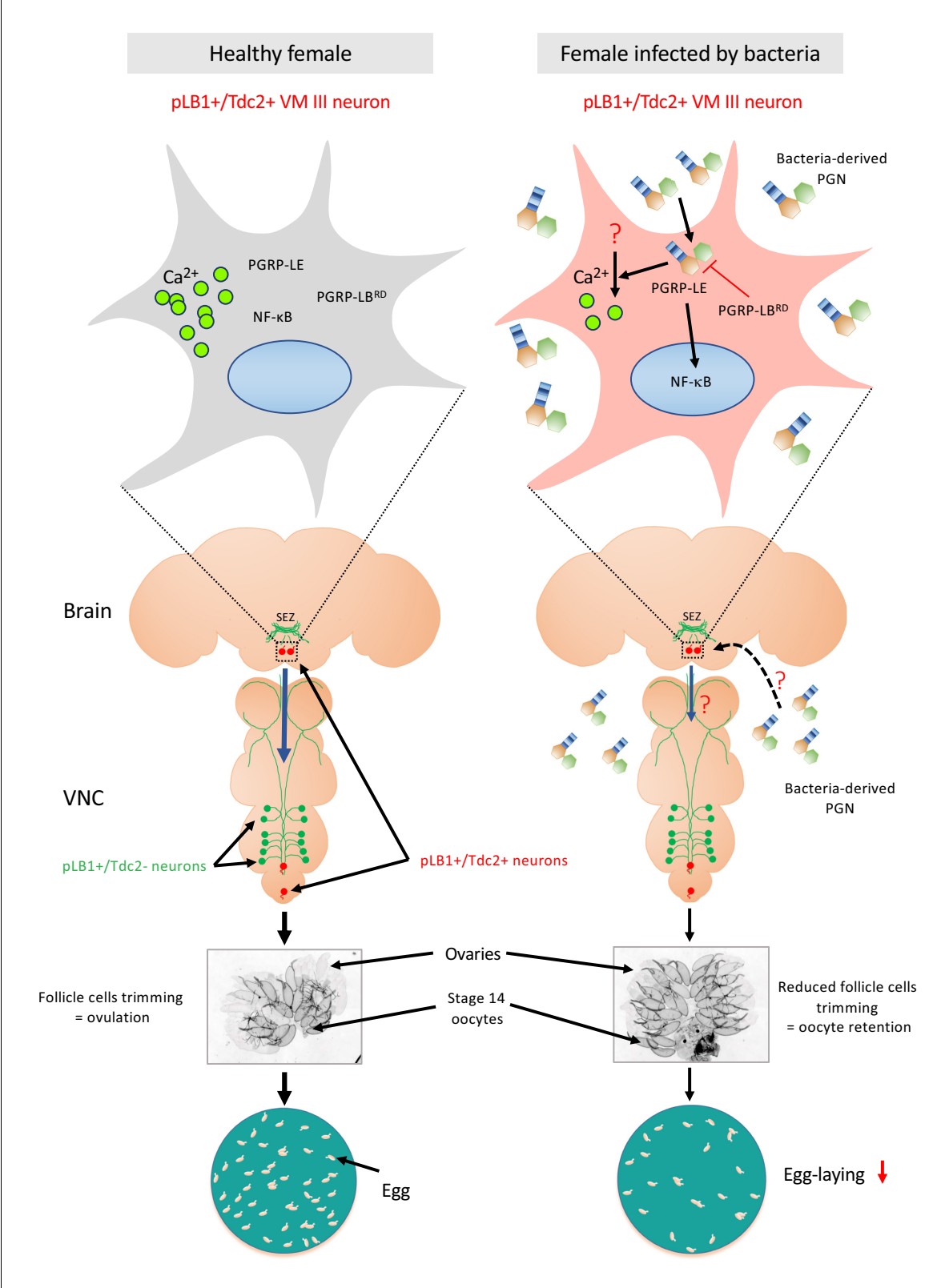

**Figure 11.** Diagram summarizing the effect of peptidoglycan sensing by pLB1+/Tdc2+ brain neurons on female oviposition. (**Left**) Around 20 neurons distributed in the brain and the ventral nerve cord (VNC) express immune genes such as PGRP-LB (called pLB1+ neurons, labeled in green and red). Among them, very few are also expressing the enzyme Tdc2 indicating that they are octopaminergic neurons (labeled in red), with the most robust pLB1 expression for those localized in the brain ventral midline, the VM cluster (delineated by the box with dashed line and schematically magnified in *Figure 11 continued on next page*

*Figure 11 continued*

the top box). In homeostasic conditions, mated females produce mature oocytes surrounded by follicular cells. Rupture of these follicular cells (trimming) leads to egg ovulation. (**Right**) Upon bacterial infection either systemic or enteric, cell wall peptidoglycan (PGN) is released by proliferating bacteria and transported into the hemolymph. Intracytosolic peptidoglycan sensing via the PGRP-LE Rc, in the very few brain pLB1+/Tdc2+ neurons (but not in the VNC pLB1+/Tdc2+ neurons), leads to cell-autonomous NF-κB pathway activation. This causes a reduction of follicular cells trimming in mature (stage 14) oocytes, hence a reduction of egg-laying in infected females. Since some pLB1+/Tdc2+ neurons of the VM cluster (VM III sub-cluster specifically) are sending descending projections toward the VNC and that the direct addition of peptidoglycan reduces their intracytosolic calcium level, we believe that peptidoglycan directly reduces the activity of these cells. Then, these brain pLB1+/Tdc2+ cells could mediate their effect via secondary neurons present in the VNC. It remains to be understood how PGN from the hemolymph is reaching brain neurons, how PGN stimulation reduces calcium levels, how this is linked to NF-κB pathway signaling and whether secondary neurons modulate ovulation. Moreover, we believe that pLB1+/Tdc2- neurons control other behaviors that are probably modulated by infection.

DOI: https://doi.org/10.7554/eLife.50559.041

LB homology arms into the Bluescript vector using the following primers (fw 5' arm: CGGGC TGCAGGAATTCCAAACAGCTCGCACGCAAAATACAA, rv 5' arm: AACAGCTCCTCGCCCTTGC TCACGACCTTGGGCGCAGCTGGC; fw 3' arm: GCTGTACAAGCACCGGTCCACGTAGGCTGGA TTGGAGGGCCCTCA, rv 3' arm: GGGCCCCCCCTCGAGGCTGCCGCCGAAATCAATCCAATAGC). Guide RNA (GCTGCGCCCAAGGTCTAGGC), was cloned into pCFD3–dU6: 3 gRNA (RRID:Addg-ene_49410). y[1] M{w[+mC]=nos-Cas9.P}ZH-2A w[*] embryos were injected with both donor and guide vectors (pCFD3-gRNA ; P donor PGRP-LB::GFP). F1 larvae were screened for GFP expression and positive line were confirmed molecularly.

## Oviposition assay

In order to ease the quantification of the laid eggs, a blue food dye (E133, Le meilleur du chef) was incorporated (1%) into the media used for the oviposition assays (Blue-tube). When the egg-laying index was used, it corresponds to the ratio between the number of eggs laid by a treated female and the average number of eggs per tube laid by the untreated animals during a specific time. An oviposition ratio of 1 indicates that the treatment did not impact the oviposition of the tested female during the time course of the experiment. *PGN injections*: one-day-old animals were harvested from tubes kept at 25°C. Males and females were mixed in one tube with no more than 40 individuals per tube and the proportion male: female was 1:1. Tubes were kept at 25°C and flies shifted to fresh tubes every 2 days. On day 5, females were used for injections. PGN or endotoxin-free water was used and injected using a nanojector (Nanojet II, Drummond Scientific Company, PA, USA). PGN is from *E. coli* (Invivogen, ref 14C14-MM, CA, USA) and was resuspended in endotoxin-free water at 200 μg/mL. 60 nL of PGN solution was injected in the thorax. All the flies including control animals

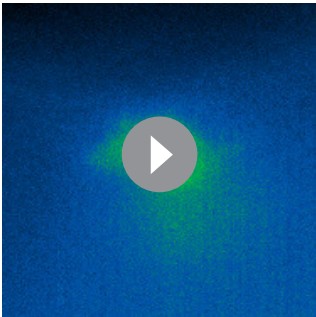

**Video 1.** Effect of Ringer's solution stimulation on Tdc2>GCaMP6s VM II/III neurons *in vivo.* GFP recording of an *in vivo* Tdc2>GCaMP6s fly brain in the VM II/III octopaminergic sub-clusters region. Brains of flies from which the head capsule has been removed were exposed to Ringer's solution. GFP signal was recorded every 500 ms.

DOI: https://doi.org/10.7554/eLife.50559.032

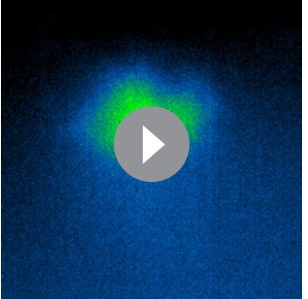

**Video 2.** Effect of peptidoglycan solution stimulation on Tdc2>GCaMP6sVM II/III neurons *in vivo.* GFP recording of an *in vivo* Tdc2>GCaMP6s fly brain in the VM II/III octopaminergic sub-clusters region. Brains of flies from which the head capsule has been removed were exposed to peptidoglycan solution. GFP signal was recorded every 500 ms.

DOI: https://doi.org/10.7554/eLife.50559.033

**Video 3.** Effect of Ringer's solution stimulation on pLB1>FLP/Tub>Gal80>,Tdc2>GCaMP6s VM III neurons *ex vivo*. GFP recording of an *ex vivo* pLB1>FLP/Tub>Gal80>,Tdc2>GCaMP6s fly brain in the VM III octopaminergic sub-cluster region. Dissected brains were mounted in Ringer's solution and stimulated with the same control solution. GFP signal was recorded every 2 s.
DOI: https://doi.org/10.7554/eLife.50559.034

**Video 4.** Effect of peptidoglycan solution stimulation on pLB1>FLP/Tub>Gal80>,Tdc2>GCaMP6s VM III neurons *ex vivo*. GFP recording of an *ex vivo* pLB1>FLP/Tub>Gal80>,Tdc2>GCaMP6s fly brain in the VM III octopaminergic cluster region. Dissected brains were mounted in Ringer's solution and stimulated with peptidoglycan solution. GFP signal was recorded every 2 s.
DOI: https://doi.org/10.7554/eLife.50559.035

were anesthetized on $CO_2$ pad. *Egg-lay assay*: Treated animals were then transferred on fresh Blue-tube with dry yeast (Fermipan) added on top of each tube right before the egg-lay period. When PGN injection was tested, animals were allowed to lay eggs for 6h, two females per tube. Otherwise, flies were one per tube and laid eggs during 24h. In order to maximize the efficiency of the transgenes, animals were stored at 29°C one day before the treatment and kept at this temperature during the egg-lay assay (except for experiments involving thermosensitive transgenes). Injections were always performed between ZT0 and ZT6. Eggs were counted for each tube 6h or 24h later. The eggs were not blindly counted. Raw egg counts are available as Data source.

## Mating assay

Virgin females were collected after eclosion and kept in groups of 10–15 individuals at 25°C, whereas naïve males where singularly isolated after eclosion and kept at 25°C. Mating experiments were performed in a behavioral room at 24°C. To assay fly receptivity, a 6 days-old virgin female was introduced in a laboratory made chamber (17 mm diameter x 8 mm height) with a naïve w- male. During 10 min, flies recovered from the flipping and adapted to the new chamber. Then, flies were recorded with a camera (Logitech HD pro webcam c920) for 1h and receptivity was quantified as the percentage of flies that mated within this time. Latency was defined as the time at which mating was starting. Successfully mated females were isolated in vials where they laid eggs for 48h at 25°C. Mated females were again introduced in a chamber with a naïve w- male to assay re-mating. The latter was defined as the percentage of mated pairs within 1h.

## Ovaries content and trimming quantification

Flies were reared and harvested as for the oviposition assays. After a 20s EtOH bath, animals were dissected at RT in 1X phosphate-buffered saline (PBS), then ovaries were fixed for 10 min in 4% paraformaldehyde on ice and rinsed three times in 1X PBS. The ovaries were then incubated with DAPI in the dark for 10 min. Finally, ovaries were gently opened on a glass slide in a 1X PBS drop. Oocytes stages, apoptotic events and trimming were visually quantified per ovary using DAPI and an Axio-Imager APO Z1 apotome microscope (Zeiss, Germany).

## Imaging and Immuno-cytochemistry

Adult brains and VNCs were dissected in PBS and fixed for 15 min in 4% paraformaldehyde at RT. After fixation, the tissues were rinsed three times for 10 min in PBS-T (PBS + 0.3% Triton X-100) and blocked in 2,5% bovine serum albumin (BSA; Sigma-Aldrich) in PBS-T for 30 min. Next, samples were incubated with the first antibody diluted in 0,5% BSA in PBS-T overnight at 4°C. The tissues were then washed three times and incubated with the secondary antibody diluted in 0,5% BSA in PBS-T for 2h at RT. Samples were rinsed three times and mounted on slides using Vectashield (Vector Laboratories, Ca, USA) fluorescent mounting medium, with or without DAPI. Images were captured with either a Leica SP8 confocal microscope (in this case, tissues were scanned with 20X oil immersion objective) or an AxioImager APO Z1 apotome microscope (10X or 20x air objectives were used). For the detection of endogenous PGRP-LB::GFP, brains of PGRP-LB::GFP Tdc2-Gal4/UAS-

Tomato-mCD8 flies were dissected in PBS, fixed on ice for 3 min, washed three times and then mounted without immuno-staining. Images were captured with a Spinning Disk Ropper 2 Cam.

## Calcium imaging

For *in vivo* calcium imaging studies, fed mated females were aged from 5 to 7 days. The preparation consisted of a fly suspended by the neck on a plexiglass block (2 × 2×2.5 cm). Flies were anesthetized on ice for 1h. The flies, with the proboscis facing the center of the block, were immobilized using an insect pine (0.1 mm diameter) placed on the neck. The ends of the pin were fixed on the block with beeswax (Deiberit 502, Siladent, 209212). Then the head was glued on the block with a drop of rosin (Gum rosin, Sigma-Aldrich, 60895; dissolved in ethanol at 70%) to avoid any movements. Therefore, the anterior part of the head is oriented towards the objective of the microscope. Flies were placed in a humidified box to allow the rosin to harden for 1h. A plastic coverslip with a hole corresponding to the head width was placed on top of the head and fixed on the block with beeswax. The plastic coverslip was sealed on the cuticle with two-component silicon (Kwik-Sil, World Precision Instruments). 100 µL of Ringer's saline (130 mM NaCl, 5 mM KCl, 2 mM $MgCl_2$, 2 mM $CaCl_2$, 36 mM saccharose, 5 mM HEPES, pH 7.3) were placed on the head. The antenna area, the tracheas, and the fat body were removed. The gut was cut without damaging the brain to allow visual access to the ventral part of the SEZ. The exposed brain was rinsed twice with 100 µL of Ringer's saline. GCaMP6s fluorescence was viewed with a Leica DM600B microscope under a 25x water objective. GCaMP6s was excited using a Lumencor diode light source at 482 nm ±25. Emitted light was collected through a 505–530 nm band-pass filter. Images were collected every 500ms using an Orca Flash 4.0 camera and processed using Leica MM AF 2.2.9. Each experiment consisted of 70 to 100 images (before application) followed by 160 images of recording after the addition of 100 µL Ringer's saline (control) or 100 µL of PGN solution (200 µg PGN/mL diluted in Ringer's saline for a final PGN concentration on the preparation of 100 µg/mL).

For the *ex vivo* calcium imaging, 5–7 day-old females were immobilized on ice and brains were dissected in Ringer's saline and were located in a silicone (SYLGARD 184 Silicone Elastomer Kit, DOW) coated cover slip. To avoid any movements during the recording, brains were fixed to the silicone support by using two insect pins at the level of the optical lobes. A 50 µL drop of Ringer's solution was used to cover the brain. GCaMP6s fluorescence was recorded with a Confocal spinning disk Yokogawa coupled with a Nikon Ti Eclipse inverted microscope and 2 CMOS capture cameras (Evolve 512) under a 20x air objective. For *GCaMPs excitation*, a laser wavelength at 491 nm (30% laser power; time of exposure 200 ms) was used. Images were taken every 2s and each experiment consisted in around 40 measurements (before application) followed by 100 recording images after the addition of 50 µL Ringer's saline (control) or 50 µL of PGN solution (for a final PGN concentration on the preparation of 100 µg/mL). Data were analyzed using FIJI (https://fiji.sc/) as previously described (*Silbering et al., 2012*).

## Statistical analyses and graphics

The Prism software (GraphPad Prism, RRID:SCR_002798) was used for statistical analyses. Our sets of data were tested for normality using the D'Agostino-Pearson omnibus test, and some of our data did not pass the normality test. Consequently, we used non-parametric tests for all the data sets, that is the unpaired ANOVA, Kruskal-Wallis test and specifically the Dunn's multiple comparisons test as well as the unpaired Mann-Whitney two-tailed test. In addition, for mating and remating datasets, we used the Fisher exact t-test. Moreover, we do not show one experiment representative of the different biological replicates, but all the data generated during the independent experiments in one graph.

## Antibodies table

| Antibody | Source | Dilution |
|---|---|---|
| Chicken anti-GFP | Aves Labs Cat# GFP-1020, RRID:AB_10000240 | 1:1000 |

*Continued on next page*

*Continued*

| Antibody | Source | Dilution |
| --- | --- | --- |
| Rabbit anti-RFP | Rockland Cat# 600-401-379, RRID:AB_2209751 | 1:1000 |
| Rat anti-RFP [5F8] | ChromoTek Cat# 5f8-100, RRID:AB_2336064 | 1:1000 |
| Mouse anti-NC82 | DSHB Cat# nc82, RRID:AB_2314866 | 1:40 |
| Rat anti-Elav-7E8A10 | DSHB Cat# Rat-Elav-7E8A10 anti-elav, RRID:AB_528218 | 1:50 |
| Rabbit anti-Tdc2 | Abcam Cat# ab128225, RRID:AB_11142389 | 1:1000 |
| Rabbit anti-A-Allatostatin | Jena Bioscience (ABD-062) | 1:2000 |
| Rabbit anti-CCAP | Jena Bioscience (ABD-033) | 1:6000 |
| Rat anti-leukokinin | P. Herrero's lab gift | 1:1000 |
| Rabbit anti-leukokinin | Dick R Nässel's lab gift | 1:2000 |
| Alexa Fluor 488 Donkey anti-Chicken IgY (IgG) (H+L) | Jackson Immuno Research Labs Cat# 703-545-155, RRID:AB_2340375 | 1:500 |
| Alexa Fluor568 donkey anti-mouse IgG (H+L) | Thermo Fisher Scientific Cat# A10037, RRID:AB_2534013 | 1:500 |
| Alexa Fluor647 donkey anti-mouse IgG (H+L) | Jackson Immuno Research Labs Cat# 715-605-151, RRID:AB_2340863 | 1:500 |
| Alexa Fluor 488 donkey anti-rabbit IgG (H+L) | Thermo Fisher Scientific Cat# A-21206, RRID:AB_2535792 | 1:500 |
| Alexa Fluor 568 donkey anti-rabbit IgG (H+L) | Thermo Fisher Scientific Cat# A10042, RRID:AB_2534017 | 1:500 |
| Cy 3 donkey anti-rat IgG (H+L) | Jackson Immuno Research Labs Cat# 712-165-153, RRID:AB_2340667 | 1:500 |
| Alexa Fluor647 donkey anti-rat IgG (H+L) | Jackson Immuno Research Labs Cat# 712-605-153, RRID:AB_2340694 | 1:500 |
| Rabbit anti-Bursicon | B. White's lab gift | 1 :2500 |

## Detailed lines, conditions and, statistics for the figures

Lines and conditions used for *Figure 1*

For 1A and 1B:

Genotypes of tested animals

- pLB1-Gal4; UAS-Tomato-mCD8

Reagents and tools

- antibody against nc82; antibody against Tomato; Leica SP8 confocal microscope

For 1C, 1D:
Genotypes of tested animals

- pLB1-Gal4, UAS > stop > GFPmCD8; LexAop-FLP/nSyb-LexA

Reagents and tools

- antibody against nc82; antibody against GFP; Leica SP8 confocal microscope

For 1F:
Genotypes of tested animals (at 23°C: n flies/mean eggs exp1/mean eggs exp2 //at 29°C: n flies/ mean eggs exp1/mean eggs exp2)

- pLB1-Gal4/+ (20/43.7/49.4 //20/52.5/51.6)
- UAS-TRPA1/+ (16/29.17/50.5 //19/39.4/46.6)
- Elav-Gal80; UAS-TRPA1/+ (20/50.3/48.4 //20/52.7/45.7)
- pLB1-Gal4/UAS-TRPA1 (17/43.4/45 //19/78/85)
- Elav-Gal80; UAS-TRPA1/pLB1-Gal4 (19/41.7/41.7 //19/46.6/58.4)

## Detailed statistics for *Figure 1F*

| Dunn's multiple comparisons test | Significant | Summary | Adjusted P Value |
|---|---|---|---|
| pLB1-Gal4/+ vs. +/UAS-TRPA1; Elav-Gal80 | No | ns | >0,9999 |
| pLB1-Gal4/+ vs. +/UAS-TRPA1 | No | ns | >0,9999 |
| pLB1-Gal4/+ vs. pLB1-Gal4/UAS-TRPA1 | Yes | **** | <0,0001 |
| pLB1-Gal4/+ vs. pLB1-Gal4/UAS-TRPA1; Elav-Gal80 | No | ns | >0,9999 |
| +/UAS-TRPA1; Elav-G80 vs. 0 > UAS-TRPA1 | No | ns | >0,9999 |
| +/UAS-TRPA1; Elav-G80 vs. pLB1 > UAS-TRPA1 | Yes | **** | <0,0001 |
| +/UAS-TRPA1; Elav-G80 vs. LB1 > UAS-TRPA1; Elav-Gal80 | No | ns | 0,0793 |
| +/UAS-TRPA1 vs. pLB1 > UAS-TRPA1 | Yes | **** | <0,0001 |
| +/UAS-TRPA1 vs. p LB1-Gal4/UAS-TRPA1; Elav-Gal80 | No | ns | >0,9999 |
| pLB1-Gal4/UAS-TRPA1 vs. pLB1-Gal4/UAS-TRPA1; Elav-Gal80 | Yes | * | 0,0189 |

For 1G:
Genotypes of tested animals (at 29°C: n flies/mean eggs exp1/mean eggs exp2/mean eggs exp3)

- pLB1-Gal4/+ (59/71/73/72)
- UAS-TβH-IR/+ (58/74/70/72)
- pLB1-Gal4/UAS- TβH-IR (59/50/54/60.5)

## Detailed statistics for *Figure 1G*

| Dunn's multiple comparisons test | Significant | Summary | Adjusted P Value |
|---|---|---|---|
| pLB1-Gal4/UAS-TβH IR vs+/UAS-TβH-IR | Yes | **** | <0,0001 |
| pLB1-Gal4/UAS-TβH-IR vs. pLB1-Gal4/+ | Yes | **** | <0,0001 |
| +/UAS-TβH-IR vs. pLB1-Gal4/+ | No | ns | >0,9999 |

## Lines and conditions used for *Figure 2*

For 2A-A'':
Genotypes of tested animals

- pLB1-Gal4, UAS > stop > GFPmCD8; LexAop-FLP/nSyb-LexA

Reagents and tools

- antibody against Tdc2; antibody against GFP; antibody against Elav; Leica SP8 confocal microscope

For 2B-B':
Genotypes of tested animals

- pLB1-Gal4, UAS > stop > GFPmCD8; LexAop-FLP/nSyb-LexA

Reagents and tools

- antibody against Tdc2; antibody against GFP; antibody against nc82; Leica SP8 confocal microscope

## Lines and conditions used for *Figure 2—figure supplement 1*

For 2A-A''':
Genotypes of tested animals

- pLB1-Gal4, UAS > stop > GFPmCD8; LexAopFLP/Tdc2-LexA

Reagents and tools

- antibody against Tdc2; antibody against GFP; antibody against nc82; Leica SP8 confocal microscope

For 2B-B''':
Genotypes of tested animals

- pLB1-Gal4, UAS > stop > GFPmCD8; LexAopFLP/Tdc2-LexA

Reagents and tools

- antibody against Tdc2; antibody against GFP; antibody against nc82; Leica SP8 confocal microscope

## Lines and conditions used for *Figure 2—figure supplement 2*

Genotypes of tested animals

- Tdc2-LexA, LexAop-GFPnls/LexAop-GFPmCD8

Reagents and tools

- antibody against GFP; antibody against nc82; Leica SP8 confocal microscope

## Lines and conditions used for *Figure 2—figure supplement 3*

Genotypes of tested animals

- pLB1-Gal4, UAS > stop > GFPmCD8; LexAop-FLP/nSyb-LexA

Reagents and tools

- antibody against Tdc2; antibody against GFP; Leica SP8 confocal microscope

## Lines and conditions used for *Figure 2—figure supplement 4*

Genotypes of tested animals

- pLB1-Gal4, UAS > stop > GFPmCD8; LexAopFLP/nSyb-LexA

Reagents and tools

- antibody against Allatostatin A; antibody against GFP; antibody against nc82; Leica SP8 confocal microscope

## Lines and conditions used for *Figure 2—figure supplement 5*

Genotypes of tested animals

- pLB1-Gal4, UAS > stop > GFPmCD8; LexAopFLP/nSyb-LexA

Reagents and tools

- antibody against Bursicon (from B. White); antibody against GFP; antibody against nc82; Leica SP8 confocal microscope

## Lines and conditions used for *Figure 2—figure supplement 6*

Genotypes of tested animals

- pLB1-Gal4, UAS > stop > GFPmCD8; LexAopFLP/nSyb-LexA

Reagents and tools

- antibody against CCAP (From B. White); antibody against GFP; antibody against nc82; Leica SP8 confocal microscope

## Lines and conditions used for *Figure 2—figure supplement 7*

Genotypes of tested animals

- pLB1-Gal4, UAS > stop > GFPmCD8; LexAopFLP/nSyb-LexA

Reagents and tools

- antibody against Leucokinin (from D. Nassel and P. Herrero); antibody against GFP; antibody against nc82; Leica SP8 confocal microscope

## Lines and conditions used for *Figure 3*

For 3A:

Genotypes of tested animals (n flies/mean % exp1/mean % exp2/mean % exp3/mean% exp4/mean% exp5/mean % exp6)

- pLB1-Gal4/+ (70/50%/50%/60%/60%/50%/55%)
- UAS-Kir2.1/+ (70/60%/40%/80%/80%/50%/40%)
- Tdc2-Gal4/+ (80/60%/60%/60%/40%/60%/50%)
- Tdc2-Gal4/UAS-Kir2.1 (80/100%/70%/100%/70%/50%/67%)
- pLB1-Gal4/UAS-Kir2.1 (80/70%/50%/80%/60%/30%/47%)

## Detailed statistics for *Figure 3A*

| Fisher's exact test | Significant | Summary | Adjusted P Value |
| --- | --- | --- | --- |
| pLB1-Gal4/+ vs. UAS-Kir2.1/+ | No | ns | 1 |
| pLB1-Gal4/+ vs. Tdc2-Gal4/+ | No | ns | 0,87 |
| pLB1-Gal4/+ vs. Tdc2-Gal4/UAS-Kir2.1 | Yes | ** | 0009 |

*Continued on next page*

*Continued*

| Fisher's exact test | Significant | Summary | Adjusted P Value |
|---|---|---|---|
| pLB1-Gal4/+ vs. pLB1-Gal4/UAS-Kir2.1 | No | ns | 0.74 |
| Tdc2-Gal4/+ vs. UAS-Kir2.1/+ | No | ns | 0,87 |
| Tdc2-Gal4/+ vs. Tdc2-Gal4/UAS-Kir2.1 | Yes | * | 0018 |
| Tdc2-Gal4/+ vs. pLB1-Gal4/UAS-Kir2.1 | No | ns | 1 |
| UAS-Kir2.1/+ vs. Tdc2-Gal4/UAS-Kir2.1 | Yes | ** | 0009 |
| UAS-Kir2.1/+ vs. pLB1-Gal4/UAS-Kir2.1 | No | ns | 0,74 |
| Tdc2-Gal4/UAS-Kir2.1 vs. pLB1-Gal4/UAS-Kir2.1 | Yes | * | 0028 |

For 3B:

Genotypes of tested animals (n flies/mean time in seconds exp1/mean time in seconds exp2/mean time in seconds exp3/mean time in seconds exp4)

- pLB1-Gal4/+ (27/1528/1382/1395/1480)
- UAS-Kir2.1/+ (24/1643/1160/1087/1740)
- Tdc2-Gal4/+ (31/1896/640/1729/1171)
- Tdc2-Gal4/UAS-Kir2.1 (40/770/674/815/868)
- pLB1-Gal4/UAS-Kir2.1 (34/1214/931/1109/1443)

## Detailed statistics for *Figure 3B*

| Mann-Whitney test | Significant | Summary | Adjusted P Value |
|---|---|---|---|
| pLB1-Gal4/+ vs. UAS-Kir2.1/+ | No | ns | 0,7341 |
| pLB1-Gal4/+ vs. Tdc2-Gal4/+ | No | ns | 0,5641 |
| pLB1-Gal4/+ vs. Tdc2-Gal4/UAS-Kir2.1 | Yes | ** | 0,0081 |
| pLB1-Gal4/+ vs. pLB1-Gal4/UAS-Kir2.1 | No | ns | 0,3415 |
| Tdc2-Gal4/+ vs. UAS-Kir2.1/+ | No | ns | 0,8652 |
| Tdc2-Gal4/+ vs. Tdc2-Gal4/UAS-Kir2.1 | Yes | * | 0,0272 |
| Tdc2-Gal4/+ vs. pLB1-Gal4/UAS-Kir2.1 | No | ns | 0,7180 |
| UAS-Kir2.1/+ vs. Tdc2-Gal4/UAS-Kir2.1 | Yes | * | 0,0234 |
| UAS-Kir2.1/+ vs. pLB1-Gal4/UAS-Kir2.1 | No | ns | 0,6584 |
| Tdc2-Gal4/UAS-Kir2.1 vs. pLB1-Gal4/UAS-Kir2.1 | Yes | * | 0,0471 |

For 3C-D':

Genotypes of tested animals

- Males pLB1-Gal4, UAS > stop > GFPmCD8; LexAop-FLP/nSyb-LexA

Reagents and tools

- antibody against Tdc2; antibody against GFP; antibody against nc82; Leica SP8 confocal microscope

For 3E-F':
Genotypes of tested animals

- pLB1-Gal4, UAS > stop > GFPmCD8/Dsx-FLP

Reagents and tools

- antibody against Tdc2; antibody against GFP; Leica SP8 confocal microscope

## Lines and conditions used for *Figure 3—figure supplement 1*

Genotypes of tested animals; (n flies/mean % exp1/mean % exp2/mean % exp3/mean% exp4) pLB1-Gal4/+ (44/0%/0%/0%/0%)

- UAS-Kir2.1/+ (42/0%/0%/14%/0%)
- Tdc2-Gal4/+ (55/0%/0%/0%/0%)
- Tdc2-Gal4/UAS-Kir2.1 (53/20%/20%/4%/0%)
- pLB1-Gal4/UAS-Kir2.1 (42/0%/0%/4%/0%)

## Detailed statistics for *Figure 3—figure supplement 1*

| Fisher's exact test | Significant | Summary | Adjusted P Value |
|---|---|---|---|
| pLB1-Gal4/+ vs. Tdc2-Gal4/UAS-Kir2.1 | No | ns | 0,2 |
| Tdc2-Gal4/+ vs. Tdc2-Gal4/UAS-Kir2.1 | No | ns | 0,1 |
| UAS-Kir2.1/+ vs. Tdc2-Gal4/UAS-Kir2.1 | No | ns | 0,6 |
| Tdc2-Gal4/UAS-Kir2.1 vs. pLB1-Gal4/UAS-Kir2.1 | No | ns | 0,6 |

## Lines and conditions used for *Figure 3—figure supplement 2*

Genotypes of tested animals

- pLB1-Gal4, UAS > stop > GFPmCD8; LexAop-FLP/nSyb-LexA

Reagents and tools

- antibody against Tdc2; antibody against GFP; antibody against nc82; Leica SP8 confocal microscope

## Lines and conditions used for *Figure 3—figure supplement 3*

Genotypes of tested animals

- pLB1-Gal4, UAS > stop > GFPmCD8; LexAop-FLP/nSyb-LexA

Reagents and tools

- antibody against Tdc2; antibody against GFP; antibody against nc82; Leica SP8 confocal microscope

## Lines and conditions used for *Figure 3—figure supplement 4*

For 4A-A''':
Genotypes of tested animals

- pLB1-Gal4, UAS > stop > GFPmCD8; LexAop-FLP/Dsx-LexA

Reagents and tools

- antibody against Tdc2; antibody against GFP; Leica SP8 confocal microscope

For 4B-B''':
Genotypes of tested animals

- pLB1-Gal4, UAS > stop > GFPmCD8; LexAop-FLP/Dsx-FLP

Reagents and tools

- antibody against Tdc2; antibody against GFP; Leica SP8 confocal microscope

## Lines and conditions used for *Figure 4*

For 4A:

Genotypes of tested animals; (at 29°C: n flies/mean eggs exp1/mean eggs exp2/mean eggs exp3)

- pLB1-Gal4/+ (34/51/65/-)
- pLB1-Gal4, Tdc2-LexA/+ (51/70/63/63)
- UAS-TTx/+ (38/61/72/-)
- Tub >Gal80>; UAS-TTx; LexAop-FLP/+ (49/61/64/63)
- pLB1-Gal4/UAS-TTx (39/27/44)
- pLB1-Gal4, TDC2-LexA/UAS-TTx (20/39/-/-)
- pLB1-Gal4/Tub >Gal80>; UAS-TTx; LexAop-FLP (50/65/71/62)
- pLB1-Gal4, Tdc2-LexA/Tub >Gal80>; UAS-TTx; LexAop-FLP (54/40/41/42)

## Detailed statistics for *Figure 4A*

| Dunn's multiple comparisons test | Significant | Summary | Adjusted P Value |
|---|---|---|---|
| +/Tub > Gal80>; UAS-TTx; LexAop-FLP vs. +/UAS-TTx | No | ns | >0,9999 |
| +/Tub > Gal80>; UAS-TTx; LexAop-FLP vs. pLB1-Gal4,pTdc2-LexA/+ | No | ns | >0,9999 |
| +/Tub > Gal80>; UAS-TTx; LexAop-FLP vs. pLB1-Gal4/Tub > Gal80>; UAS-TTx; LexAop-FLP | No | ns | >0,9999 |
| +/Tub > Gal80>; UAS-TTx; LexAop-FLP vs. LB1_G4,Tdc2_LexA/Tub > Gal80>; UAS-TTx; LexAop-FLP | Yes | **** | <0,0001 |
| +/Tub > Gal80>; UAS-TTx; LexAop-FLP vs. pLB1-Gal4/UAS-TTx | Yes | **** | <0,0001 |
| +/Tub > Gal80>; UAS-TTx; LexAop-FLP vs. pLB1-Gal4,Tdc2-LexA/UAS-TTx | Yes | **** | <0,0001 |
| +/Tub > Gal80>; UAS-TTx; LexAop-FLP vs. pLB1-Gal4/+ | No | ns | >0,9999 |
| +/UAS-TTx vs. pLB1-Gal4,Tdc2-LexA/+ | No | ns | >0,9999 |

*Continued on next page*

*Continued*

| Dunn's multiple comparisons test | Significant | Summary | Adjusted P Value |
|---|---|---|---|
| +/UAS-TTx vs. pLB1-Gal4/Tub > Gal80>; UAS-TTx;LexAop-FLP | No | ns | >0,9999 |
| +/UAS-TTx vs. pLB1-Gal4,Tdc2-LexA/Tub > Gal80>; UAS-TTx;LexAop-FLP | Yes | **** | <0,0001 |
| +/UAS-TTx vs. pLB1Gal4/UAS-TTx | Yes | **** | <0,0001 |
| +/UAS-TTx vs. pLB1Gal4,Tdc2-LexA/UAS-TTX | Yes | **** | <0,0001 |
| +/UAS-TTx vs. pLB1-Gal4/+ | No | ns | 0,5856 |
| pLB1-Gal4,Tdc2-LexA/+ vs. pLB1-Gal4/Tub > Gal80>; UAS-TTx; LexAop-FLP | No | ns | >0,9999 |
| pLB1-Gal4,Tdc2-LexA/+ vs. pLB1-Gal4,Tdc2-LexA/Tub > Gal80>; UAS-TTx;LexAop-FLP | Yes | **** | <0,0001 |
| pLB1-Gal4,Tdc2-LexA/+ vs. pLB1-Gal4/UAS-TTx | Yes | **** | <0,0001 |
| pLB1-Gal4,Tdc2-LexA/+ vs. pLB1-Gal4,Tdc2-LexA/UAS-TTx | Yes | **** | <0,0001 |
| pLB1-Gal4,Tdc2-LexA/+ vs. pLB1-Gal4/+ | No | ns | >0,9999 |
| pLB1-Gal4/Tub > Gal80>; UAS-TTx; LexAop-FLP vs. pLB1-Gal4,Tdc2-LexA/Tub > Gal80>; UAS-TTx;LexAop-FLP | Yes | **** | <0,0001 |
| pLB1-Gal4/Tub > Gal80>;UAS-TTx; LexAop-FLP vs. pLB1-Gal4/UAS-TTx | Yes | **** | <0,0001 |
| pLB1-Gal4/Tub > Gal80>; UAS-TTx; LexAop-FLP vs. pLB1-Gal4,Tdc2-LexA/UAS-TTx | Yes | **** | <0,0001 |
| pLB1-Gal4/Tub > Gal80>;UAS-TTx; LexAop-FLP vs. pLB1-Gal4/+ | No | ns | 0,3232 |
| pLB1-Gal4,Tdc2-LexA/Tub > Gal80>; UAS-TTx;LexAop-FLP vs. pLB1-Gal4/UAS-TTx | No | ns | >0,9999 |
| pLB1-Gal4,Tdc2-LexA/Tub > Gal80>; UAS-TTx;LexAop-FLP vs. LB1-Gal4; Tdc2-LexA/UAS-TTx | No | ns | >0,9999 |
| pLB1-Gal4,Tdc2-LexA/Tub > Gal80>; UAS-TTx;LexAop-FLP vs. pLB1-Gal4/+ | Yes | *** | 0,0003 |
| pLB1-Gal4/UAS-TTx vs. pLB1-Gal4,Tdc2-LexA/UAS-TTx | No | ns | >0,9999 |
| pLB1-Gal4/UAS-TTx vs. pLB1-Gal4/+ | Yes | **** | <0,0001 |
| pLB1-Gal4,Tdc2-LexA/UAS-TTx vs. pLB1-Gal4/+ | Yes | ** | 0,0054 |

For 4B:

Genotypes of tested animals (at 23°C: n flies/mean eggs exp1/mean eggs exp2/mean eggs exp3 //at 29°C: n flies/mean eggs exp1/mean eggs exp2/mean eggs exp3)

- Tdc2-LexA/+ (35/42/45/36//37/42/39/41)
- HS-FLP; pLB1-Gal4/+ (40/38.5/39/36//39/39.5/42/43)
- pLB1-Gal4, Tdc2-LexA/+ (40/38/40/32//39/44/38/36)
- pLB1-Gal4/+ (40/44/42/31//39/45/46/32)
- UAS > stop > TRPA1; LexAop-FLP/+ (40/45/45/34//40/48/47/43)
- pLB1-Gal4/UAS > stop > TRPA1; LexAop-FLP (40/39/37/35//40/44/44/42)
- Tdc2-LexA/UAS > stop > TRPA1; LexAop-FLP (40/34/32/35//37/35/37/34)
- HS-FLP; pLB1-Gal4/UAS > stop > TRPA1; LexAop-FLP (39/37/36/31//40/59/60/49)
- pLB1-Gal4, Tdc2-LexA/UAS > stop > TRPA1; LexAop-FLP (40/37/32/28//40/54/57/51)

## Detailed statistics for *Figure 4B*

| Dunn's multiple comparisons test | Significant? | Summary | Adjusted P Value |
|---|---|---|---|
| Tdc2Lex > 0 vs. LB1Gal4/HS-FLP > 0 | No | ns | >0,9999 |
| Tdc2Lex > 0 vs. LB1Gal4/Tdc2Lex > 0 | No | ns | >0,9999 |
| Tdc2Lex > 0 vs. LB1Gal4 > 0 | No | ns | >0,9999 |
| Tdc2Lex > 0 vs. 0 > stop > TRPA1/LexAopFLP | No | ns | >0,9999 |
| Tdc2Lex > 0 vs. LB1G4/Tdc2Lex > stop > TRPA1/LexAopFLP | Yes | **** | <0,0001 |
| Tdc2Lex > 0 vs. LB1G4/HS-FLP > stop > TRAP1/LexAopFLP | Yes | **** | <0,0001 |
| Tdc2Lex > 0 vs. LB1G4 > stop > TRPA1/LexaopFLP | No | ns | >0,9999 |
| Tdc2Lex > 0 vs. Tdc2Lex > stop > TRPA1/Lexaop-FLP | No | ns | 0,7602 |
| LB1Gal4/HS-FLP > 0 vs. LB1Gal4/Tdc2Lex > 0 | No | ns | >0,9999 |
| LB1Gal4/HS-FLP > 0 vs. LB1Gal4 > 0 | No | ns | >0,9999 |
| LB1Gal4/HS-FLP > 0 vs. 0 > stop > TRPA1/LexAopFLP | No | ns | >0,9999 |
| LB1Gal4/HS-FLP > 0 vs. LB1G4/Tdc2Lex > stop > TRPA1/LexAopFLP | Yes | **** | <0,0001 |
| LB1Gal4/HS-FLP > 0 vs. LB1G4/HS-FLP > stop > TRAP1/LexAopFLP | Yes | **** | <0,0001 |
| LB1Gal4/HS-FLP > 0 vs. LB1G4 > stop > TRPA1/LexaopFLP | No | ns | >0,9999 |
| LB1Gal4/HS-FLP > 0 vs. Tdc2Lex > stop > TRPA1/Lexaop-FLP | No | ns | >0,9999 |
| LB1Gal4/Tdc2Lex > 0 vs. LB1Gal4 > 0 | No | ns | >0,9999 |
| LB1Gal4/Tdc2Lex > 0 vs. 0 > stop > TRPA1/LexAopFLP | No | ns | >0,9999 |
| LB1Gal4/Tdc2Lex > 0 vs. LB1G4/Tdc2Lex > stop > TRPA1/LexAopFLP | Yes | **** | <0,0001 |
| LB1Gal4/Tdc2Lex > 0 vs. LB1G4/HS-FLP > stop > TRAP1/LexAopFLP | Yes | **** | <0,0001 |
| LB1Gal4/Tdc2Lex > 0 vs. LB1G4 > stop > TRPA1/LexaopFLP | No | ns | >0,9999 |
| LB1Gal4/Tdc2Lex > 0 vs. Tdc2Lex > stop > TRPA1/Lexaop-FLP | No | ns | 0,6815 |
| LB1Gal4 > 0 vs. 0 > stop > TRPA1/LexAopFLP | No | ns | >0,9999 |
| LB1Gal4 > 0 vs. LB1G4/Tdc2Lex > stop > TRPA1/LexAopFLP | Yes | **** | <0,0001 |
| LB1Gal4 > 0 vs. LB1G4/HS-FLP > stop > TRAP1/LexAopFLP | Yes | **** | <0,0001 |
| LB1Gal4 > 0 vs. LB1G4 > stop > TRPA1/LexaopFLP | No | ns | >0,9999 |
| LB1Gal4 > 0 vs. Tdc2Lex > stop > TRPA1/Lexaop-FLP | No | ns | 0,6919 |

*Continued on next page*

*Continued*

| Dunn's multiple comparisons test | Significant? | Summary | Adjusted P Value |
|---|---|---|---|
| 0 > stop > TRPA1/LexAopFLP vs. LB1G4/Tdc2Lex > stop > TRPA1/LexAopFLP | Yes | **** | <0,0001 |
| 0 > stop > TRPA1/LexAopFLP vs. LB1G4/HS-FLP > stop > TRAP1/LexAopFLP | Yes | **** | <0,0001 |
| 0 > stop > TRPA1/LexAopFLP vs. LB1G4 > stop > TRPA1/LexaopFLP | No | ns | >0,9999 |
| 0 > stop > TRPA1/LexAopFLP vs. Tdc2Lex > stop > TRPA1/Lexaop-FLP | No | ns | >0,9999 |
| LB1G4/Tdc2Lex > stop > TRPA1/LexAopFLP vs. LB1G4/HS-FLP > stop > TRAP1/LexAopFLP | No | ns | >0,9999 |
| LB1G4/Tdc2Lex > stop > TRPA1/LexAopFLP vs. LB1G4 > stop > TRPA1/LexaopFLP | Yes | **** | <0,0001 |
| LB1G4/Tdc2Lex > stop > TRPA1/LexAopFLP vs. Tdc2Lex > stop > TRPA1/Lexaop-FLP | Yes | **** | <0,0001 |
| LB1G4/HS-FLP > stop > TRAP1/LexAopFLP vs. LB1G4 > stop > TRPA1/LexaopFLP | Yes | **** | <0,0001 |
| LB1G4/HS-FLP > stop > TRAP1/LexAopFLP vs. Tdc2Lex > stop > TRPA1/Lexaop-FLP | Yes | ** | 0,0012 |
| LB1G4 > stop > TRPA1/LexaopFLP vs. Tdc2Lex > stop > TRPA1/Lexaop-FLP | No | ns | >0,9999 |

For 4C:

Genotypes of tested animals (at 29°C: n flies for water injection: PGN injection /mean eggs laid post water injection: PGN injection exp1/ mean eggs laid post water injection: PGN injection exp2)

- pLB1-Gal4, Tdc2-LexA/+ (51:51/32:19/19:10)
- Tub >Gal80>; LexAop-FLP; UAS-Fadd-IR/+ (52:52/26:16/28:17)
- Tdc2-LexA/Tub >Gal80>; LexAop-FLP; UAS-Fadd-IR (58:58/24:14/25:14)
- pLB1-Gal4/Tub >Gal80>; LexAop-FLP; UAS-Fadd-IR (58:58/26:16/33:17)
- pLB1-Gal4, Tdc2-LexA/Tub >Gal80>; LexAop-FLP; UAS-Fadd-IR (47:47/37:30/23:22)

Reagents and tools

- PGN/Water/Nanojector

## Detailed statistics for *Figure 4C*

| Dunn's multiple comparisons test | Significant | Summary | Adjusted P Value |
|---|---|---|---|
| pLB1-Gal4,Tdc2-LexA/Tub > Gal80>; LexAop-FLP;UAS-Fadd IR vs. pLB1-Gal4,Tdc2-LexA/+ | Yes | *** | 0,0001 |
| pLB1-Gal4,Tdc2-LexA/Tub > Gal80>; LexAop-FLP;UAS-Fadd IRvs. pLB1-Gal4/Tub > Gal80>; LexAop-FLP;UAS-Fadd IR | Yes | **** | <0,0001 |
| pLB1-Gal4,Tdc2-LexA/Tub > Gal80>; LexAop-FLP;UAS-Fadd IR vs. Tdc2-LexA/Tub > Gal80>; LexAop-FLP;UAS-Fadd IR | Yes | **** | <0,0001 |
| pLB1-Gal4,Tdc2-LexA/Tub > Gal80>; LexAop-FLP;UAS-Fadd IR vs. +/Tub > Gal80>; LexAop-FLP;UAS-Fadd IR | Yes | **** | <0,0001 |
| pLB1-Gal4,Tdc2-LexA/+ vs. pLB1-Gal4/Tub > Gal80>; LexAop-FLP;UAS-Fadd IR | No | ns | 0,5730 |

*Continued on next page*

*Continued*

| Dunn's multiple comparisons test | Significant | Summary | Adjusted P Value |
|---|---|---|---|
| pLB1-Gal4,Tdc2-LexA/+ vs. Tdc2-LexA/Tub > Gal80>; LexAop-FLP;UAS-Fadd IR | No | ns | >0,9999 |
| pLB1-Gal4,Tdc2-LexA/+ vs. +/Tub > Gal80>; LexAop-FLP;UAS-Fadd IR | No | ns | >0,9999 |
| pLB1-Gal4/Tub > Gal80>; LexAop-FLP;UAS-Fadd IR vs. Tdc2-LexA/Tub > Gal80>; L exAop-FLP;UAS-Fadd IR | No | ns | >0,9999 |
| pLB1-Gal4/Tub > Gal80>; LexAop-FLP;UAS-Fadd IR vs. +/Tub > Gal80>; LexAop-FLP;UAS-Fadd IR | No | ns | >0,9999 |
| Tdc2-LexA/Tub > Gal80>; LexAop-FLP;UAS-Fadd IR vs. +/Tub > Gal80>; LexAop-FLP;UAS-Fadd IR | No | ns | >0,9999 |

## Lines and conditions used for *Figure 5*

For 5A:

Genotypes of tested animals (at 29°C: n flies/mean eggs exp1/mean eggs exp2)

- pLB1-Gal4, UAS > stop > Kir2.1/+ (40/63/63)
- HS-FLP/+ (34/58/58)
- OTD-FLP/+ (39/63/66)
- pLB1-Gal4, UAS > stop > Kir2.1/HS-FLP (40/35/39)
- pLB1-Gal4, UAS > stop > Kir2.1/OTD-FLP (40/46/49)

## Detailed statistics for *Figure 5A*

| Dunn's multiple comparisons test | Significant | Summary | Adjusted P Value |
|---|---|---|---|
| HS-FLP/+ vs. pLB1-Gal4, UAS > stop > Kir2.1/+ | No | ns | 0,3044 |
| HS-FLP/+ vs. OTD-FLP/+ | No | ns | 0,1434 |
| HS-FLP/+ vs. pLB1-Gal4, UAS > stop > Kir2.1/OTD-FLP | Yes | ** | 0,0060 |
| HS-FLP/+ vs. pLB1-Gal4, UAS > stop > Kir2.1/HS-FLP | Yes | **** | <0,0001 |
| pLB1-Gal4, UAS > stop > Kir2.1/+ vs. OTD-FLP/+ | No | ns | >0,9999 |
| pLB1-Gal4,UAS > stop > Kir2.1/+ vs. pLB1-Gal4,UAS > stop > Kir2.1/OTD-FLP | Yes | **** | <0,0001 |
| pLB1-Gal4,UAS > stop > Kir2.1/+ vs. pLB1-Gal4,UAS > stop > Kir2.1/HS-FLP | Yes | **** | <0,0001 |
| OTD-FLP/+ vs. pLB1-Gal4, UAS > stop > Kir2.1/OTD-FLP | Yes | **** | <0,0001 |
| OTD-FLP/+ vs. pLB1Gal4, UAS > stop > Kir2.1/HS-FLP | Yes | **** | <0,0001 |
| pLB1-Gal4,UAS > stop > Kir2.1/OTD-FLP vs. pLB1-Gal4, UAS > stop > Kir2.1/HS-FLP | Yes | * | 0,0255 |

For 5B:

Genotypes of tested animals (at 29°C: n flies/mean eggs exp1/mean eggs exp2/ mean eggs exp3)

- UAS-Kir2.1/+ (50/51/70/70)
- Tsh-LexA, LexAop-Gal80/+ (48/50/70/68)

- pLB1-Gal4/+ (49/51/59/56)
- Tsh-LexA, LexOp-Gal80; UAS-Kir2.1/+ (50/46/70/68)
- pLB1-Gal4/Tsh-LexA, LexOp-Gal80 (50/39/67/67)
- pLB1-Gal4/UAS-Kir2.1 (69/28/28/41)
- pLB1-Gal4/Tsh-LexA, LexOp-Gal80; UAS-Kir2.1 (56/21/37/40)

## Detailed statistics for *Figure 5B*

| Dunn's multiple comparisons test | Significant | Summary | Adjusted P Value |
|---|---|---|---|
| pLB1-Gal4/UAS-Kir2.1 vs. pLB1-Gal4/UAS-Kir2.1; Tsh-LexA, LexAop-Gal80 | No | ns | >0,9999 |
| pLB1-Gal4/UASKir2.1 vs. pLB1-Gal4/Tsh-LexA, LexAop-Gal80 | Yes | **** | <0,0001 |
| pLB1-Gal4/UAS-Kir2.1 vs. +/Tsh-LexA, LexAop-Gal80; UAS-Kir2.1 | Yes | **** | <0,0001 |
| pLB1-Gal4/UAS-Kir2.1 vs. +/UAS-Kir2.1 | Yes | **** | <0,0001 |
| pLB1-Gal4/UAS-Kir2.1 vs. +/Tsh-LexA, Lexaop-Gal80 | Yes | **** | <0,0001 |
| pLB1-Gal4/UAS-Kir2.1 vs. pLB1-Gal4/+ | Yes | **** | <0,0001 |
| pLB1-Gal4/UASKir2.1; Tsh-LexA, LexAop-Gal80 vs. pLB1-Gal4/Tsh-LexA, LexAop-Gal80 | Yes | **** | <0,0001 |
| pLB1-Gal4/UAS-Kir2.1; Tsh-LexA, LexAop-Gal80 vs. +/Tsh-LexA, LexAop-Gal80; UAS-Kir2.1 | Yes | **** | <0,0001 |
| pLB1-Gal4/UAS-Kir2.1; Tsh-LexA, LexAop-Gal80vs. +/UAS-Kir2.1 | Yes | **** | <0,0001 |
| pLB1-Gal4/UAS-Kir2.1; Tsh-LexA, LexAop-Gal80 vs. +/Tsh-LexA, LexAop-Gal80 | Yes | **** | <0,0001 |
| pLB1-Gal4/UAS-Kir2.1; Tsh-LexA, LexAop-Gal80 vs. pLB1-Gal4/+ | Yes | **** | <0,0001 |
| pLB1-Gal4/Tsh-LexA, LexAop-Gal80 vs. +/Tsh-LexA, LexAop-Gal80;UAS-Kir2.1 | No | ns | >0,9999 |
| pLB1-Gal4/Tsh-LexA, LexAop-Gal80 vs. +/UAS-Kir2.1 | No | ns | >0,9999 |
| pLB1-Gal4/Tsh-LexA, LexAop-Gal80 vs. +/Tsh-LexA, LexAop-Gal80 | No | ns | >0,9999 |
| pLB1-Gal4/Tsh-LexA, LexAop-Gal80 vs. pLB1-Gal4/+ | No | ns | >0,9999 |
| +/Tsh-LexA, LexAop-Gal80;UAS-Kir2.1 vs. +/UAS-Kir2.1 | No | ns | >0,9999 |

*Continued on next page*

*Continued*

| Dunn's multiple comparisons test | Significant | Summary | Adjusted P Value |
|---|---|---|---|
| +/Tsh-LexA, Lexaop-Gal80;UAS-Kir2.1vs. +/Tsh-LexA, LexAop-Gal80 | No | ns | >0,9999 |
| +/Tsh-LexA, LexAop-Gal80;UAS-Kir2.1 vs. pLB1-Gal4/+ | No | ns | >0,9999 |
| +/UAS-Kir2.1 vs. +/Tsh-LexA, LexAop-Gal80 | No | ns | >0,9999 |
| +/UAS-Kir2.1 vs. pLB1-Gal4/+ | No | ns | 0,1350 |
| +/Tsh-LexA, LexAop-Gal80 vs. pLB1-Gal4/+ | No | ns | 0,3431 |

For 5C:

Genotypes of tested animals (at 29°C: n flies for water injection: PGN injection/mean eggs laid post water injection: PGN injection exp1/mean eggs laid post water injection: PGN injection exp2)

- UAS-Fadd-IR/+ (40:40/40:26/27:13)
- pLB1-Gal4/+ (40:40/43:28/30:17)
- Tsh-Gal4/+ (40:40/45:28/25:14)
- pLB1-Gal4/UAS-Fadd-IR (40:36/43:40/28:26)
- Tsh-Gal4/UAS-Fadd-IR (28:29/33:22/23:11)

Reagents and tools

- PGN/Water/Nanojector

## Detailed statistics for *Figure 5C*

| Dunn's multiple comparisons test | Significant | Summary | Adjusted P Value |
|---|---|---|---|
| pLB1-Gal4/+ vs. Tsh-Gal4/UAS-FaddIR | No | ns | >0,9999 |
| pLB1-Gal4/+ vs. Tsh-Gal4/+ | No | ns | >0,9999 |
| pLB1-Gal4/+ vs. pLB1-Gal4/UAS-FaddIR | Yes | **** | <0,0001 |
| pLB1-Gal4/+ vs. +/UAS-FaddIR | No | ns | >0,9999 |
| Tsh-Gal4/UAS-FaddIRvs. Tsh-Gal4/+ | No | ns | >0,9999 |
| Tsh-Gal-4/UAS-FaddIR vs. pLB1-Gal4/UAS-FaddIR | Yes | **** | <0,0001 |
| Tsh-Gal4/UAS-FaddIR vs. +/UAS-FaddIR | No | ns | >0,9999 |
| Tsh-Gal4/+vs. pLB1-Gal4/UAS-FaddIR | Yes | **** | <0,0001 |
| Tsh-Gal4/+ vs. +/UAS-FaddIR | No | ns | >0,9999 |
| pLB1-Gal4/UAS-FaddIR vs. +/UAS-FaddIR | Yes | **** | <0,0001 |

For 5D:

Genotypes of tested animals (at 29°C: n flies/mean ratio eggs water vs PGN exp1+exp2)

- UAS-Fadd-IR/+ (40/0.65)
- Tsh-LexA, LexOp-Gal80; UAS-Fadd-IR/+ (36/0.56)
- Tdc2-Gal4/+ (40/0.61) pLB1-Gal4/+ (38/0.66)
- Tdc2-Gal4/UAS-Fadd-IR (40/0.96)
- pLB1-Gal4/UAS-Fadd-IR (38/1.05)
- Tdc2-Gal4/Tsh-LexA, LexOp-Gal80; UAS-Fadd-IR (14/1.00)
- pLB1-Gal4/Tsh-LexA, LexOp-Gal80; UAS-Fadd-IR (32/0.92)

Reagents and tools

- PGN/Water/Nanojector

## Detailed statistics for *Figure 5D*

| Dunn's multiple comparisons test | Significant? | Summary | Adjusted P Value |
|---|---|---|---|
| Tdc2-Gal4/UAS-Fadd IR; Tsh-Lexa, LexOp-Gal80 vs. +/UAS-Fadd IR;Tsh-LexA, LexAop-Gal80 | Yes | * | 0,0236 |
| Tdc2-Gal4/UAS-Fadd IR; Tsh-LexA, LexOp-Gal80 vs. Tdc2-Gal4/+ | No | ns | 0,0759 |
| Tdc2-Gal4/UAS-Fadd IR; Tsh-LexA, LexOp-Gal80 vs. Tdc2-Gal4/UAS-Fadd IR | No | ns | >0,9999 |
| Tdc2-Gal4/UAS-Fadd IR; Tsh-LexA, LexOp-Gal80 vs. +/UAS-Fadd IR | No | ns | 0,1743 |
| Tdc2-Gal4/UAS-Fadd IR; Tsh-LexA, LexOp-Gal80 vs. pLB1-Gal4/+ | No | ns | 0,2508 |
| Tdc2-Gal4/UAS-Fadd IR; Tsh-LexA, LexOp-Gal80 vs. pLB1-Gal4/UAS-Fadd IR | No | ns | >0,9999 |
| Tdc2-Gal4/UAS-Fadd IR; Tsh-LexA, LexOp-Gal80 vs. pLB1-Gal4/UAS-Fadd IR; Tsh-LexA, LexOp-Gal80 | No | ns | >0,9999 |
| +/UAS-Fadd IR;Tsh-LexA, LexAopGal80 vs. Tdc2-Gal4/+ | No | ns | >0,9999 |
| +/UAS-Fadd IR;Tsh-LexA, LexOp-Gal80 vs. Tdc2-Gal4/UAS-Fadd IR | Yes | *** | 0,0001 |
| +/UAS-Fadd IR;Tsh-LexA, LexOp-Gal80 vs. +/UAS-Fadd IR | No | ns | >0,9999 |
| +/UAS-Fadd IR;Tsh-LexA, LexOp-Gal80 vs. pLB1-Gal4/+ | No | ns | >0,9999 |
| +/UAS-Fadd IR;Tsh-LexA, LexOp-Gal80 vs. pLB1-Gal4/UAS-Fadd IR | Yes | **** | <0,0001 |
| +/UAS-Fadd IR;Tsh-LexA, LexOp-Gal80 vs. pLB1-Gal4/UAS-Fadd IR; Tsh-LexA, LexOp-Gal80 | Yes | ** | 0,0036 |
| Tdc2-Gal4/+ vs. Tdc2-Gal4/UAS-Fadd IR | Yes | *** | 0,0008 |
| Tdc2-Gal4/+ vs. +/UAS-Fadd IR | No | ns | >0,9999 |
| Tdc2-Gal4/+ vs. pLB1-Gal4/+ | No | ns | >0,9999 |
| Tdc2-Gal4/+ vs. pLB1-Gal4/UAS-Fadd IR | Yes | **** | <0,0001 |

*Continued on next page*

*Continued*

| Dunn's multiple comparisons test | Significant? | Summary | Adjusted P Value |
|---|---|---|---|
| Tdc2-Gal4/+ vs. pLB1-Gal4/UAS-Fadd IR;Tsh-LexA, LexAop-Gal80 | Yes | * | 0,0180 |
| Tdc2-Gal4/UAS-Fadd IR vs. +/UAS-Fadd IR | Yes | ** | 0,0036 |
| Tdc2-Gal4/UAS-Fadd IR vs. pLB1-Gal4/+ | Yes | ** | 0,0078 |
| Tdc2-Gal4/UAS-Fadd IR vs. pLB1-Gal4/UAS-Fadd IR | No | ns | >0,9999 |
| Tdc2-Gal4/UAS-Fadd IR vs. pLB1-Gal4/UAS-Fadd IR;Tsh-LexA, LexAop-Gal80 | No | ns | >0,9999 |
| +/UAS-Fadd IR vs. pLB1-Gal4/+ | No | ns | >0,9999 |
| +/UAS-Fadd IR vs. pLB1-Gal4/UAS-Fadd IR | Yes | **** | <0,0001 |
| +/UAS-Fadd IR vs. pLB1Gal4/UAS-Fadd IR; Tsh-LexA, LexAop-Gal80 | No | ns | 0,0601 |
| pLB1-Gal4/+ vs. pLB1-Gal4/UAS-Fadd IR | Yes | **** | <0,0001 |
| pLB1-Gal4/+ vs. pLB1-Gal4/UAS-Fadd IR;Tsh-LexA, LexAop-Gal80 | No | ns | 0,1049 |
| pLB1-Gal4/UAS-Fadd IR vs. pLB1-Gal4/UAS-Fadd IR; Tsh-LexA, LexAop-Gal80 | No | ns | >0,9999 |

## Lines and conditions used for *Figure 5—figure supplement 1*
For 1A-C''':
Genotypes of tested animals

- pLB1-Gal4, UAS > stop > GFPmCD8/OTD-FLP

Reagents and tools

- antibody against Tdc2; antibody against GFP; Leica SP8 confocal microscope

For 1D-D':
Genotypes of tested animals

- pLB1-Gal4; UAS-Tomato-mCD8/Tsh-LexA, LexAop-Gal80

Reagents and tools

- antibody against nc82; antibody against GFP; Leica SP8 confocal microscope

## Lines and conditions used for *Figure 6*
Genotypes of tested animals

- pLB1-LexA/Tdc2-Gal4, UAS > stop > GFPmCD8; Lexaop-FLP

Reagents and tools

- antibody against GFP; antibody against nc82; Leica SP8 confocal microscope

## Lines and conditions used for *Figure 7*
Genotypes of tested animals

- pgrp-lb::gfp/Tdc-2-Gal4, UAS-Tomato-mCD8GFP

Reagents and tools

- No antibody staining; Leica SP8 confocal microscope

## Lines and conditions used for *Figure 8*

For A-C: *in vivo*

Genotypes and number (n) of tested animals

- Tdc2-Gal4, UAS-GCaMP6s
- Ringer's: n = 8 PGN: n = 13

Reagents and tools

- Leica SP8 confocal microscope, PGN, Ringer's solution

For D-F: *ex vivo*

Genotypes and number (n) of tested animals

- pLB1-LexA; LexAop-FLP/Tub > Gal80>; Tdc2-Gal4, UAS-GCaMP6s
- Ringer's; n = 10 PGN; n = 12

Reagents and tools

- Spinning Disk Ropper 2 Cam, PGN, Ringer's solution

## Detailed statistics for *Figure 8C*

Mann-Whitney test

| Table Analyzed | Data 1 |
| --- | --- |
| Column B | PGN 100 µg/mL |
| vs | vs |
| Column C | PGN 100 µg/mL VM1 |
| Mann Whitney test | |
| P value | 0,0010 |
| Exact or approximate P value? | Gaussian Approximation |
| P value summary | *** |
| Are medians signif. different? (p<0.05) | Yes |
| One- or two-tailed P value? | Two-tailed |
| Sum of ranks in column B,C | 97, 134 |
| Mann-Whitney U | 6000 |

## Detailed statistics for *Figure 8F*

| Table Analyzed | Data 1 |
| --- | --- |
| Column A | Ringer's Solution |
| vs | vs |
| Column B | PGN 100microg/mL |
| Mann Whitney test | |
| P value | 0,0001 |
| Exact or approximate P value? | Gaussian Approximation |

*Continued on next page*

*Continued*

| Table Analyzed | Data 1 |
|---|---|
| P value summary | *** |
| Are medians signif. different? (p<0.05) | Yes |
| One- or two-tailed P value? | Two-tailed |
| Sum of ranks in column A,B | 173, 80 |
| Mann-Whitney U | 2000 |

## Lines and conditions used for *Figure 9*

For 9A:
  Genotypes of tested animals

  • w-

  Reagents and tools

  • Axio-Imager APO Z1 apotome microscope

For 9B:
Genotypes of tested animals (at 25˚C: n flies for water injection: PGN injection)
(mix of two independent experiments; mean stage 1–6 post water injection: PGN injection/mean stage 7–9 post water injection: PGN injection/mean stage 10 post water injection: PGN injection/ mean stage 11–13 post water injection: PGN injection/mean stage 14 post water injection: PGN injection/mean Apoptotic oocytes post water injection: PGN injection)

  • w- (17:20)
  • 6h p. i. (33:32.7/33.2:35.7/5:3.85/8.5:5/18.1:27/2.1:3.7)
  • 24h p. i. (36.7:36.5/36.1:35.2/9.5:7.3/11.2:8.1/17.3:23/0.5:0.3)
  • 48h p. i. (36.7:39/39:36/7:6/7.8:9/20.5:21.8/0.35:0.45)

  Reagents and tools

  • PGN/Water/Nanojector

## Detailed statistics for *Figure 9B*

| Table Analyzed | oocytes count 6 hr water/PGN combo set 2 + set 1 |
|---|---|
| Column G | st14 PGN200 |
| vs. | vs, |
| Column A | st14 water |
| Mann-Whitney test | |
| P value | 0,0085 |
| Exact or approximate P value? | Exact |
| P value summary | ** |
| Significantly different (p<0.05)? | Yes |
| One- or two-tailed P value? | Two-tailed |
| Sum of ranks in column A,G | 238, 465 |
| Mann-Whitney U | 85 |
| Difference between medians | |
| Median of column A | 15, n = 17 |
| Median of column G | 24, n = 20 |
| Difference: Actual | 9 |

*Continued on next page*

*Continued*

| Table Analyzed | oocytes count 6 hr water/PGN combo set 2 + set 1 |
| --- | --- |
| Difference: Hodges-Lehmann | 7 |

| Table Analyzed | oocytes count 6 hr water/PGN combo set 2 + set 1 |
| --- | --- |
| Column H | st11-13 PGN200 |
| vs. | vs, |
| Column B | st 11–13 water |
| Mann Whitney test | |
| P value | 0,0004 |
| Exact or approximate P value? | Exact |
| P value summary | *** |
| Significantly different (p<0.05)? | Yes |
| One- or two-tailed P value? | Two-tailed |
| Sum of ranks in column B,H | 434,5, 268,5 |
| Mann-Whitney U | 58,5 |
| Difference between medians | |
| Median of column B | 9, n = 17 |
| Median of column H | 5, n = 20 |
| Difference: Actual | -4 |
| Difference: Hodges-Lehmann | -3 |

| Table Analyzed | oocytes count 6 hr water/PGN combo set 2 + set 1 |
| --- | --- |
| Column L | Apoptotic oocytes |
| vs. | vs, |
| Column F | Apoptotic oocytes |
| Mann Whitney test | |
| P value | 0,0350 |
| Exact or approximate P value? | Exact |
| P value summary | * |
| Significantly different (p<0.05)? | Yes |
| One- or two-tailed P value? | Two-tailed |
| Sum of ranks in column F,L | 255,5, 447,5 |
| Mann-Whitney U | 102,5 |
| Difference between medians | |
| Median of column F | 0, n = 17 |
| Median of column L | 3,5, n = 20 |
| Difference: Actual | 3,5 |

*Continued on next page*

*Continued*

| Table Analyzed | oocytes count 6 hr water/PGN combo set 2 + set 1 |
|---|---|
| Difference: Hodges-Lehmann | 1 |

| Table Analyzed | oocytes count 24 hr water/PGN combo set 2 + set1 |
|---|---|
| Column H | st11-13 PGN200 |
| vs. | vs, |
| Column B | st 11–13 water |
| Mann Whitney test | |
| P value | 0,0155 |
| Exact or approximate P value? | Exact |
| P value summary | * |
| Significantly different (p<0.05)? | Yes |
| One- or two-tailed P value? | Two-tailed |
| Sum of ranks in column B,H | 432,5, 308,5 |
| Mann-Whitney U | 98,5 |
| Difference between medians | |
| Median of column B | 12, n = 18 |
| Median of column H | 8,5, n = 20 |
| Difference: Actual | −3,5 |
| Difference: Hodges-Lehmann | -3 |

| Table Analyzed | oocytes count 24 hr water/PGN combo set 2 + set1 |
|---|---|
| Column I | st10 PGN200 |
| vs. | vs, |
| Column C | st 10 water |
| Mann Whitney test | |
| P value | 0,0266 |
| Exact or approximate P value? | Exact |
| P value summary | * |
| Significantly different (p<0.05)? | Yes |
| One- or two-tailed P value? | Two-tailed |
| Sum of ranks in column C,I | 426, 315 |
| Mann-Whitney U | 105 |
| Difference between medians | |
| Median of column C | 9,5, n = 18 |
| Median of column I | 6, n = 20 |
| Difference: Actual | −3,5 |
| Difference: Hodges-Lehmann | -2 |

For 9C:

Genotypes of tested animals (n flies/mean eggs day 1 (21°C) exp1-2/mean eggs day 2 (29°C) exp 1–2/mean eggs day 3 (21°C) exp1−2/mean eggs day 4 (21°C) exp 1–2/mean eggs day 5 (21°C) exp 1–2)

- pLB1-Gal4/+ (60/46.9–30.5/57.8–43.1/46.6–34.5/28.4–24.1/31.4–18.5)
- Tub-Gal80ts, UAS-Kir2.1/+ (60/48–27.5/60-40.6/45–32.8/29-24.3/31–18)
- pLB1-Gal4/Tub-Gal80ts, UAS-Kir2.1 (60/50-32/40.1–24.8/29.3–23.4/29.3–26.7/29.3–19.5)

## Detailed statistics for *Figure 9C*

| Dunn's multiple comparisons test | Significant? | Summary | Adjusted P Value |
|---|---|---|---|
| T80ts > Kir 29d2 vs. LB1 > 0 29d2 | No | ns | >0,9999 |
| T80ts > Kir 29d2 vs. Lb1/T80ts > Kir 29d2 | Yes | **** | <0,0001 |
| LB1 > 0 29d2 vs. Lb1/T80ts > Kir 29d2 | Yes | **** | <0,0001 |
| T80ts > Kir 23d3 vs. LB1 > 0 23d3 | No | ns | 0,6535 |
| T80ts > Kir 23d3 vs. Lb1/T80ts > Kir 23d3 | Yes | **** | <0,0001 |
| LB1 > 0 23d3 vs. Lb1/T80ts > Kir 23d3 | Yes | **** | <0,0001 |

For 9D:
Genotypes of tested animals

- pLB1-Gal4/Tub-Gal80ts, UAS-Kir2.1

Reagents and tools

- Axio-Imager APO Z1 apotome microscope

For 9E:
Genotypes of tested animals (n flies Day 1-2-3-4 (mix of two independent experiments)); mean stage 1–6 Day 1-2-3−4/mean stage 7–9 Day 1-2-3−4/mean stage 10 Day 1-2-3−4/mean stage 11–13 Day 1-2-3−4/mean stage 14 Day 1-2-3−4/mean Apoptotic oocytes Day 1-2-3-4)

- pLB1-Gal4/+ (10-11-9-10; 30.9–31.8-31.4–28.6/30.1–30.7-32-25.3/6.6–3.9-6.4–4.2-/8–4.9-7.4–5.7/9.8–27.9-4.4–10.3/0.7–1.8-0.7–0.4)
- Tub-Gal80ts, UAS-Kir2.1/+ (10-10-9-10; 33.9–34.1-34-31.3/34.5–30.2-35-28.2/7.3–4.1-4.6–6.1/9.6–5.4-7.4–3.8/9.2–26.1-13.8-6/0-1.1–0.5-0.2)
- pLB1-Gal4/Tub-Gal80ts, UAS-Kir2.1 (10-9-11-10; 32-32-31.9–29.1/33.4–31.9-31.5–28.7/6.8–6.1-4.3–3.9/10.3–3.8-5-6.4/8.2–45.2-29.6–9.4/0.4-0-0.9–1)

## Detailed statistics for *Figure 9E*
Day 2 (29°C)

| Dunn's multiple comparisons test | Significant? | Summary | Adjusted P Value |
|---|---|---|---|
| st14 Lb1 > 0 vs. st14 0/T80ts > Kir | No | ns | >0,9999 |
| st14 Lb1 > 0 vs. st14 LB1/T80ts > Kir | Yes | ** | 0,0013 |

*Continued on next page*

*Continued*

| Dunn's multiple comparisons test | Significant? | Summary | Adjusted P Value |
|---|---|---|---|
| st14 0/T80ts > Kir vs. st14 LB1/T80ts > Kir | Yes | *** | 0,0007 |

| Dunn's multiple comparisons test | Significant? | Summary | Adjusted P Value |
|---|---|---|---|
| Apoptotic oocytes LB1 > 0 vs. Apoptotic oocytes 0/T80ts > Kir | No | ns | 0,3395 |
| Apoptotic oocytes LB1 > 0 vs. Apoptotic oocytes LB1/T80ts > Kir | Yes | ** | 0,0021 |
| Apoptotic oocytes 0/T80ts > Kir vs. Apoptotic oocytes LB1/T80ts > Kir | No | ns | 0,2155 |

Day 3 (21°C)

| Dunn's multiple comparisons test | Significant? | Summary | Adjusted P Value |
|---|---|---|---|
| st14 Lb1 > 0 vs. st14 0/T80ts > Kir | No | ns | 0,0586 |
| st14 Lb1 > 0 vs. st14 LB1/T80ts > Kir | Yes | *** | 0,0001 |
| st14 0/T80ts > Kir vs. st14 LB1/T80ts > Kir | No | ns | 0,2082 |

## Lines and conditions used for *Figure 9—figure supplement 1*
Genotypes of tested animals

- w-

Reagents and tools

- PGN/Water/Nanojector; Axio-Imager APO Z1 apotome microscope, DAPI

## Lines and conditions used for *Figure 9—figure supplement 2*
Genotypes of tested animals

- pLB1-Gal4/+
- Tub-Gal80ts, UAS-Kir2.1/+
- pLB1-Gal4/Tub-Gal80ts, UAS-Kir2.1

Reagents and tools

- Axio-Imager APO Z1 apotome microscope, DAPI

## Lines and conditions used for *Figure 10*
For 10A-C:
Genotypes of tested animals

- pLB1-Gal4/+
- UAS-Kir2.1/+
- pLB1-Gal4/UAS-Kir2.1

Reagents and tools

- DAPI
- Axio-Imager APO Z1 apotome microscope

For 10D:
Genotypes of tested animals

- w-

Reagents and tools

- DAPI
- Axio-Imager APO Z1 apotome microscope

For 10E and 10F:
Genotypes of tested animals (at 29°C: n flies/mean stage 14 exp1/mean stage 14 exp2/mean stage 14 exp3//trimming % exp1/trimming % exp2/trimming % exp3)

- pLB1-Gal4/+ (64/11/12/15//30/24/19)
- UAS-Kir2.1/+ (64/14/12/8//22/24/18)
- pLB1-Gal4/UAS-Kir2.1 (61/25/26/35//8/7/3)

## Detailed statistics for *Figure 10E*

| Dunn's multiple comparisons test | Significant | Summary | Adjusted P Value |
|---|---|---|---|
| pLB1-Gal4/+ vs. +/UAS-Kir2.1 | No | ns | >0,9999 |
| pLB1-Gal4/+ vs. pLB1-Gal4/UAS-Kir2.1 | Yes | **** | <0,0001 |
| +/UAS-Kir2.1 vs. pLB1-Gal4/UAS-Kir2.1 | Yes | **** | <0,0001 |

## Detailed statistics for *Figure 10F*

| Dunn's multiple comparisons test | Significant | Summary | Adjusted P Value |
|---|---|---|---|
| pLB1-Gal4/+vs. +/UAS-Kir2.1 | No | ns | 0,2023 |
| pLB1-Gal4/+ vs. pLB1-Gal4/UAS-Kir2.1 | Yes | **** | <0,0001 |
| +/UAS-Kir2.1 vs. pLB1-Gal4/UAS-Kir2.1 | Yes | **** | <0,0001 |

For 10G and 10H:
Genotypes of tested animals

- w-

(at 29°C: n flies for water injection: PGN injection/mean stage 14 post water injection: PGN injection exp1/mean stage14 post water injection: PGN injection exp2/mean stage14 post water injection: PGN injection exp3//trimming% water: PGN exp1/trimming % water: PGN exp2/trimming % water: PGN exp3)

- (55:57/12:20/14:27/15:23//18:8/19:7/19:11)

Reagents and tools

- PGN/Water/Nanojector

## Detailed statistics for *Figure 10G*

**Mann-Whitney test**

| | |
|---|---|
| P value | <0,0001 |
| Exact or approximate P value? | Exact |
| P value summary | **** |
| Significantly different (p<0.05)? | Yes |
| One- or two-tailed P value? | Two-tailed |
| Sum of ranks in column A,B | 2312, 4016 |
| Mann-Whitney U | 772 |
| Difference between medians | |
| Median of column A | 12, n = 55 |
| Median of column B | 21, n = 57 |
| Difference: Actual | 9 |
| Difference: Hodges-Lehmann | 8 |

## Detailed statistics for *Figure 10H*

**Mann-Whitney test**

| | |
|---|---|
| P value | <0,0001 |
| Exact or approximate P value? | Exact |
| P value summary | **** |
| Significantly different (p<0.05)? | Yes |
| One- or two-tailed P value? | Two-tailed |
| Sum of ranks in column A,B | 3979, 2350 |
| Mann-Whitney U | 696,5 |
| Difference between medians | |
| Median of column A | 0,2, n = 55 |
| Median of column B | 0,07407, n = 57 |
| Difference: Actual | −0,1259 |
| Difference: Hodges-Lehmann | −0,1003 |

## Acknowledgements

We thank R Tavignot for help in screening for PGRP-LB::GFP transformants and JC Patel for advices with calcium imaging. This work was supported by CNRS and Agence Nationale de la recherche, ANR-11-LABX-0054 (Investissements d'Avenir–Labex INFORM), ANR BACNEURODRO (ANR-17-CE16-0023-01), Institut Universitaire de France to JR. This work was performed using France-Biol-maging infrastructure supported by (ANR-10 INBS-04–01). Research in YG's laboratory is supported by the CNRS, the 'Université de Bourgogne Franche-Comté", the Conseil Régional Bourgogne Franche-Comte (PARI grant), the FEDER (European Funding for Regional Economical Development), and the European Council (ERC starting grant, GliSFCo-311403).

## Additional information

### Funding

| Funder | Grant reference number | Author |
|---|---|---|
| Agence Nationale de la Recherche | 11-LABX-0054 | Julien Royet |

The funders had no role in study design, data collection and interpretation, or the decision to submit the work for publication.

### Author contributions

Ambra Masuzzo, Conceptualization, Formal analysis, Investigation, Writing—original draft, Writing—review and editing; Gérard Manière, Conceptualization, Validation, Investigation, Methodology; Annelise Viallat-Lieutaud, Conceptualization, Data curation, Investigation; Émilie Avazeri, Olivier Zugasti, Investigation; Yaël Grosjean, Conceptualization, Supervision, Project administration; C Léopold Kurz, Conceptualization, Data curation, Formal analysis, Supervision, Validation, Investigation, Methodology, Writing—original draft, Writing—review and editing; Julien Royet, Conceptualization, Supervision, Funding acquisition, Validation, Writing—original draft, Project administration, Writing—review and editing

### Author ORCIDs

Ambra Masuzzo (iD) https://orcid.org/0000-0002-2933-8924
C Léopold Kurz (iD) https://orcid.org/0000-0001-7081-3208
Julien Royet (iD) https://orcid.org/0000-0002-5671-4833

### Decision letter and Author response

Decision letter https://doi.org/10.7554/eLife.50559.047
Author response https://doi.org/10.7554/eLife.50559.048

## Additional files

### Supplementary files

• Transparent reporting form DOI: https://doi.org/10.7554/eLife.50559.073

### Data availability

All data generated or analysed during this study are included in the manuscript and supporting files.

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
