## [Decision Letter]

Thank you for submitting your work entitled "Peptidoglycan-dependent NF-κB activation in a small subset of brain octopaminergic neurons controls female oviposition" for consideration by *eLife*. Your article has been reviewed by a Senior Editor, a Reviewing Editor, and three reviewers. The following individuals involved in review of your submission have agreed to reveal their identity: Balint Z Kacsoh (Reviewer #3).

Our decision has been reached after consultation between the reviewers and the reviewing editor. Based on these discussions and the individual reviews below, we regret to inform you that your work will not be considered further for publication in *eLife*.

While all the reviewers find that the paper has many merits, they found that this paper contains fewer surprising results than the first one published in *eLife*. Thus, it was decided to reject the paper but allow resubmission. This will eventually give you more time to revise the paper and submit to *eLife* if you think that you can address the main critics. We cannot guarantee that reviewers will not come up with new objections upon resubmission, but we do make an attempt to find the same reviewers (unless you tell us not to).

To summarize, some of the main criticisms were:

1) Use ex vivo or in vivo Ca-imaging or methods such as CalexA or CaMPari to confirm that these neurons can be activated by PGN. Show that they indeed express PGRP-LB protein.

2) Strengthen the conclusions by showing that activation of just the PLB1+/ Tdc2+ neurons using TrpA1 or channel rhodopsin mimics the phenotype to test whether activation of these few neurons is enough to recapitulate the behavior.

3) Better characterize the output of these neurons.

4) Better characterize the egg deposition defects.

This is not a requirement, but the reviewing editor would find it interesting if the authors could decipher how Relish regulates the activity of these neurons. Can the silencing of Relish or Fadd in these neurons affect a gene involved in OA synthesis?

Finally, it should be mentioned that in a recent PNAS article, the laboratory of A. Heddi has generated (based on PGRP-LE) a detector of TCT that might allow its tracing and. this could be useful to mark TCT on sections.

At the time of current review of the manuscript, the editors and reviewers felt this paper would be appropriate for publication in *eLife* if the authors can reinforce their conclusions and provide more mechanistic insights on the process.

Reviewer #1:

In the present study, Masuzzo et al., follow up with their previous work (Kurz et al., 2017), in which they showed that bacterial infection or PGN reduces female egg laying through PGRP-LB-expressing OA neurons. Here, they use a battery of genetic intersectional methods to narrow down the responsible neurons to 1-2 OANs in the female's brain. Overall, the study is convincing, and the authors have made a remarkable effort to include all statistics and detailed description of the methods. Nevertheless, compared to their previous work, this paper contains fewer surprising results, and still leaves many of the previously unanswered questions open.

Major points:

1) Discussion section: the authors write 'However, precisely mapping the PG localization in the brain and identifying the exact path followed by this gut-born bacteria metabolite to reach the brain will require PGN tracing methods which are not yet available.' While I agree that this specific experiment is difficult, the paper would be strengthened significantly by the demonstration that PGN activates PGRP-LB expressing neurons in the brain. This can be achieved by ex vivo or in vivo Ca-imaging or methods such as CalexA or CaMPari. The lack of knowledge of how PGN or bacteria recruit PGRP-LB signaling in the brain is, in my opinion, the most significant weakness of this study. It appears to me that this could be addressed in a short time frame of a couple of months. In lieu of such evidence, the link between PGN and egg laying-regulating LB1-Gal4+ neurons in the brain remains weak.

2) Discussion section: the authors state that '2 brain and 10 VNC neurons are reproducibly expressing the PGN degrading enzyme PGRP-LB'. I assume this is based on the expression of pLB1-Gal4, which was used in Kurz et al., to rescue the PGRP-LB loss of function phenotype. Can the authors be 100% sure that all pLB1-Gal4 expressing cells truly express PGRP-LB without an antibody staining?

Reviewer #2:

In a previous (Kurz and Royet, 2017) elegant study, Kurz and Royet reported that internal sensing of bacterial PGN by a group of PRGP-LB expressing neurons are responsible for reducing female oviposition. They observed that PRGP-LB neurons are likely octopaminergic, and activation of NF-κB in these neurons are important for reduction in oviposition. Furthermore, they reported that inactivation of these PRGP-LB positive neurons by tetanus toxin, or activation by temperature sensitive TRPA1 channel can modulate egg laying. These observations raised two important questions: how octopaminergic neurons are activated by PGN, and how activation of these neurons in the CNS/PNS influence oviposition. In this paper they have taken the first step towards addressing these questions. Taking advantage of the genetic tools available in *Drosophila* they have beautifully isolated a small subset (in fact could be a single neuron) of neurons in the octopaminergic VM cluster in SOG that is necessary for the PGN-mediated alteration in oviposition.

Major issues:

While the observations reported in this paper are interesting, one can argue that it does not provide any new insight to the problem than what they have already published. Nonetheless, if indeed a single neuron can mediate PGN-dependent alteration of a very specific behavior, I think that should be worth publishing. However, to make that conclusion the authors at least need to do the following experiments and these experiments should not take more than two months. First, activation of just the PLB1+/ Tdc2+ neurons using TrpA1 or channelrhodopsin to test whether activation of these few neurons is enough to recapitulate the behavior. Other than few examples of 'command neurons", there are not too many examples where activation of one or two neurons recapitulates a specific behavior. Second, if the model is correct than the activity of these PLB1+/Tdc2+ neurons in SOG should be influenced by PGN. Again, using the same driver line, they should express calcium sensor GCamp and monitor activity as a change in fluorescence following application of PGN. The kinetics of the response would also carry valuable information. They can perform these experiments in isolated brain preparation with appropriate control. Finally, I would like to know if they have attempted to follow the output of these neurons from SOG or whether they have any clue to where it is going.

Reviewer #3:

Masuzzo et al., present their work, entitled, "Peptidoglycan-dependent NF-κB activation in a small subset of brain octopaminergic neurons controls female oviposition." In this study, the authors build upon their previous work (Kurz et al., 2017), which demonstrated that internal sensing of bacterial peptidoglycan reduces *Drosophila* female oviposition via NF-κB pathway activation in some neurons. However, until this paper, the neurons remained unknown. This study identifies several ventral nerve cord and brain octopaminergic neurons, expressing a NF-κB pathway components. Furthermore, the authors find that the brain neurons, specifically in the sub-esophageal zone, are required for oviposition depression, whereby activation triggers an egg-laying drop in response to infection. This was shown via peptidoglycan exposure or inhibition of PGRP-LB+ neuron activity, which subsequently impairs a specific ovulation event, thus linking PGN detection to the reduction of egg production.

The study uses particular GAL4 driver lines, i.e. pLB1, to observe a neuron requirement of TbBh-IR (Tyrosine decarboxylase) (Figure 1G), where when knocked down, flies oviposit fewer eggs. Using this finding, the authors stained against Tdc2 in brains and ventral nerve cords expressing pLB1>GFP to find the particular neurons of images. The authors then tested against other types of neuro peptides to rule out effects of other neuronal subtypes. While this demonstrated a series of negative results, they are very important to the identity of these cell types.

The authors then turn to identifying what pLB1 neurons do by comparing them to previously identified Tdc2 neurons. They authors do this inactivating them with UAS-Kir and find increased receptivity in female flies with marked Tdc2 neurons but not pLB1 neurons. Interestingly, pLB1+/Tdc2+ neurons were also detected in males, even though an egg laying response is the output (Figure 3C-D). To test functional significance, an intersectional line was generated and tested in an inducible manner via GAL80 and Tetx and kir, yielding strong reduction of oviposition. This was very elegantly performed in a temperature and temporally controlled manner. Additionally, the authors find that suppression of egg-laying drop 6 hours post PGN injection mediated by the knockdown of the NF-ΚB cascade in Tdc2+ and pLB1+ cells was not observed when only thoracic neurons were targeted, demonstrating functional relevance. The authors take this a step further and analyze the ovary, where they observe changes in follicle cell trimming (marked decrease in animals with PLb1GAL4>UASKir2.1) as the means by which fewer eggs are laid. This showed a remarkable increase in stage 14 oocyte numbers when pLb1 neurons were inhibited.

This is an extremely elegant and important study that highlights how a few neurons can respond to external stimuli and change subsequent internal physiology, in an octopamine-dependent manner. Flies are able to tune their responses and behavior by bacteria-derived metabolite! The manuscript is well written and reads as a story with a very logical flow. Additionally, I would like to commend the authors on the images, which are very high quality and I really enjoy having the cartoon depiction of the brains to accompany images (i.e. Figure 1).

Major comments:

I am left wanting more with respect to Figure 5 with respect to permanence or reversibility of the phenotype. The evidence that follicle cell trimming is altered and that there is an accumulation of stage 14 egg chambers is compelling. However, I am left wondering what the physiological relevance is to this particular phenotype. I would like to know if the eggs are simply delayed (diapause phenotype) and that these eggs will be laid, or perhaps they will not be laid at all? Thus, I would like to see an experiment that turns pLB1 GAL4 marked neurons back on (i.e. use UAS-shibireTS to turn off neurons to get ovary effect at 30C, then move them back to 22C and see how egg counts change) in order to see what the dynamics are of these eggs that are not yet laid. Alternatively, using the Kir construct with a GAL80 for temporal control in the same manner would be excellent. I believe that this experiment, regardless of the answer, will be important to perform to get at the underlying mechanism of what happens to the eggs. Images of ovaries (and associated quantification) after turning the pLB1 neurons back on would also be desired. Thus, can egg counts recover and does the ovary recover are important questions to finish answering.

Additionally, I do not see stage 11-12 egg chambers in the images of Figure 5C, but rather only stage 14 and stage 7/8 and below. Panels A and B have these later stage (11) egg chambers and have apoptosis on the DAPI staining, a hallmark of normal development. I would like to see this quantification as well, not on any new experimental animals, but on the existing ovary image. I believe that the authors have identified some really cool developmental biology regulation, which is not pointed out in the manuscript, but that should be highlighted. Perhaps these are not representative images, but the means by which the egg chambers are arrested is important to note. At minimum, the distribution of egg chamber stages should be quantified and talked about in the text. If the distribution is changed as the images suggest, please speculate on mechanism in the Discussion section - the mechanism does not need to be elucidated, just postulated about. This has the potential to be really exciting.

I believe that these two points are essential to the manuscript, as they could more fully demonstrate the functional significance what the physiological mechanism is that the neurons are communicating.

[Editors' note: further revisions were requested prior to acceptance, as described below.]

Thank you for submitting your article "Peptidoglycan-dependent NF-κB activation in a small subset of brain octopaminergic neurons controls female oviposition" for consideration by *eLife*. Your article has been reviewed by Utpal Banerjee as the Senior Editor, a Reviewing Editor, and three reviewers. The following individuals involved in review of your submission have agreed to reveal their identity: Balint Z Kacsoh (Reviewer #1); Kausik Si (Reviewer #3).

The reviewers have discussed the reviews with one another and the Reviewing Editor has drafted this decision to help you prepare a revised submission. We are pleased to inform you that the reviewers were rather positive about the manuscript and suggest a number of revisions that you should try to address. I suspect that most of them involve text modification.

Summary:

Masuzzo et al., present their work, entitled, "Peptidoglycan-dependent NF-κB activation in a small subset of brain octopaminergic neurons controls female oviposition." In this study, the authors build upon their previous work (Kurz et al., 2017), which demonstrated that internal sensing of bacterial peptidoglycan reduces *Drosophila* female oviposition via NF-κB pathway activation in some neurons. Here, they further characterize the very specific neuronal circuitry, involved in the immune induced oviposition behaviours phenotype. This study now is a major achievement in the burgeoning field of the intersection of neuronal signaling and immunity.

Essential revisions:

1) In Figure 8, the authors now show the effect of PGN stimulation on GCaMP fluorescence in vivo and ex vivo using either a broad Tdc2-Gal4 driver or their specific intersectional approach. In Figure 8A, the authors provide images of Tdc2 > GCaMP before and after PGN addition. Unfortunately, from these images it is unclear what we are looking at exactly given that Tdc2 labels many neurons. At minimum the authors should provide a scheme to explain the figure.

2) The authors point out that PGN reduces GCaMP fluorescence very transiently in their in vivo experiments. In the ex vivo experiment however, this is not the case. Here, the inhibition appears permanent/lasting. What explains this difference?

3) The authors speculate which genes might be regulated by PGN in LB1 neurons. They do not mention, however, how PGN might regulate calcium levels in these cells. This should be added to the Discussion section given that this is likely the main reason for the behavioral phenotype they observe.

---

## [Author Response]

While all the reviewers find that the paper has many merits, they found that this paper contains fewer surprising results than the first one published in eLife.

Of course, it is difficult to be authors and reviewers at the same time, but we do believe that the original manuscript contained exciting and surprising results. We were indeed very surprised that flies needed only a few neurons to regulate egg-laying post infection and we thought that genetically identifying this, or these, very few neuron(s) was an important achievement which really narrowed down the question. And this surprise was somewhat shared by reviewer 2 who wrote “there are not too many examples where activation of one or two neurons recapitulates a specific behavior”. In addition, we did not expect that these neurons will be located in the brain but rather in the VNC where most egg-laying controlling neurons are. These are the main reasons why we decided to submit our work to *eLife*. We would like also to mention that it is mentioned in *eLife* home page that this journal is interested to publish follow up of a previous manuscript. Which was exactly the case here. In any case, we are now providing new results in the revised version, which we think, really allow us to go further into the molecular mechanisms by which bacteria-derived PGN modulate fly behavior.

Thus, it was decided to reject the paper but allow resubmission. This will eventually give you more time to revise the paper and submit to eLife if you think that you can address the main critics. We cannot guarantee that reviewers will not come up with new objections upon resubmission, but we do make an attempt tofind the same reviewers (unless you tell us not to).To summarize, some of the main criticisms were:1) Use ex vivo or in vivo Ca-imaging or methods such as CalexA or CaMPari to confirm that these neurons can be activated by PGN.

We have performed in vivo and ex vivo calcium imaging with purified PGN and demonstrated that PGN stimulation induces a drop of Calcium level in pLB1 neurons.

Show that they indeed express PGRP-LB protein.

We have generated a fly line in which all endogenous PGRP-LB isoforms are GFP tagged and showed that endogenous PGRP-LB is indeed expressed in pLB1 neurons (and also in some non pLB1).

2) Strengthen the conclusions by showing that activation of just the PLB1+/ Tdc2+ neurons using TrpA1 or channel rhodopsin mimics the phenotype to test whether activation of these few neurons is enough to recapitulate the behavior.

We have performed the requested experiment that indicates that TrpA1 ectopic expression in pLB1+/Tdc2+ neurons is sufficient to stimulate egg laying.

3) Better characterize the output of these neurons.

We have, via the generation and characterization of a new pLB1lexA fly line, identified the precise nature of most plB1+/Tdc2+ neurons and showed that, at least for one the them, it sends projections towards the VNC.

4) Better characterize the egg deposition defects.

We have performed the requested experiments that confirms that stage 14 oocytes accumulation is transient and can be also obtained by only manipulating pLB1+ neurons.

Reviewer #1:In the present study, Masuzzo et al., follow up with their previous work (Kurz et al., 2017), in which they showed that bacterial infection or PGN reduces female egg laying through PGRP-LB-expressing OA neurons. Here, they use a battery of genetic intersectional methods to narrow down the responsible neurons to 1-2 OANs in the female's brain. Overall, the study is convincing, and the authors have made a remarkable effort to include all statistics and detailed description of the methods. Nevertheless, compared to their previous work, this paper contains fewer surprising results, and still leaves many of the previously unanswered questions open.Major points:1) Discussion section: the authors write 'However, precisely mapping the PG localization in the brain and identifying the exact path followed by this gut-born bacteria metabolite to reach the brain will require PGN tracing methods which are not yet available.' While I agree that this specific experiment is difficult, the paper would be strengthened significantly by the demonstration that PGN activates PGRP-LB expressing neurons in the brain. This can be achieved by ex vivo or in vivo Ca-imaging or methods such as CalexA or CaMPari. The lack of knowledge of how PGN or bacteria recruit PGRP-LB signaling in the brain is, in my opinion, the most significant weakness of this study. It appears to me that this could be addressed in a short time frame of a couple of months. In lieu of such evidence, the link between PGN and egg laying-regulating LB1-Gal4+ neurons in the brain remains weak.

We agree with this reviewer that experiments showing a direct effect of PGN on pLB1+ neuronal activity will help us understand how PGN is regulating fly behavior. We had indeed intensively tried to combine pLB1-Gal4 driver with various Ca sensors (GCamp, CaMPari….) to test this. However, the pLB1-Gal4 driver is very weak and did not allow us to see any fluorescent expression of these brains before PGN stimulation making it impossible to monitor Calcium level after PGN stimulation. We circumvented this problem by two means.

- First, we teamed up with the laboratory of Yäel Grosjean in Dijon, France, to measure the effects of PGN stimulation on Tdc2 neurons in live flies using GCaMP sensor. Our results show that stimulation of brain of Tdc2G4-UAS-GCaMP6s with 100 µg/mL of PGN induces a decrease of GFP staining in VMII/III Tdc2 positive neurons. These new results are now presented in Figure 8.

- To precisely narrow down the effects of PGN on pLB1+/Tdc2+ neurons, we generated a new fly line, pLB1-lexA using the exact same DNA fragment that the one used to produce pLB1-Gal4 fly line. When used in combination with Tdc2-Gal4, we were able to detect GCaMP expression in pLB1+/Tdc2+ neurons. Using an ex vivo approach, dissected brains in which GCamp6S is expressed only in pLB1+/Tdc2+ neuron were stimulated with PGN. New results presented in Figure 8 show that PGN stimulation induces a drop of calcium levels in pLB1+/Tdc2+ VMIII neuron.

-Altogether these data show that PGN stimulation ofpLB1+/Tdc2+ VMIII neuron induces a drop of intracellular calcium levels in these cells and hence probably inhibits these neurons. Consistently, our genetic experiments showed that functional pLB1+/Tdc2+ neuron inactivation (via Kir.2.1 or TTx overexpression) also led to egg-laying drop. This suggested a model in which PGN recognition in brain pLB1+/Tdc2+ cell, inhibits this neuron, resulting in temporary egg-laying drop.

2) Discussion section: the authors state that '2 brain and 10 VNC neurons are reproducibly expressing the PGN degrading enzyme PGRP-LB'. I assume this is based on the expression of pLB1-Gal4, which was used in Kurz et al., to rescue the PGRP-LB loss of function phenotype. Can the authors be 100% sure that all pLB1-Gal4 expressing cells truly express PGRP-LB without an antibody staining?

- As mentioned by the reviewer, our previous *eLife* paper demonstrated isoforms specific rescue of PGRP-LB mutant egg-laying deficit post infection by pLB1-Ga4 mediated expression of PGRP-LB-RD. This indeed suggested that the pLB1-Gal4 driver is, at least partially mimicking endogenous expression of the PGRP-LB-RD isoform. To further and more directly demonstrated it, we generated two new fly lines. pLB1-LexA already mentioned above. When crossed to appropriate fluorescent reporters, this line highlighted the brain neurons already identified with pLB1-Gal4. In addition, two AL2/Tdc2 + neurons were labelled with this line. The second fly line that we generated is PGRP-LB::GFP in which the GFP cDNA was inserted, via CRISPR, at the endogenous PGRP-LB locus. Since the GFP is inserted in an exon shared by all isoforms, it is expected that PGRP-LB-RA, RC, and RD isoforms will be GFP-tagged. By imaging PGRP-LB::GFP brains, we revealed that some VM and AL2 octopaminergic neurons are clearly expressing endogenous PGRP-LB protein (new Figure 7). This further confirmed that the pLB1 (Gal4 and LexA) drivers are expressed in at least some of the cells that produce PGRP-LB. Few cells were also identified that were PGRP-LB::GFP positive and Tdc2 -, showing that pLB1 labels only a subset of PGRP-LB positive cells in the brain.

Reviewer #2:In a previous (Kurz and Royet, 2017) elegant study, Kurz and Royet reported that internal sensing of bacterial PGN by a group of PRGP-LB expressing neurons are responsible for reducing female oviposition. They observed that PRGP-LB neurons are likely octopaminergic, and activation of NF-κB in these neurons are important for reduction in oviposition. Furthermore, they reported that inactivation of these PRGP-LB positive neurons by tetanus toxin, or activation by temperature sensitive TRPA1 channel can modulate egg laying. These observations raised two important questions: how octopaminergic neurons are activated by PGN, and how activation of these neurons in the CNS/PNS influence oviposition. In this paper they have taken the first step towards addressing these questions. Taking advantage of the genetic tools available in Drosophila they have beautifully isolated a small subset (in fact could be a single neuron) of neurons in the octopaminergic VM cluster in SOG that is necessary for the PGN-mediated alteration in oviposition.Major points:While the observations reported in this paper are interesting, one can argue that it does not provide any new insight to the problem than what they have already published. Nonetheless, if indeed a single neuron can mediate PGN-dependent alteration of a very specific behavior, I think that should be worth publishing. However, to make that conclusion the authors at least need to do the following experiments and these experiments should not take more than two months. First, activation of just the PLB1+/ Tdc2+ neurons using TrpA1 or channelrhodopsin to test whether activation of these few neurons is enough to recapitulate the behavior. Other than few examples of 'command neurons", there are not too many examples where activation of one or two neurons recapitulates a specific behavior.

- As requested by this reviewer, we have performed functional experiments in pLB1+/Tdc2+ neurons. Our previous data (Kurz et al., 2017) indicated that expression of the temperature sensitive TRPA1 channel in pLB1 + cells was sufficient to trigger an egg-laying boost. Using intersectional genetic strategy, we now showed that TRPA1 expression in pLB1+/Tdc2 + neurons produces similar results.

- In conclusion, when either activated or inhibited, pLB1 +/Tdc2+ neurons behave as pLB1 + cells as far as egg-laying behavior is concerned.

Second, if the model is correct than the activity of these PLB1+/Tdc2+ neurons in SOG should be influenced by PGN. Again, using the same driver line, they should express calcium sensor GCamp and monitor activity as a change in fluorescence following application of PGN. The kinetics of the response would also carry valuable information. They can perform these experiments in isolated brain preparation with appropriate control.

We agree with this reviewer that experiments showing a direct effect of PGN on pLB1 neuron activity will help us understand how PGN is regulating neuronal activity. We had indeed intensively tried to combine pLB1-Gal4 driver with various Ca sensors (GCamp, CaMPari….) to test this. However, the pLB1-Gal4 driver is very weak and did not allow us to see any fluorescent expression of these brains before stimulation making it impossible to monitor Calcium level after PGN stimulation. We circumvented this problem by two means.

- First, we teamed up with the laboratory of Yäel Grosjean in Dijon to measure the effects of PGN stimulation on Tdc2 neurons in live flies using GCaMP sensor. Our results show that stimulation of brain of Tdc2G4-UAS-GCaMP6s with 100 µg/mL of PGN induces a decrease of GFP staining in VMII/III Tdc2 positive neurons.

- To precisely narrow down the effects of PGN on pLB1 +/Tdc2 + neurons, we generated a new fly line, pLB1-lexA using the exact same DNA fragment that the one used to generate pLB1-Gal4 fly line. When used in combination with Tdc2-Gal4, we were able to detect GCaMP expression in pLB1/Tdc2 neurons. Using an ex vivo approach, dissected brains in which GCamp6S is expressed only in pLB1 + /Tdc2 + neuron were stimulated with PGN. New results presented in Figure 8 show that PGN stimulation induces a drop of calcium levels in pLB1 +/Tdc2+ VMIII neuron.

Altogether these data show that PGN stimulation of pLB1+/Tdc2+ VMIII neuron induces a drop of intracellular calcium levels in these cells and hence probably inhibits these neurons. Consistently, our genetic experiments showed that functional pLB1+/Tdc2+ neuron inactivation (via Kir.2.1 or TTx overexpression) also led to egg-laying drop. This suggested a model in which PGN recognition in brain pLB1+/Tdc2+ cell, inhibits this neuron, resulting in temporary egg-laying drop.

Finally, I would like to know if they have attempted to follow the output of these neurons from SOG or whether they have any clue to where it is going.

The pLB1-Gal4 driver that has been used throughout our previous paper (Kurz et al., 2017) and in this manuscript is very weak. Although we have tried very hard to combine it with all the possible UAS-fluorochromes, we had a very hard time to follow the neuronal projections of pLB1-Gal4 cells. To overcome this limitation, we have generated a new pLB1-LexA fly line using the exact same DNA fragment that the one used to generate pLB1-Gal4 fly line. When used in intersectional strategy to unlock the strong Tdc2-Gal4 driver, this line highlights the brain neurons already identified with pLB1-Gal4. In addition, two AL2/Tdc2 + neurons were labelled with this line. Interestingly, this line allowed us to see some neuronal projections of pLB1/Tdc2 cells which helped us to get the exact identity of the pLB1+/Tdc2 + neurons taking advantages of previously reported brain octopaminergic neuron map.

- In addition, this new tool allowed us to identify neurites coming from the pLB1/Tdc2 VMIII neurons and descending via the cervical connectives toward the VNC (New Figure 6).

Reviewer #3:Masuzzo et al., present their work, entitled, "Peptidoglycan-dependent NF-κB activation in a small subset of brain octopaminergic neurons controls female oviposition." In this study, the authors build upon their previous work (Kurz et al., 2017), which demonstrated that internal sensing of bacterial peptidoglycan reduces Drosophila female oviposition via NF-κB pathway activation in some neurons. However, until this paper, the neurons remained unknown. This study identifies several ventral nerve cord and brain octopaminergic neurons, expressing a NF-κB pathway components. Furthermore, the authors find that the brain neurons, specifically in the sub-esophageal zone, are required for oviposition depression, whereby activation triggers an egg-laying drop in response to infection. This was shown via peptidoglycan exposure or inhibition of PGRP-LB+ neuron activity, which subsequently impairs a specific ovulation event, thus linking PGN detection to the reduction of egg production.The study uses particular GAL4 driver lines, i.e. pLB1, to observe a neuron requirement of TbBh-IR (Tyrosine decarboxylase) (Figure 1G), where when knocked down, flies oviposit fewer eggs. Using this finding, the authors stained against Tdc2 in brains and ventral nerve cords expressing pLB1>GFP to find the particular neurons of images. The authors then tested against other types of neuro peptides to rule out effects of other neuronal subtypes. While this demonstrated a series of negative results, they are very important to the identity of these cell types.The authors then turn to identifying what pLB1 neurons do by comparing them to previously identified Tdc2 neurons. They authors do this inactivating them with UAS-Kir and find increased receptivity in female flies with marked Tdc2 neurons but not pLB1 neurons. Interestingly, pLB1+/Tdc2+ neurons were also detected in males, even though an egg laying response is the output (Figure 3C-D). To test functional significance, an intersectional line was generated and tested in an inducible manner via GAL80 and Tetx and kir, yielding strong reduction of oviposition. This was very elegantly performed in a temperature and temporally controlled manner. Additionally, the authors find that suppression of egg-laying drop 6 hours post PGN injection mediated by the knockdown of the NF-ΚB cascade in Tdc2+ and pLB1+ cells was not observed when only thoracic neurons were targeted, demonstrating functional relevance. The authors take this a step further and analyze the ovary, where they observe changes in follicle cell trimming (marked decrease in animals with PLb1GAL4>UASKir2.1) as the means by which fewer eggs are laid. This showed a remarkable increase in stage 14 oocyte numbers when pLb1 neurons were inhibited.This is an extremely elegant and important study that highlights how a few neurons can respond to external stimuli and change subsequent internal physiology, in an octopamine-dependent manner. Flies are able to tune their responses and behavior by bacteria-derived metabolite! The manuscript is well written and reads as a story with a very logical flow. Additionally, I would like to commend the authors on the images, which are very high quality and I really enjoy having the cartoon depiction of the brains to accompany images (i.e. Figure 1).Major comments:I am left wanting more with respect to Figure 5 with respect to permanence or reversibility of the phenotype. The evidence that follicle cell trimming is altered and that there is an accumulation of stage 14 egg chambers is compelling. However, I am left wondering what the physiological relevance is to this particular phenotype. I would like to know if the eggs are simply delayed (diapause phenotype) and that these eggs will be laid, or perhaps they will not be laid at all? Thus, I would like to see an experiment that turns pLB1 GAL4 marked neurons back on (i.e. use UAS-shibireTS to turn off neurons to get ovary effect at 30C, then move them back to 22C and see how egg counts change) in order to see what the dynamics are of these eggs that are not yet laid. Alternatively, using the Kir construct with a GAL80 for temporal control in the same manner would be excellent. I believe that this experiment, regardless of the answer, will be important to perform to get at the underlying mechanism of what happens to the eggs. Images of ovaries (and associated quantification) after turning the pLB1 neurons back on would also be desired. Thus, can egg counts recover and does the ovary recover are important questions to finish answering.Additionally, I do not see stage 11-12 egg chambers in the images of Figure 5C, but rather only stage 14 and stage 7/8 and below. Panels A and B have these later stage (11) egg chambers and have apoptosis on the DAPI staining, a hallmark of normal development. I would like to see this quantification as well, not on any new experimental animals, but on the existing ovary image. I believe that the authors have identified some really cool developmental biology regulation, which is not pointed out in the manuscript, but that should be highlighted. Perhaps these are not representative images, but the means by which the egg chambers are arrested is important to note. At minimum, the distribution of egg chamber stages should be quantified and talked about in the text. If the distribution is changed as the images suggest, please speculate on mechanism in the Discussion section - the mechanism does not need to be elucidated, just postulated about. This has the potential to be really exciting.I believe that these two points are essential to the manuscript, as they could more fully demonstrate the functional significance what the physiological mechanism is that the neurons are communicating.

- To respond to this reviewer concerns, we have performed two novel experiments.

- The first one is aimed at better characterizing the effects on PGN in the long term, using a kinetic. This allowed us to answer whether the blockage in egg-laying is temporary, delayed or more definitive. For that purpose, females were injected with either PGN or water and the number of oocytes at different stages and the number of apoptotic oocytes were quantified at 6, 24 and 48 hours post injection. The results show that at 6 hours post PGN injection, a slight reduction of stage 10 and 11-13 oocytes associated with an accumulation of stage 14 oocytes. At the same time point, PGN injection is inducing a slight increase of apoptotic oocytes. At 24 hours post PGN injection, the tendency is the same but apoptotic oocytes are no longer observed. At 48 hours post injection, all the measured parameters are identical in water and PGN injected females showing that the effects are fully reversible.

- The second experiment is aimed at testing whether and how a temporary blockage of pLB1 neuron is impacting ovary physiology. For that purpose, the UAS-Kir transgene was combined to the Gal80Ts modulator to temporary inactivate pLB1 cells. When pLB1-Gal4, UAS-Kir, Gal80ts flies were shifted from 21 to 29°C, an accumulation of stage 14 oocytes was observed. While this phenotype was still present 24 hours after shifting the flies back to 21°C, this was no longer the case after 48 hours. It should be mentioned that no sign of apoptosis was detected in these experiments. These experiments show that pLB1 neuron inactivation mimics most phenotypes observed when PGN is injected and that this phenomenon is reversible (New Figure 9, Figure 9—figure supplement 1 and Figure 10).

[Editors' note: further revisions were requested prior to acceptance, as described below.]

Thank you for submitting your article "Peptidoglycan-dependent NF-κB activation in a small subset of brain octopaminergic neurons controls female oviposition" for consideration by eLife. Your article has been reviewed by Utpal Banerjee as the Senior Editor, a Reviewing Editor, and three reviewers. The following individuals involved in review of your submission have agreed to reveal their identity: Balint Z Kacsoh (Reviewer #1); Kausik Si (Reviewer #3).The reviewers have discussed the reviews with one another and the Reviewing Editor has drafted this decision to help you prepare a revised submission We are pleased to inform you that the reviewers were rather positive about the manuscript and suggest a number of revisions that you should try to address. I suspect that most of them involve text modification.Summary:Masuzzo et al., present their work, entitled, "Peptidoglycan-dependent NF-κB activation in a small subset of brain octopaminergic neurons controls female oviposition." In this study, the authors build upon their previous work (Kurz et al., 2017), which demonstrated that internal sensing of bacterial peptidoglycan reduces Drosophila female oviposition via NF-κB pathway activation in some neurons. Here, they further characterize the very specific neuronal circuitry, involved in the immune induced oviposition behaviours phenotype. This study now is a major achievement in the burgeoning field of the intersection of neuronal signaling and immunity.Essential revisions:1) In Figure 8, the authors now show the effect of PGN stimulation on GCaMP fluorescence in vivo and ex vivo using either a broad Tdc2-Gal4 driver or their specific intersectional approach. In Figure 8A, the authors provide images of Tdc2 > GCaMP before and after PGN addition. Unfortunately, from these images it is unclear what we are looking at exactly given that Tdc2 labels many neurons. At minimum the authors should provide a scheme to explain the figure.

We added a schematic representation of the brain and of the plane of imaging explaining the type of neurons we are imaging. There is a frontal view pointing to the VM cluster and a sagittal view detailing the sub-cluster. It is now the Figure 8—figure supplement 1 and the reference and legend had been added in the main text.

2) The authors point out that PGN reduces GCaMP fluorescence very transiently in their in vivo experiments. In the ex vivo experiment however, this is not the case. Here, the inhibition appears permanent/lasting. What explains this difference?

We suggest some possibilities in the text to explain the differences, directly below the description of the corresponding results.

3) The authors speculate which genes might be regulated by PGN in LB1 neurons. They do not mention, however, how PGN might regulate calcium levels in these cells. This should be added to the Discussion section given that this is likely the main reason for the behavioral phenotype they observe.

We added a section in the Discussion section to propose possible mechanisms involved. This remains speculative at this point.